# One Initialization to Rule them All: Fine-tuning via Explained Variance Adaptation

## Abstract

Foundation models (FMs) are pre-trained on large-scale datasets and then fine-tuned on a downstream task for a specific application. The most successful and most commonly used fine-tuning method is to update the pre-trained weights via a low-rank adaptation (LoRA). LoRA introduces new weight matrices that are usually initialized at random with a uniform rank distribution across model weights. Recent works focus on *weight-driven* initialization or learning of adaptive ranks during training. Both approaches have only been investigated in isolation, resulting in slow convergence or a uniform rank distribution, in turn leading to sub-optimal performance. We propose to enhance LoRA by initializing the new weights in a *data-driven* manner by computing singular value decomposition (SVD) on minibatches of activation vectors. Then, we initialize the LoRA matrices with the obtained right-singular vectors and re-distribute ranks among all weight matrices to explain the maximal amount of variance across layers. This results in our new method **E**xplained **V**ariance **A**daptation (EVA). We apply EVA to a variety of fine-tuning tasks ranging from language generation and understanding to image classification and reinforcement learning. EVA exhibits faster convergence than competitors and attains the highest average score across a multitude of tasks per domain while reducing the number of trainable parameters.

## 1 Introduction

Foundation models (Bommasani et al., 2021, FMs) are usually trained on large-scale data and then fine-tuned towards a particular downstream task. This training paradigm has led to significant advancements in the realm of language modeling (OpenAI, 2023; Touvron et al., 2023a; Reid et al., 2024), computer vision (Dehghani et al., 2023; Oquab et al., 2023), and reinforcement learning (Brohan et al., 2023; Zitkovich et al., 2023). With an increasing number of model parameters, the process of fine-tuning becomes prohibitively expensive. This results in the need for efficient alternatives to fine-tuning *all* parameters of the pre-trained model.

Parameter-efficient fine-tuning (PEFT) approaches are commonly used as an effective alternative to full fine-tuning (FFT). PEFT methods modify the pre-trained model by introducing a small number of new trainable parameters, while the pre-trained weights remain frozen. This leads to a substantial reduction in computational cost, both in terms of time and space. A particularly successful approach, LoRA (Hu et al., 2022), introduces new weights in the form of a low-rank decomposition for each weight matrix in the pre-trained model. After training, the new weights can be readily merged into the pre-trained weights without any additional inference latency. Recent research has explored various extensions to LoRA, such as different initialization schemes and adaptive rank allocation (see Table 1). Weight-driven initialization schemes are constrained to the information stored in the pre-trained weights. Further, adaptive rank allocation techniques usually optimize the ranks during the fine-tuning process which results in additional complexity for computing importance scores of ranks. Both approaches have merely been investigated in isolation thus far.

We propose a new method that extends LoRA with adaptive rank allocation and data-driven initialization by leveraging information from the downstream task. During the fine-tuning process, information of the downstream task is stored in the newly introduced weights of LoRA. Our aim is to make fine-tuning more efficient by initializing the LoRA weights in a manner such that they already contain the maximum possible amount of information from the downstream task. This way, the fine-tuning

Figure 1: **Left:** We perform incremental SVD on activation vectors for the first $T$ minibatches to obtain the right singular vectors. **Middle:** We sort all right-singular vectors according to their explained variance given by their respective singular values and only keep the top-k. **Right:** We allocate the top-k vectors as initialization for $A$ and continue the standard LoRA fine-tuning procedure.

process is more efficient as it only needs to be learned what information to maintain or discard which results in faster convergence and improved downstream performance. We can obtain an initialization that is optimal in propagating the most amount of information into the linear subspace spanned by LoRA via SVD on activation vectors after passing minibatches of downstream data through the model. The right-singular vectors obtained by SVD represent the projection onto the principal components, and their corresponding singular values quantify each component's contribution to the total variance. We initialize the downprojection of LoRA with those vectors to obtain an initialization that propagates the most information of the downstream data. Given a fixed rank budget, we maximize the information propagated through the model by sorting the vectors in descending order according to their singular values and allocate the top-k vectors to their respective weight matrices. This results in an adaptive rank allocation that can be computed at the beginning of training which allocates more complexity to weights where components explain less variance. We call the resulting method EVA, which is short for **E**xplained **V**ariance **A**daptation. Importantly, this procedure can be performed within the first few minibatches of LoRA fine-tuning without significant computational overhead.

We demonstrate the benefits of EVA on an array of downstream tasks, namely language generation and understanding, image classification, and reinforcement learning (RL). EVA consistently improves average performance across a multitude of tasks on each domain compared to LoRA and other recently proposed initialization or rank redistribution methods. For language generation, we fine-tune 7B-9B parameter language models on math and reasoning tasks, where EVA attains the highest average performance. Further, on a set of language understanding tasks, EVA improves the average performance compared to competitors. On image classification we fine-tune a pre-trained vision transformer (Dosovitskiy et al., 2021) on a set of 19 diverse tasks. We find that EVA attains the highest average score and improves over LoRA and established extensions thereof, with most gains on in-domain data. For our RL experiments we conduct fine-tuning on continuous control tasks and find that EVA significantly exceeds performance of LoRA and even exceeds performance of full fine-tuning (FFT) when combined with DoRA (Liu et al., 2024a). Finally, we demonstrate that EVA is pareto-dominant as our rank re-distribution reduces the amount of trainable parameters while improving performance. Our contributions are as follows:

- We propose a novel data-driven initialization scheme for LoRA by leveraging incremental SVD on minibatches of activation vectors.
- We propose a data-driven heuristic for adaptive rank allocation based on explained variance.
- We demonstrate the effectiveness of EVA across a variety of different domains.

## 2    RELATED WORK

**LoRA** (Hu et al., 2022) has sparked widespread interest in leveraging low-rank decompositions for fine-tuning due to its simplicity. Building on the success of LoRA, a number of other variants have been proposed (Kopiczko et al., 2024; Zi et al., 2023; Babakniya et al., 2023; Dettmers et al., 2023; Li et al., 2023; Nikdan et al., 2024; Liu et al., 2024a; Zhang et al., 2023a; Hayou et al., 2024; Chavan

Table 1: Comparison of EVA to existing initialization schemes for LoRA. Existing works either focus on weight initialization *or* adaptive rank allocation. EVA **combines** data-driven initialization with adaptive rank allocation to enhance convergence and downstream performance.

| Method | Initialization | Adaptive ranks |
|---|---|---|
| LoRA (Hu et al., 2022) | Random | ✗ |
| AdaLoRA (Zhang et al., 2023a) | Random | ✓ |
| PiSSA (Meng et al., 2024) | Weight-driven | ✗ |
| OLoRA (Büyükakyüz, 2024) | Weight-driven | ✗ |
| LoRA-GA (Wang et al., 2024) | Data-driven | ✗ |
| EVA (Ours) | Data-driven | ✓ |

et al., 2023). The most similar variants to EVA are AdaLoRA (Zhang et al., 2023a) and LoRA-GA (Wang et al., 2024). AdaLoRA adaptively alters the number of ranks for LoRA matrices during fine-tuning. Other more recent approaches learn gates to switch ranks on or off during fine-tuning (Liu et al., 2024b; Meo et al., 2024). In contrast, the data-driven initialization allows EVA to redistribute ranks for each LoRA matrix prior to fine-tuning. LoRA-GA is concurrent work that approximates the gradient of the original weight matrix via SVD, requiring computation of the gradients with respect to the original weights. Contrary, EVA initializes $A$ via the right-singular vectors of minibatches of activation vectors, and is therefore less computationally expensive.

**Initialization of LoRA matrices** Common initialization schemes for neural networks (He et al., 2015; Glorot & Bengio, 2010) were designed to stabilize training of deep neural networks based on activation functions and depth. In the context of PEFT, Hu et al. (2022) and Liu et al. (2022) explored data-driven initialization by either pre-training on a different task first, or by unsupervised pre-training on the task at hand. Contrary, EVA does not require any gradient update steps, therefore it is much more efficient. Similarly, Nikdan et al. (2024) utilize a warm-up stage in LoRA fine-tuning, where gradients with respect to LoRA weights are used to initialize a sparse matrix for sparse adaptation (Sung et al., 2021) in combination with LoRA. Alternatively, Babakniya et al. (2023) initialize LoRA matrices using SVD on weight matrices obtained after a few steps of full fine-tuning for federated learning with heterogeneous data. Meng et al. (2024) use the main directions of the pre-trained weights to initialize the LoRA matrices. In contrast, EVA takes a data-driven approach to initialize the LoRA matrices. Similar initialization schemes were proposed for training deep networks from scratch (Mishkin & Matas, 2016; Krähenbühl et al., 2016).

**Increasing efficiency of LoRA** Several works have investigated how to increase efficiency of LoRA fine-tuning. Kopiczko et al. (2024) decrease the memory complexity by keeping both $A$ and $B$ frozen while merely training newly-introduced scaling vectors. This way, only random seeds for initializing $A$ and $B$ need to be stored. Another prominent approach is quantization (Dettmers et al., 2022), which has been successfully combined with LoRA (Dettmers et al., 2023). Other LoRA variants are compatible with quantization (Nikdan et al., 2024; Valipour et al., 2023; Meng et al., 2024). It has also been shown that initialization can improve fine-tuning quantized models (Li et al., 2023).

## 3 METHOD

We aim at initializing LoRA weights in a data-driven manner by leveraging data from the downstream task. Since EVA builds on LoRA (Hu et al., 2022), we first briefly explain LoRA in Section 3.1. Then, we explain the two essential steps conducted in EVA, namely (i), computing a data-driven initialization for the low-rank decomposition of LoRA matrices via SVD on activation vectors (Section 3.2), and (ii), adaptive assignment of ranks across all layers to maximize the explained variance throughout the pre-trained model (Section 3.3).

### 3.1 LOW-RANK ADAPTATION (LORA)

LoRA adds new trainable weights which are computed via an outer product of low-rank matrices (Hu et al., 2022). This is motivated by the low intrinsic dimensionality of language models (Aghajanyan

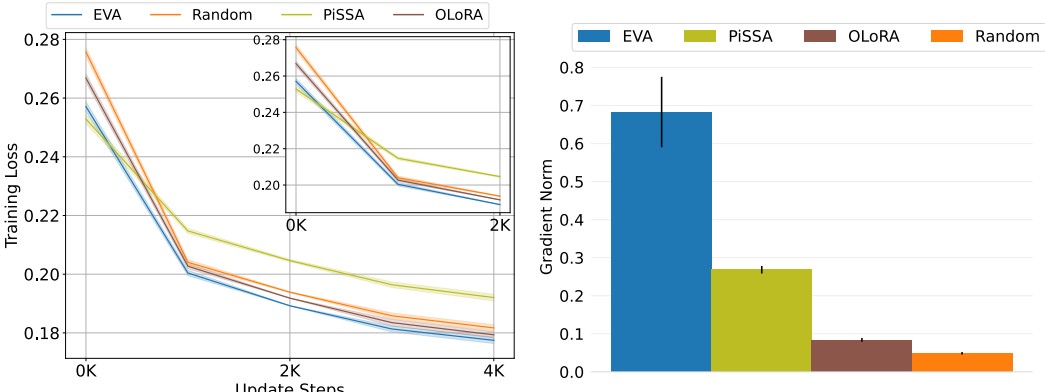

Figure 2: **Left:** Training loss for fine-tuning Llama-3.1-8B on the MetaMathQA dataset. We compare EVA to other initialization methods OLoRA, PiSSA, and random initialization (LoRA). We show mean and standard deviation across three random seeds. **Right:** Mean and standard deviation of gradient norm at the beginning of training for EVA, PiSSA, OLoRA and Random initialization of LoRA matrices. EVA exhibits significantly larger gradient norm.

et al., 2021) and relies on the assumption that the gradients during fine-tuning are also of low rank (Gur-Ari et al., 2018; Zhang et al., 2023b; Gauch et al., 2022). Let $x \in \mathbb{R}^{d \times 1}$ be the input to a pre-trained weight matrix $W \in \mathbb{R}^{k \times d}$. Then, LoRA introduces new weight matrices $A$ and $B$ as a low-rank decomposition $h = Wx + BAx$, where $B \in \mathbb{R}^{k \times r}$ and $A \in \mathbb{R}^{r \times d}$. The rank $r$ is a hyperparameter with $r \ll k$. During fine-tuning, $W$ remains frozen while $A$ and $B$ are updated. Usually, $B$ is initialized with zeros and $A$ at random, such that fine-tuning starts from the pre-trained model. Additionally, a hyperparamter $\alpha$ is used to scale $BAx$ by $\frac{\alpha}{r}$.

### 3.2 DATA-DRIVEN INITIALIZATION OF LOW-RANK ADAPTATION

Our aim is to obtain an effective initialization for $A$ to find a linear subspace that preserves the most information of the downstream task, i.e. that explains the most variance. To this end, we perform SVD on batches of activation vectors $X \in \mathbb{R}^{b \times d}$ to obtain the right-singular vectors, which constitute the directions that capture most of the variance (see Figure 1, left). More formally, we collect batches of activations $X^i$ for $N$ pre-trained weight matrices $W^i \in \{W^1, ..., W^N\}$ that are selected for fine-tuning. Subsequently, we compute the SVD on each $X^i$ to obtain the right-singular vectors $v^i_{j,:}$ and their respective singular values $\sigma^i_j$ as

$$X^i = U^i \Sigma^i V^{i\top} \approx \sum_{j=1}^{k} u^i_{:,j} \sigma^i_j v^i_{j,:} \tag{1}$$

Here, $U$ and $V$ are the left- and right-singular vectors, respectively, and $\Sigma$ is a diagonal matrix containing the singular values. Note that in practice we compute only the top-k components and not the full SVD using truncated SVD (Halko et al., 2011) which is the optimal approximation of $X^i$ as verified by the Eckart-Young theorem (Eckart & Young, 1936). Generally, the stacked right-singular vectors $V^i_{:r,:}$ are equivalent to a projection onto the principal components of the covariance matrix of $X^i$ (see proof in Appendix H). Therefore, $V^i_{:r,:}$ propagates the maximum amount of information of $X^i$. By setting $A^i = V^i_{:r,:}$ the downprojection $X^i A^i$ must contain the most information about $X^i$ according to the data processing inequality (Beaudry & Renner, 2012), as the maximum amount of information $B$ can contribute is $B^i = V^{i\top}_{:r,:}$. The gradient w.r.t. $A^i$ and $B^i$ is

$$\frac{\partial \mathcal{L}}{\partial B^i} = \frac{\partial \mathcal{L}}{\partial W} A^{i\top} \quad \text{and} \quad \frac{\partial \mathcal{L}}{\partial A^i} = B^{i\top} \frac{\partial \mathcal{L}}{\partial W}, \tag{2}$$

respectively. The fine-tuning process is concerned with storing information about the data in the weights $B^i A^i$. By choosing $A^i = V^i_{:r}$ we guarantee that the maximum amount of information is available at the beginning of training, such that it only needs to be learned what information to keep, i.e. what parts of $X^i A^i$ are relevant for the downstream task.

Naively, we could simply collect batches of activations and stack them into a single matrix and perform SVD. However, this results in excessive memory overhead as we usually deal with large datasets and models. To reduce the memory requirements, we incrementally update $V_{:r,:}^i$ as proposed in Ross et al. (2008) which is based on the sequential Karhunen-Loeve algorithm (Levy & Lindenbaum, 2000). This process is independent of the dataset size, therefore the computation of the singular values and their respective vectors is constant in time and memory complexity. For further details on the incremental update step of the SVD we refer to Appendix F.

After each update step in the incremental SVD we check whether $V^i$ has converged via cosine similarity, i.e. $\text{cossim}(v_{j,:}^{i,t-1}, v_{j,:}^{i,t}) \geq \tau \quad \forall \quad 1 \leq j \leq r$. Then, we initialize $A^i = V_{:r,:}^i$ and stop computing incremental SVD for inputs to $W^i$. We continue this procedure until all $V_{:r,:}^i$ have converged. We illustrate the full incremental SVD procedure on a sequence of data batches in Algorithm 2 and discuss complexity of this procedure in Appendix F.

### 3.3 ADAPTIVE RANK ALLOCATION

The singular values provide an estimate of the amount of variance each component in $V_{:r,:}^i$ explains. Leveraging this, we can redistribute ranks across weight matrices of the pre-trained model such that the maximum amount of variance is explained. This can be done by allocating more ranks to layers that propagate more information, i.e., explain more variance. The variance explained by each component in $V_{:r,:}^i$ is given by their explained variance ratio

$$\xi_j^i = \frac{\sigma_j^{i^2}}{(M-1)||\boldsymbol{\sigma}^i||_1}, \qquad (3)$$

---

**Algorithm 1** Fine-tuning via EVA

**Input:** FM $\psi(\cdot)$, $\rho$, rank $r$, dataset $\mathcal{D}$
1: **while** `not all_converged`$(\psi)$ **do**
2: $\quad X \leftarrow \psi(\text{next}(\mathcal{D}))$ $\qquad \triangleright$ get activations
3: $\quad V_{\text{new}}, \boldsymbol{\xi} \leftarrow \text{Incremental-SVD}(X, \rho r)$
4: $\quad$ **if** `isclose`$(V_{\text{old}}, v_{\text{new}})$ **then**
5: $\quad\quad$ `wrap_and_initialize`$(W_j, V_{\text{new}})$
6: $\quad$ **end if**
7: $\quad V_{old} \leftarrow V_{new}$
8: **end while**
9: `redistribute_ranks`$(\psi, \boldsymbol{\xi}, V_{\text{new}})$
10: `lora_finetune`$(\psi, X)$

---

where $|| \cdot ||_1$ denotes the $\ell_1$ norm, $\boldsymbol{\sigma}^i$ is a vector containing all $r$ singular values, and $M$ is the total number of samples used for the incremental SVD. We sort the components $v_{j,:}^i$ for each weight matrix in descending order according to their explained variance ratio $\xi_j^i$ (see Figure 1, middle). Then, we assign the top-k components to their respective pre-trained weights, which results in adaptive rank allocation (see Figure 1, right). Additionally, we introduce a hyperparameter $\rho \in [1, \infty)$ which controls the uniformity of the rank distribution. $\rho$ determines the number of ranks that we compute during SVD and increasing $\rho$ allows for an increasingly heterogeneous rank distribution. Further, $\rho$ controls the maximum number of ranks a weight matrix can receive. For each $W^i$ we compute $r\rho$ components, i.e., we assign $k = r\rho$ in Equation (1), resulting in $Nr\rho$ components in total. For the redistribution we only use the top-$l$, with $l = Nr$, components according to their explained variance ratio $\xi_j^i$. Thus, setting $\rho = 1$, results in a uniform rank distribution as in LoRA, but initialized according to EVA. Therefore, $\rho$ provides us with the means to change the rank distribution in a controlled manner prior to fine-tuning at the initialization stage. In practice we found that the redistribution converges for values of $\rho > 2$ (see Appendix G). Finally, we initialize $B$ with zeros and perform standard LoRA fine-tuning. In Algorithm 1 we provide pseudocode for EVA.

## 4 EXPERIMENTS

First, we elaborate on implementation details of EVA in Section 4.1. Then, we show results for fine-tuning large language models (LLMs) on math and reasoning tasks in Section 4.2 and language understanding tasks in Section 4.3. Further we show results for image classification in Section 4.4 and decision making tasks in Section 4.5. Finally, in Section 4.6 we demonstrate that the computational overhead induced by EVA over LoRA is negligible and that incremental SVD converges and is invariant to batch order and batch size.

### 4.1 IMPLEMENTATION DETAILS

We follow the standard LoRA training procedure from Hu et al. (2022). Similar to Kalajdzievski (2023), we found LoRA training to be very sensitive to the scaling parameter $\alpha$. Therefore, we set

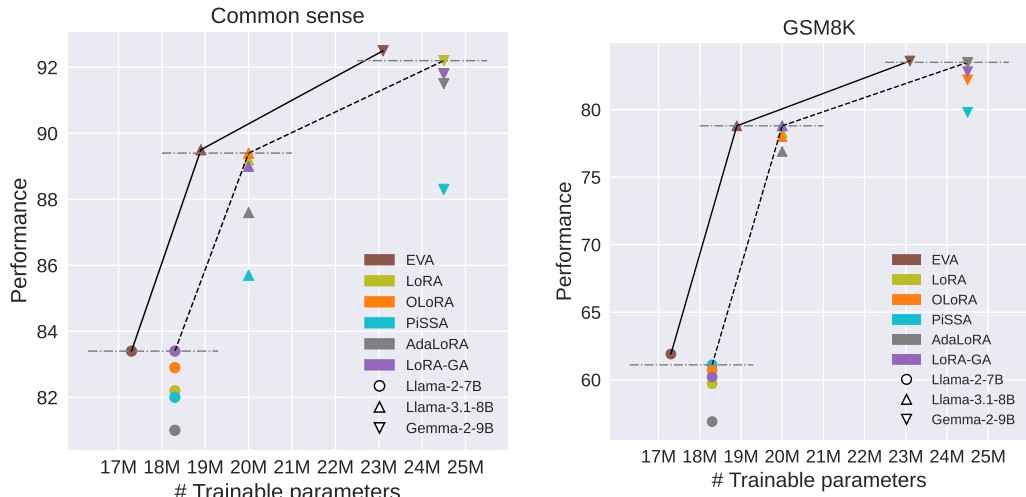

Figure 3: Performance of EVA, OLoRA, PiSSA, LoRA-GA, and LoRA for fine-tuning Llama-2-7B, Llama-3.1-8B, and Gemma-2-9B on eight common sense reasoning tasks (left), and MetaMathQA, subsequently evaluated on GSM8K (right).

$\alpha = 1$ for all our experiments as we found this to be the most stable setting and only tuned the learning rate. We apply EVA to pre-trained weights only, i.e., we do not initialize newly introduced classifier heads. Following Zhang et al. (2023a), we apply LoRA adapters to all pre-trained weight matrices except for the embedding layer. For EVA we always search over $\rho \in \{1, 2\}$ to cover both uniform uniform and adaptive rank allocation and report the best score. For $\rho = 2$ we also set $\alpha = \alpha \frac{r_{new}}{r_{old}}$ to preserve the same scaling factor as set initially. All models we used for fine-tuning are publicly available on the huggingface hub (Wolf et al., 2020). For the implementation of baselines we leverage the widely used PEFT library (Mangrulkar et al., 2022). Across experiments we highlight the highest scores in boldface and underline the second-highest.

## 4.2 LANGUAGE GENERATION

We fine-tune three different LLMs, namely Llama-2-7B (Touvron et al., 2023b), Llama-3.1-8B (Dubey et al., 2024), and Gemma-2-9B (Rivière et al., 2024) on common sense and math reasoning benchmarks. For common sense reasoning we follow Liu et al. (2024a) and amalgamate a training set consisting of BoolQ (Christopher et al., 2019), PIQA (Bisk et al., 2020), SIQA (Sap et al., 2019), HellaSwag (Zellers et al., 2019), Winogrande (Sakaguchi et al., 2020), ARC-e and ARC-c (Clark et al., 2018) and OpenBookQA (Mihaylov et al., 2018). We apply all methods listed in Table 1 to all three models and additionally add a comparison to DoRA (Liu et al., 2024a) and EVA+DoRA, which combines EVA with DoRA. We train all methods with rank $r = 16$ and a learning rate of $5e - 4$ for three random seeds. Further details on the fine-tuning settings can be found in Appendix B.

We present average performance over all eight common sense reasoning tasks in Figure 3, left. Across models we found that $\rho = 2$ yields the highest performance while it also notably decreases the number of trainable parameters compared to all other LoRA-based methods (see Table 11 in Appendix B), resultin in an improved pareto-front. For a comparison to EVA with uniform rank distribution see Table 10 in Appendix B. We report the per-task results in Table 7 in Appendix B. Even though there is a fluctuation on a per-task basis, EVA attains the highest average score across all tasks. Moreover, we conduct experiments where we add rank-stabilization (Kalajdzievski, 2023), different learning rates for $A$ and $B$, or different values for $\alpha$ in Table 9 in Appendix B. Additionally, we provide results for leveraging the components that explain the *least* amount of variance in Table 12, which results in worse performance compared to EVA. Finally, EVA as well as EVA+DoRA are consistently among the best performing methods on all individual tasks. This highlights the effectiveness of EVA's data-driven initialization and rank allocation.

For the math fine-tuning experiments, we fine-tune all models on the MetaMathQA dataset (Yu et al., 2024) for one epoch with the same hyperparameters as for the common sense reasoning tasks and evaluate them on GSM8K (Cobbe et al., 2021) (see Figure 3, left) and MATH (Hendrycks et al., 2021) (see Figure 4) datasets. We also report the performances for each method on each model and task in Table 8 in Appendix B. Generally, we again observe that EVA is pareto-dominant compared to all competitors on both datasets as it trains fewer parameters while mostly resulting in improved performance. Specifically, EVA attains the highest performance on the GSM8K dataset for Gemma-2-9B using $\rho = 2$. For Llama-2-7B and Llama-3.1-8B the best performing method is EVA+DoRA using $\rho = 1$ closely followed by EVA. On MATH, EVA+DoRA performs best for Llama-2-7B with $\rho = 1$, while EVA attains the highest score for Llama-3.1-8B with $\rho = 1$ and Gemma-2-9B with $\rho = 2$. For a comprehensive overview on the effect of rank re-distribution on different model types for both downstream tasks see Table 10. Our results indicate that

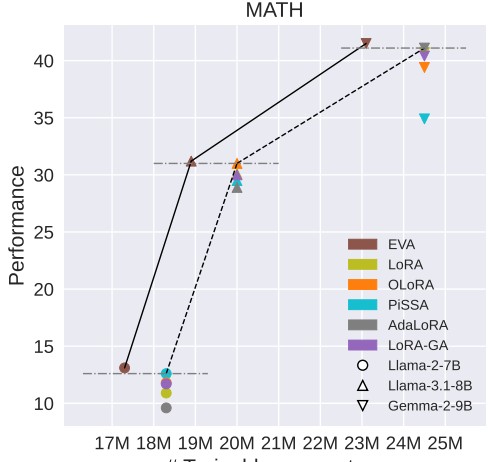

Figure 4: Performance of EVA, OLoRA, PiSSA, LoRA-GA, and LoRA for fine-tuning Llama-2-7B, Llama-3.1-8B, and Gemma-2-9B on MATH after fine-tuning on the MetaMathQA dataset.

the performance of adaptive rank allocation depends on a combination of the selected model and the downstream task. We further analyze the resulting rank distributions for different values of $\rho$ for Llama-2-7B and their effect on downstream performance in Appendix G. Finally, we provide additional results for Llama-2-7B on code fine-tuning tasks in Appendix B.

## 4.3 LANGUAGE UNDERSTANDING

We train RoBERTa_Large (Liu et al., 2019) and DeBERTav3_Base (He et al., 2023) on the GLUE benchmark (Wang et al., 2019). The GLUE benchmark comprises eight downstream tasks, such as natural language inference, or sentiment analysis. Additionally to learning rate, we also search over different ranks within a maximal rank budget ($r \leq 16$). For further details about datasets, implementation, or hyperparameters, we refer to Appendix C. We also add FFT as a baseline, but neglect EVA+DoRA due to time constraints and report Matthew's correlation for CoLA, Pearson correlation for STS-B, and accuracy for the remaining tasks in Table 2. EVA ($\rho = 2$) attains the highest average score across all tasks for both RoBERTa_Large and DeBERTav3_Base. Interestingly, DoRA usually only slightly improves over LoRA on low resource tasks (RTE, MRPC), while performing worse in high resource tasks (MNLI, QNLI, QQP, SST2). We also compare LoRA to EVA in Table 17 in Appendix C for different rank budgets, where EVA consistently improves over LoRA. We visualize resulting rank distribution patterns for different GLUE tasks in Appendix C. More ranks are assigned to higher layers of the query, key, and value projections in the self-attention, while the remaining weights often receive less ranks. This is a consistent pattern for both, DeBERTav3_Base and RoBERTa_Large and in line with the reduced number of trainable parameters for larger models.

## 4.4 IMAGE CLASSIFICATION

We investigate the efficacy of EVA on the VTAB-1K (Zhai et al., 2019) benchmark, which has been widely used to evaluate PEFT methods. VTAB-1K comprises 19 image classification tasks that are divided into natural images, specialized images (medical images and remote sensing), and structured images (e.g. orientation prediction, depth estimation or object counting). We fine-tune a DINOv2-g/14 model (Oquab et al., 2023) that consists of around 1.1B parameters. For implementation details and hyperparameters see Appendix D. Our results are shown in Table 3 and we additionally report error bars in Table 20. EVA and EVA+DoRA with ($\rho = 2$) attain the best and second-best average accuracy across all tasks, respectively. Interestingly, EVA mainly improves over competitors on the natural tasks, i.e. in-domain datasets. LoRA performs best on the specialized tasks and full fine-tuning (FFT)

Table 2: Comparison of all methods for RoBERTa$_{Large}$ (top) and DeBERTav3$_{Base}$ (bottom) on GLUE tasks. We report mean and standard deviation of Matthew's correlation for CoLA, Pearson correlation for STS-B, matched accuracy for MNLI, and accuracy for remaining tasks. For CoLA, RTE, MRPC, and STS-B we average over five seeds and for the remaining tasks over three seeds.

| Method | MNLI | QNLI | QQP | SST2 | CoLA | MRPC | RTE | STS-B | Avg |
|---|---|---|---|---|---|---|---|---|---|
| FFT | 90.2 | 94.7 | **92.2** | **96.4** | 68.0 | 90.9 | 86.6 | 92.4 | 88.93 |
| LoRA | 90.7$_{\pm.1}$ | 94.8$_{\pm.1}$ | 92.0$_{\pm.0}$ | 96.2$_{\pm.3}$ | 69.1$_{\pm.5}$ | 91.1$_{\pm.6}$ | 88.1$_{\pm1.1}$ | 92.3$_{\pm.1}$ | 89.29 |
| AdaLoRA | 90.5$_{\pm.1}$ | 94.8$_{\pm.2}$ | 90.6$_{\pm.1}$ | 96.1$_{\pm.2}$ | 68.2$_{\pm.7}$ | 90.7$_{\pm.6}$ | 84.4$_{\pm.9}$ | 91.8$_{\pm.1}$ | 88.39 |
| PiSSA | 90.1$_{\pm.1}$ | 94.7$_{\pm.0}$ | 91.0$_{\pm.0}$ | 96.1$_{\pm.2}$ | 68.7$_{\pm1.3}$ | 90.4$_{\pm.6}$ | 87.6$_{\pm.5}$ | 92.5$_{\pm.3}$ | 88.89 |
| OLoRA | **90.9**$_{\pm.1}$ | **95.0**$_{\pm.1}$ | 92.0$_{\pm.2}$ | 96.3$_{\pm.3}$ | 69.0$_{\pm1.5}$ | 91.0$_{\pm1.0}$ | 87.9$_{\pm1.2}$ | 92.4$_{\pm.1}$ | 89.32 |
| EVA | 90.8$_{\pm.1}$ | **95.0**$_{\pm.2}$ | 92.1$_{\pm.1}$ | 96.2$_{\pm.1}$ | **69.5**$_{\pm1.4}$ | **91.4**$_{\pm.8}$ | **88.8**$_{\pm1.2}$ | 92.6$_{\pm.1}$ | **89.55** |
| DoRA | 89.5$_{\pm.1}$ | 94.6$_{\pm.1}$ | 89.9$_{\pm.1}$ | 96.1$_{\pm.1}$ | 69.3$_{\pm.8}$ | 91.0$_{\pm.6}$ | 88.4$_{\pm1.2}$ | 92.4$_{\pm.1}$ | 88.90 |
| FFT | 90.1 | 94.0 | 92.4 | 95.6 | 69.2 | 89.5 | 83.8 | 91.6 | 88.28 |
| LoRA | 90.5$_{\pm.1}$ | 94.3$_{\pm.1}$ | 92.4$_{\pm.1}$ | 95.2$_{\pm.3}$ | 72.0$_{\pm1.3}$ | 91.4$_{\pm.7}$ | 88.9$_{\pm.5}$ | 91.7$_{\pm.1}$ | 89.64 |
| AdaLoRA | **90.8** | **94.6** | 92.2 | 96.1 | 71.5 | 90.7 | 88.1 | 91.8 | 89.46 |
| PiSSA | 90.1$_{\pm.3}$ | 94.1$_{\pm.1}$ | 91.8$_{\pm.1}$ | 95.8$_{\pm.1}$ | **72.7**$_{\pm1.7}$ | 90.9$_{\pm.6}$ | 86.5$_{\pm1.2}$ | 91.6$_{\pm.2}$ | 89.19 |
| OLoRA | 90.5$_{\pm.1}$ | 94.4$_{\pm.1}$ | **92.6**$_{\pm.1}$ | 96.2$_{\pm.2}$ | 72.0$_{\pm1.0}$ | 91.6$_{\pm.7}$ | 89.1$_{\pm.9}$ | 92.0$_{\pm.2}$ | 89.80 |
| EVA | 90.6$_{\pm.1}$ | 94.4$_{\pm.1}$ | 92.4$_{\pm.04}$ | 96.2$_{\pm.2}$ | 72.5$_{\pm1.3}$ | 91.8$_{\pm.6}$ | **89.4**$_{\pm.7}$ | 92.0$_{\pm.2}$ | **89.91** |
| DoRA | 89.0$_{\pm.2}$ | 94.1$_{\pm.1}$ | 88.0$_{\pm.1}$ | 94.6$_{\pm.4}$ | 70.3$_{\pm.5}$ | **91.9**$_{\pm.6}$ | 87.8$_{\pm.7}$ | 91.8$_{\pm.1}$ | 88.44 |

Table 3: Fine-tuning DINOv2-g/14 on the VTAB-1K benchmark. Best average performance is highlighted in boldface. We report average accuracy across five seeds.

| | Natural | | | | | | | Specialized | | | | Structured | | | | | | | | Average |
|---|---|---|---|---|---|---|---|---|---|---|---|---|---|---|---|---|---|---|---|---|
| | Cifar100 | Caltech101 | DTD | Flower102 | Pets | SVHN | Sun397 | Camelyon | EuroSAT | Resisc45 | Retinopathy | Clevr-Count | Clevr-Dist | DMLab | KITTI-Dist | dSpr-Loc | dSpr-Ori | sNORB-Azim | sNORB-Ele | |
| FFT | 73.1 | 89.7 | 78.4 | 99.7 | 92.2 | 89.5 | 55.5 | 74.8 | 95.0 | 88.2 | 70.5 | 93.6 | 64.2 | **63.6** | 68.8 | 92.0 | **64.3** | **50.2** | **56.8** | 76.8 |
| LoRA | 85.9 | 92.2 | 82.2 | 99.7 | 94.5 | 64.1 | **63.6** | **88.8** | **97.0** | **92.6** | 76.6 | **97.7** | 65.3 | 62.1 | 83.6 | 90.6 | 63.0 | 37.1 | 52.3 | 78.4 |
| AdaLoRA | 85.4 | 92.5 | 81.4 | 99.7 | 95.2 | 90.5 | 62.2 | 87.1 | 96.4 | 91.2 | 76.6 | 94.4 | 64.4 | 60.3 | 83.7 | 85.4 | 61.0 | 32.9 | 46.0 | 78.2 |
| PiSSA | 85.5 | 93.6 | 82.3 | 99.7 | 94.6 | 92.8 | 62.3 | 87.1 | 96.6 | 91.9 | 76.3 | 95.0 | 66.3 | 63.2 | 84.9 | 90.5 | 60.1 | 36.3 | 48.6 | 79.4 |
| OLoRA | 85.5 | 93.0 | 82.1 | 99.7 | 95.1 | 78.3 | 62.1 | 86.7 | 96.3 | 91.9 | **76.8** | 94.3 | 66.0 | 62.4 | 71.3 | 89.0 | 60.9 | 34.3 | 49.5 | 77.6 |
| EVA | 85.6 | **93.9** | 82.2 | 99.7 | 95.9 | 93.2 | 63.6 | 86.8 | 96.6 | 92.3 | 76.1 | 96.1 | 65.1 | 61.1 | 83.3 | 91.4 | 61.6 | 35.0 | 55.0 | **79.7** |
| DoRA | 85.9 | 92.7 | 82.1 | 99.7 | 95.2 | 34.4 | 61.4 | 88.6 | 96.8 | 92.4 | 76.8 | 97.6 | 65.4 | 62.7 | 84.4 | 43.2 | 63.1 | 37.8 | 52.6 | 74.4 |
| EVA+DoRA | **86.2** | 92.1 | 81.9 | 99.7 | 94.9 | 93.8 | 62.4 | 88.3 | 96.6 | 92.6 | 76.7 | 97.2 | 65.5 | 54.1 | 83.7 | 93.3 | 62.3 | 37.5 | 54.5 | 79.6 |

performs best on the structured task. However, both LoRA and FFT perform worse on the remaining tasks, leading to a worse average score compared to EVA and EVA+DoRA.

## 4.5 DECISION MAKING

We follow the single task fine-tuning experiments in Schmied et al. (2024) and fine-tune a Decision Transformer (Chen et al., 2021a, DT) on the Meta-World benchmark suite (Yu et al., 2020). Meta-World consists of a diverse set of 50 tasks for robotic manipulation, such as object manipulation, grasping, or pushing buttons. We split Meta-World according to Wolczyk et al. (2021) into 40 pre-training tasks (MT40) and 10 fine-tuning tasks (CW10). We pre-train a 12 M parameter DT on MT40 and fine-tune it on the CW10 holdout tasks. We report success rates and standard errors for each task of CW10 in Table 4. We observe that EVA significantly reduces that gap between LoRA and FFT. Furthermore, DoRA performs particularly well in this experiment and exceeds FFT performance. Finally, our EVA+DoRA even improves upon DoRA and attains the best average performance across all tasks. We report results for different rank budgets in Table 22, as well as implementation details and hyperparameters in Appendix E.

Table 4: Results for single task fine-tuning experiments on the Meta-World benchmark. We report mean success rates and standard error across three seeds for every task.

| | faucet-close | hammer | handle-press | peg-unplug | push-back | push | push-wall | shelf-place | stick-pull | window-close | Average |
|---|---|---|---|---|---|---|---|---|---|---|---|
| FFT | $1.0_{\pm.0}$ | $\underline{0.97_{\pm.03}}$ | $1.0_{\pm.0}$ | $\underline{0.77_{\pm.05}}$ | $\underline{0.87_{\pm.05}}$ | $\mathbf{1.0_{\pm.0}}$ | $\mathbf{1.0_{\pm.0}}$ | $1.0_{\pm.0}$ | $\underline{0.63_{\pm.03}}$ | $1.0_{\pm.0}$ | $0.92$ |
| LoRA | $1.0_{\pm.0}$ | $\mathbf{1.0_{\pm.0}}$ | $1.0_{\pm.0}$ | $0.6_{\pm.05}$ | $0.63_{\pm.1}$ | $\mathbf{1.0_{\pm.0}}$ | $\mathbf{1.0_{\pm.0}}$ | $1.0_{\pm.0}$ | $0.4_{\pm.09}$ | $1.0_{\pm.0}$ | $0.86$ |
| AdaLoRA | $1.0_{\pm.0}$ | $\underline{0.97_{\pm.03}}$ | $1.0_{\pm.0}$ | $0.4_{\pm.09}$ | $0.57_{\pm.1}$ | $\underline{0.97_{\pm.03}}$ | $\underline{0.97_{\pm.03}}$ | $1.0_{\pm.0}$ | $0.13_{\pm.07}$ | $1.0_{\pm.0}$ | $0.80$ |
| PiSSA | $1.0_{\pm.0}$ | $\mathbf{1.0_{\pm.0}}$ | $1.0_{\pm.0}$ | $0.43_{\pm0.11}$ | $0.57_{\pm0.03}$ | $\mathbf{1.0_{\pm.0}}$ | $\mathbf{1.0_{\pm.0}}$ | $1.0_{\pm.0}$ | $0.53_{\pm0.1}$ | $1.0_{\pm.0}$ | $0.85$ |
| OLoRA | $1.0_{\pm.0}$ | $\underline{0.97_{\pm0.03}}$ | $1.0_{\pm.0}$ | $0.57_{\pm0.1}$ | $0.63_{\pm0.03}$ | $\mathbf{1.0_{\pm.0}}$ | $\mathbf{1.0_{\pm.0}}$ | $1.0_{\pm.0}$ | $0.6_{\pm0.12}$ | $1.0_{\pm.0}$ | $0.88$ |
| EVA | $1.0_{\pm.0}$ | $\underline{0.97_{\pm.03}}$ | $1.0_{\pm.0}$ | $0.63_{\pm.03}$ | $0.77_{\pm.05}$ | $\mathbf{1.0_{\pm.0}}$ | $\mathbf{1.0_{\pm.0}}$ | $1.0_{\pm.0}$ | $\underline{0.63_{\pm.07}}$ | $1.0_{\pm.0}$ | $0.90$ |
| DoRA | $1.0_{\pm.0}$ | $\mathbf{1.0_{\pm.0}}$ | $1.0_{\pm.0}$ | $0.6_{\pm1.2}$ | $\mathbf{1.0_{\pm.0}}$ | $\mathbf{1.0_{\pm.0}}$ | $\mathbf{1.0_{\pm.0}}$ | $1.0_{\pm.0}$ | $\mathbf{0.67_{\pm1.5}}$ | $1.0_{\pm.0}$ | $\underline{0.93}$ |
| EVA+DoRA | $1.0_{\pm.0}$ | $\mathbf{1.0_{\pm.0}}$ | $1.0_{\pm.0}$ | $\mathbf{0.8_{\pm.08}}$ | $\mathbf{1.0_{\pm.0}}$ | $\mathbf{1.0_{\pm.0}}$ | $\mathbf{1.0_{\pm.0}}$ | $1.0_{\pm.0}$ | $\underline{0.63_{\pm.03}}$ | $1.0_{\pm.0}$ | $\mathbf{0.94}$ |

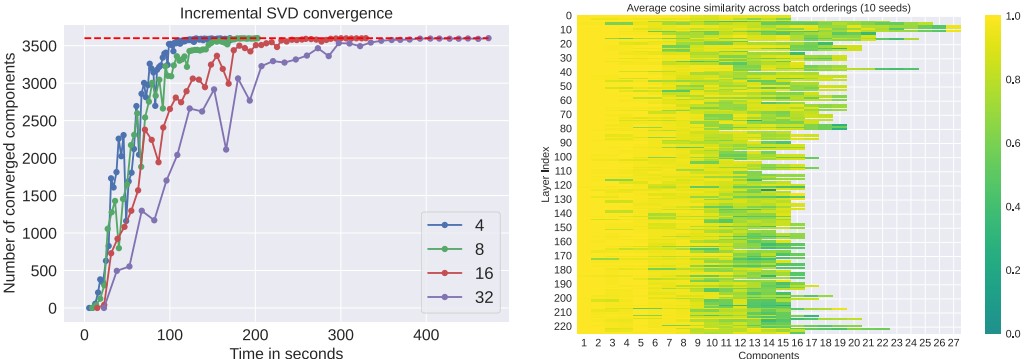

Figure 5: **Left:** Time in seconds until convergence of incremental SVD components for different batch sizes for Llama-2-7B on the MetaMathQA dataset. The dashed line indicates the total number of components. **Right:** Average cosine similarity between SVD components across 10 random seeds for permuting the batch order. The first 10 components remain mostly consistent across all permutations. While the remaining components vary, they strongly correlate with each other.

### 4.6 SVD CONVERGENCE ANALYSIS

The data-driven initialization of EVA relies on incremental SVD on minibatches of activations in the initial training stage. In Figure 5, left, we show that this process converges for Llama-2-7B on MetaMathQA for different minibatch sizes. Using a minibatch size of 4 the computation for EVA's initialization lasts for approximately 80 seconds, which corresponds to around 90 minibatches. For a batch size of 32 the computation of the SVD components takes around 500 seconds. In Figure 5, right, we additionally show, that the main components obtained via SVD mostly remain consistent across different batch orders for a batch size of 4, again for Llama-2-7B on MetaMathQA. To this end, we plot cosine similarity between components obtained via incremental SVD after rank redistribution. These results indicate that these models exhibit certain activation patterns that remain consistent across different batch orders which lead to a robust initialization for EVA. We also show that the components for different batch sizes converge to mostly the same final initialization in Appendix F.

## 5 DISCUSSION AND LIMITATIONS

**Alternative data-driven initialization schemes.** We also investigated alternative data driven initialization schemes. Such alternatives include, but are not limited to, Kernel-PCA (Schölkopf et al., 1997) or Linear Discriminant Analysis (Fisher, 1936, LDA). While Kernel-PCA can account for

non-linearities in the data, it scales with the number of datapoints, which is impractical in our setting. Further, we observed convergence instabilities for incrementally updating LDA.

**Additional latency of SVD.** EVA leads to performance improvements over LoRA, but introduces additional latency in the beginning of training for computing the data-driven initialization. In Table 23 we demonstrate that this process constitutes merely 0.2% of the actual training time for Llama-2-7B on MetaMathQA. Further, in Appendix F we also show that this process is mostly invariant to the batch size, meaning that smaller batch sizes may be used for the SVD computation, resulting in additional speedup. Since, the SVD computation does not require backpropagation and storing of optimizer states there is no overhead with respect to memory.

**Effect of rank redistribution.** Our experiments on language understanding tasks indicate that the effect of rank redistribution strongly depends on the downstream task, i.e. all models benefit from the redistribution on the common sense reasoning tasks, whereas for the math tasks a uniform rank distribution appears to perform best. In our experiments on language understanding and image classification, adaptive ranks performed best, while on decision making uniform ranks performed best. Generally the performance gap between the two is not big and since rank redistribution also leads to fewer trainable parameters we recommend to use it by default.

**What method performs well on which tasks?** We conducted fine-tuning experiments across 51 tasks and four domains and found that EVA or EVA+DoRA performs best on expectation. This is evidenced by the higher average score across multiple tasks per domain. Despite this finding, there is usually variation in the ranking of methods considering single tasks, i.e. LoRA performed better on specialized, and FFT performed best on structured images. Therefore there is no one algorithm that performs best on every single task, verifying that there is no free lunch (Wolpert & Macready, 1997).

**Reproducibility.** We provide the source code along with the submission (see Appendix A) to ensure reproducibility. Further, to make EVA more accessible to the community, we will integrate it into the widely used PEFT library (Mangrulkar et al., 2022).

## 6    CONCLUSION AND BROADER IMPACT

We propose a novel method named Explained Variance Adaptation (EVA), extending the widely used LoRA with data-driven initialization and rank re-distribution. We initialize LoRA matrices in a data-driven manner by performing SVD on minibatches of activation vectors. Further, we re-distribute ranks across weight matrices according to the amount of variance they explain. In this regard, we also introduce a hyperparameter that allows for a controlled investigation of different rank distributions. Thereby, in EVA we bind the benefits of adaptive rank allocation and data-driven initialization, resulting in one initialization to rule them all. We demonstrate performance gains of EVA over LoRA and initialization schemes thereof on a variety of domains, ranging from language to vision and RL. Our results demonstrate that EVA variants consistently reach the highest average performance on a wide range of tasks across all domains.

We believe that EVA sheds a novel view on LoRA fine-tuning, where initialization of the newly introduced weights is guided by the downstream data. As we have shown, this can boost performance on a wide variety of domains. We believe that EVA can have a significant impact on future research on fine-tuning of foundation models, because it inherits all benefits of LoRA while improving performance at no significant additional cost. In the future, we aim at investigating the effect of rank redistribution on other initialization schemes and quantization, as well as alternative data-driven initialization schemes in more detail.

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

# SUPPLEMENTARY MATERIAL

**Anonymous authors**

## CONTENTS

## A    REPRODUCIBILITY STATEMENT

We provide the source code to reproduce all our experiments in the supplementary material as a zip archive. The archive contains two sub-directories named NLU and NLG, which can be used to reproduce the results on language understanding and generation. For image classification and decision making experiments we used custom implementations which we will open-source as well. Both code directories contain instructions how to install the environment and how to execute all the parameter searches and obtain our results. Additionally, we provide a package that contains implementations for EVA along with different LoRA variants, such as DoRA, and ELoRA in the NLU code directory. We will release a unified codebase upon publication and also integrate EVA into the widely used PEFT library (Mangrulkar et al., 2022).

## B    NATURAL LANGUAGE GENERATION

We follow the experiments conducted in Hu et al. (2023) and fine-tune Llama-2-7B, Llama-3.1-8B and Gemma-2-9B on 8 common sense reasoning tasks with qa style prompts. We keep the original prompt templates unchanged aside from two minor modifications: For BoolQ we prepend the the passage field before the question and for WinoGrande we add a line "Answer format: ..." analogous to the other prompts. As done by Hu et al. (2023) as well as Liu et al. (2024a) we perform joint finetuning on all 8 tasks. We furthermore evaluate the pre-trained models mentioned above on the mathematical reasoning tasks GSM8K (Cobbe et al., 2021) and Math (Yu et al., 2024) after finetuning on MetaMathQA (Yu et al., 2024) as done in Meng et al. (2024). We keep the original prompt template for finetuning and evaluation. For all datasets we run finetuning for one epoch.

### B.1    IMPLEMENTATION DETAILS

For finetuning our code base leverages peft implementations of adapter methods LoRA, AdaLoRA, PiSSA, OLoRA and DoRA. The initialization step for EVA is a custom implementation but for finetuning we can reformulate EVA as a LoRA adapter leveraging the rank_pattern argument of peft.LoraConfig. For evaluation we leverage scripts provided by the MetaMath github repository (Yu et al., 2024) for math reasoning tasks. For common sense reasoning we make use of the lm evaluation harness project (Gao et al., 2024) and define custom tasks using the finetuning prompts. For the SVD computation for joint finetuning on the common sense reasoning tasks we experiment with random and stratified sampling of examples from the 8 tasks and do not notice a difference in performance. All training and evaluation runs for Llama-2-7B were done on 4 A100 GPUs. Runs for Llama-3.1-8B and Gemma-2-9B utilized two different nodes, one with 4 A100 GPUs and one with 4 H200 GPUs.

Table 6: hyperparameters for finetuning on common sense reasoning and math reasoning

| Training | |
| --- | --- |
| Optimizer | AdamW |
| Weight Decay | 0.0 |
| Lora Dropout | 0.0 |
| Batch Size | 32 |
| #Epoch | 1 |
| LR Schedule | Linear |
| Warmup ratio | 0.03 |
| Label Smooth | 0.0 |
| Learning Rate | 5e-4 |
| LoRA Dim | 16 |
| LoRA $\alpha$ | 1 |
| Batch Size SVD (EVA) | 16 |
| $\tau$ | 0.99 |
| Inference | |
| Beam Size | 1.0 |
| Length Penalty | 1.0 |
| repetition penalty | 1.0 |

### B.2    HYPERPARAMETER SEARCH

The reported results on language generation tasks in Table 7 and Table 8 are the best setting based on a grid search over different learning rates. We apply adapters to all linear layers including the language modelling head. Furthermore we set $\alpha = 1$ for all our experiments. We use AdamW with weight decay and a linear learning rate schedule with warm-up. We train for 1 epoch and use the final checkpoint for evaluation. All hyperparameters are summarized in Table 6

Table 5: Prompt templates with examples (red) used for finetuning on common sense and math reasoning tasks.

| Dataset | Fine-tuning Data Template |
|---|---|
| BoolQ | Passage: Drinking in public – Drinking in public is most commonly accepted. After reading this passage, please answer the following question with true or false, question: can you drink on the street in china
Answer format: true/false
the correct answer is true |
| PIQA | Please choose the correct solution to the question: When boiling butter, when it's ready, you can
Solution1: Pour it onto a plate
Solution2: Pour it into a jar
Answer format: solution 1/solution2
the correct answer is solution2 |
| SIQA | Please choose the correct answer to the question: Carson relocated somewhere new. How would you describe Carson?
Answer1: mobile
Answer2: anxious
Answer3: lonely
Answer format: answer1/answer2/answer3
the correct answer is answer1 |
| HellaSwag | Please choose the correct ending to complete the given sentence: Playing drums: People are standing behind large drums. A man
Ending1: is playing a bag pipe.
Ending2: starts to play around the drums.
Ending3: begins playing a drum set.
Ending4: begins playing the drums.
Answer format: ending1/ending2/ending3/ending4
the correct answer is ending4 |
| WinoGrande | Please choose the correct answer to fill in the blank to complete the given sentence: Ian volunteered to eat Dennis's menudo after already having a bowl because _ despised eating intestine.
Option1: Ian
Option2: Dennis
Answer format: option1/option2
the correct answer is option2 |
| ARC-e & ARC-c | Please choose the correct answer to the question: Which factor will most likely cause a person to develop a fever?
Answer1: a leg muscle relaxing after exercise
Answer2: a bacterial population in the bloodstream
Answer3: several viral particles on the skin
Answer4: carbohydrates being digested in the stomach
Answer format: answer1/answer2/answer3/answer4
the correct answer is answer2 |
| OBQA | Please choose the correct answer to the question: The sun is responsible for
Answer1: puppies learning new tricks
Answer2: children growing up and getting old
Answer3: flowers wilting in a vase
Answer4: plants sprouting, blooming and wilting
Answer format: answer1/answer2/answer3/answer4
the correct answer is answer4 |
| MetaMathQA | Below is an instruction that describes a task. Write a response that appropriately completes the request.

### Instruction:
What is the value of the cosine of 90 degrees?

### Response:
s $\\boxed{0}$.The answer is: 0 |

Table 7: Comparison of LoRA and DoRA to different initialization and rank re-distribution methods on NLG tasks. We report average performance across three seeds and respective standard deviation in Table 14. EVA+DoRA and EVA consistently attain the highest average performance across all tasks.

| Model | Method | BoolQ | PIQA | SIQA | HellaSwag | Winogrande | ARC-e | ARC-c | OBQA | Avg. |
|---|---|---|---|---|---|---|---|---|---|---|
| Llama-2-7B | LoRA | 67.2 | 83.9 | 82.0 | 94.7 | 84.0 | 87.8 | 74.1 | 84.0 | 82.2 |
| | AdaLoRA | 74.8 | 82.2 | 80.5 | 93.3 | 79.4 | 86.1 | 71.1 | 80.6 | 81.0 |
| | PiSSA | 62.6 | 84.8 | 81.2 | 94.5 | 84.8 | 87.8 | 74.8 | 85.4 | 82.0 |
| | OLoRA | 68.7 | 84.8 | 82.2 | 95.0 | 85.0 | 88.1 | 74.9 | 85.2 | 82.9 |
| | LoRA-GA | 69.0 | 85.6 | 82.3 | 95.0 | 85.0 | 88.7 | 75.9 | 85.8 | 83.4 |
| | EVA | 68.3 | 85.3 | 82.9 | 95.2 | 85.2 | 88.6 | 75.8 | 86.3 | 83.4 |
| | DoRA | 68.3 | 85.1 | 82.2 | 94.9 | 84.3 | 88.7 | 74.8 | 86.3 | 83.1 |
| | EVA+DoRA | 73.5 | 85.3 | 82.4 | 95.2 | 84.8 | 88.9 | 76.0 | 87.3 | 84.2 |
| Llama-3.1-8B | LoRA | 85.7 | 90.3 | 83.0 | 96.9 | 88.4 | 94.2 | 84.8 | 90.1 | 89.2 |
| | AdaLoRA | 83.9 | 89.5 | 81.7 | 96.2 | 86.3 | 93.7 | 82.7 | 86.8 | 87.6 |
| | PiSSA | 72.9 | 87.3 | 81.6 | 95.3 | 87.8 | 91.7 | 81.2 | 87.6 | 85.7 |
| | OLoRA | 86.0 | 90.4 | 83.9 | 97.0 | 88.6 | 94.5 | 84.7 | 90.3 | 89.4 |
| | LoRA-GA | 83.7 | 89.7 | 83.1 | 96.7 | 88.8 | 94.2 | 85.3 | 90.4 | 89.0 |
| | EVA | 85.3 | 90.4 | 83.4 | 97.0 | 89.0 | 94.4 | 86.0 | 90.3 | 89.5 |
| | DoRA | 86.2 | 90.8 | 83.4 | 96.9 | 88.6 | 94.3 | 84.9 | 89.4 | 89.3 |
| | EVA+DoRA | 85.8 | 90.8 | 83.9 | 97.1 | 89.2 | 94.4 | 85.9 | 90.5 | 89.7 |
| Gemma-2-9B | LoRA | 88.3 | 92.9 | 85.2 | 97.8 | 92.3 | 97.2 | 89.9 | 94.4 | 92.2 |
| | AdaLoRA | 87.3 | 91.8 | 84.6 | 97.3 | 91.3 | 97.0 | 90.0 | 92.6 | 91.5 |
| | PiSSA | 81.4 | 90.0 | 82.5 | 95.5 | 89.0 | 93.6 | 83.5 | 90.8 | 88.3 |
| | OLoRA | 87.7 | 92.5 | 85.2 | 97.5 | 92.5 | 96.6 | 88.7 | 93.7 | 91.8 |
| | LoRA-GA | 87.3 | 92.1 | 84.5 | 97.4 | 93.2 | 96.4 | 89.2 | 94.3 | 91.8 |
| | EVA | 88.6 | 93.0 | 85.3 | 97.9 | 92.8 | 97.5 | 90.5 | 94.5 | 92.5 |
| | DoRA | 88.3 | 92.6 | 84.9 | 97.7 | 92.2 | 97.1 | 89.9 | 94.5 | 92.1 |
| | EVA+DoRA | 88.6 | 93.1 | 85.1 | 97.9 | 92.5 | 97.3 | 89.6 | 94.8 | 92.4 |

### B.3 ADDITIONAL RESULTS

First, we present the per-task performance for the eight common sense reasoning tasks in Table 7. The respective standard deviations are shown in Table 14. Further, we show the results for all methods on the two math reasoning datasets in Table 8.

We present additional loss curves for Llama-2-7B, Llama-3.1-8B, and Gemma-2-9B on the common sense and math reasoning tasks in Figure 6. We find that EVA converges the fastest for all the different models on the different tasks.

Another experiment we conduct is to apply recently proposed changes to the scaling factor and learning rate. In Table 9 we show results for changing the scaling factor to $\alpha = \frac{2r}{\sqrt{r}}$ which results in rank stabilization (Kalajdzievski, 2023). Further, we present results for the regular setting $\alpha = 2r$ as proposed in Hu et al. (2022). Finally, we also show different learning rates for the two matrices $A$ and $B$ as proposed by Hayou et al. (2024). We make the following observations:

1. The standard setting $\alpha = 2r$ from Hu et al. (2022) leads to the worst performance

2. Rank stabilization via $\alpha = \frac{2r}{\sqrt{r}}$ significantly improves the performance of both LoRA and EVA

3. Different learning rates for $A$ and $B$ did not improve the results

To provide a comprehensive comparison about the effect of rank re-distribution, we compare uniform ranks ($\rho = 1$) to adaptive ranks ($\rho = 2$) on the common sense and math reasoning tasks in Table 10. We find that adaptive ranks consistently improves performance for Gemma-2-9B. For Llama-2-7B and Llama-3.1-8B we observe improvements on the common sense reasoning tasks only, while uniform ranks perform better on the math fine-tuning tasks.

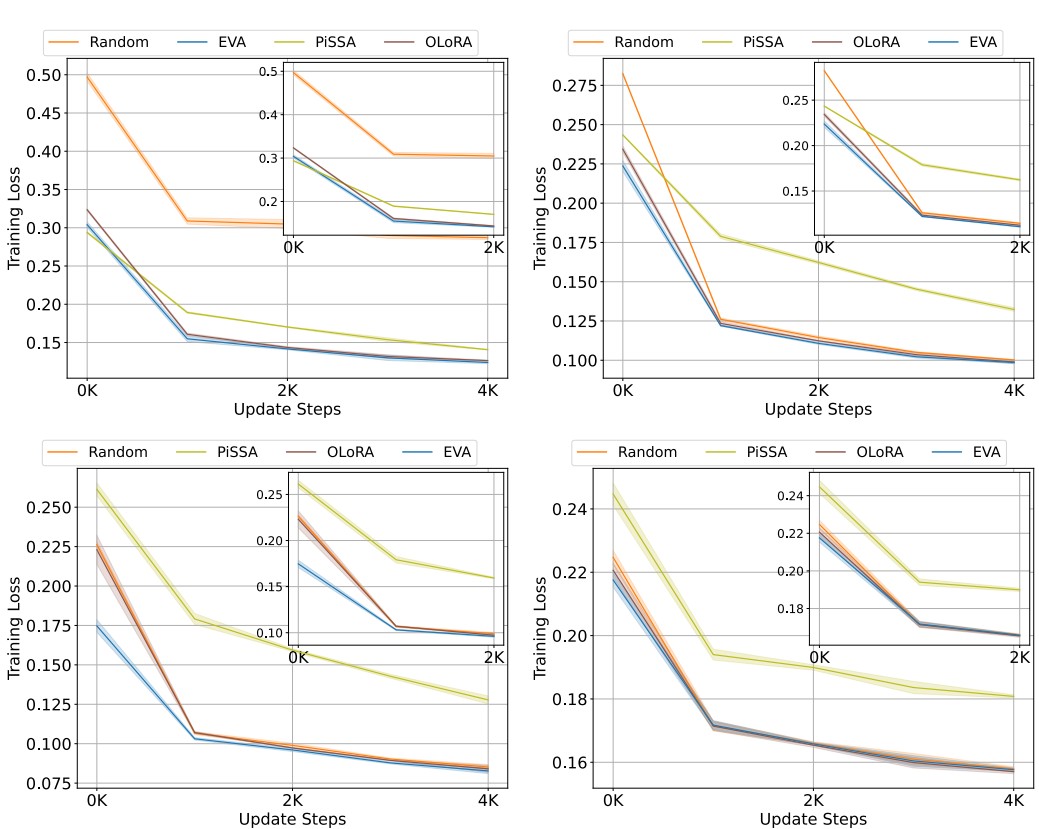

Figure 6: Loss curves for Llama-2-7B on common sense reasoning (top left), Llama-3.1-8B on common sense reasoning (top right), Gemma-2-9B on common sense reasoning (bottom right), and Gemma-2-9B on MetaMathQA. EVA consistently converges the fastest among all competitors.

Table 8: Comparison of EVA to other initialization and adaptive rank methods on GSM8K and MATH datasets. We report mean and standard deviation across three random seeds.

| Model | Method | GSM8K | MATH |
|---|---|---|---|
| Llama-2-7B | LoRA | $59.7_{\pm.8}$ | $10.9_{\pm.2}$ |
| | AdaLoRA | $56.9_{\pm.4}$ | $9.6_{\pm.2}$ |
| | PiSSA | $61.1_{\pm.3}$ | $12.6_{\pm.4}$ |
| | OLoRA | $60.7_{\pm.5}$ | $11.8_{\pm.3}$ |
| | LoRA-GA | $60.2_{\pm.6}$ | $11.7_{\pm.4}$ |
| | EVA | $61.9_{\pm.5}$ | $13.1_{\pm.3}$ |
| | DoRA | $59.8_{\pm.5}$ | $11.5_{\pm.2}$ |
| | EVA+DoRA | $\mathbf{62.5}_{\pm.8}$ | $\mathbf{13.4}_{\pm.01}$ |
| Llama-3.1-8B | LoRA | $78.3_{\pm.6}$ | $30.1_{\pm.5}$ |
| | AdaLoRA | $76.9_{\pm.2}$ | $28.9_{\pm.7}$ |
| | PiSSA | $78.8_{\pm.2}$ | $29.5_{\pm.5}$ |
| | OLoRA | $78.0_{\pm.1}$ | $31.0_{\pm.7}$ |
| | LoRA-GA | $78.8_{\pm.1}$ | $30.0_{\pm.1}$ |
| | EVA | $78.8_{\pm.3}$ | $\mathbf{31.2}_{\pm.3}$ |
| | DoRA | $77.9_{\pm.1}$ | $30.2_{\pm.5}$ |
| | EVA+DoRA | $\mathbf{79.1}_{\pm.5}$ | $30.8_{\pm.4}$ |
| Gemma-2-9B | LoRA | $83.4_{\pm.9}$ | $40.7_{\pm.2}$ |
| | AdaLoRA | $\underline{83.5}_{\pm.5}$ | $41.1_{\pm.4}$ |
| | PiSSA | $79.8_{\pm.5}$ | $34.9_{\pm.2}$ |
| | OLoRA | $82.2_{\pm.2}$ | $39.4_{\pm.6}$ |
| | LoRA-GA | $82.8_{\pm.9}$ | $40.4_{\pm.4}$ |
| | EVA | $\mathbf{83.6}_{\pm.8}$ | $\mathbf{41.5}_{\pm.3}$ |
| | DoRA | $82.5_{\pm.6}$ | $39.7_{\pm.4}$ |
| | EVA+DoRA | $82.9_{\pm.3}$ | $40.0_{\pm.6}$ |

In Table 11 we show the number of trainable parameters for EVA ($\rho = 2$) compared to LoRA on the common sense and math reasoning tasks. We find that after rank redistribution, EVA leads to improved performance while reducing the parameter count by approximately 1M. The reason for this is that parameters are usually re-distributed from higher dimensional projections to lower dimensional ones, i.e. from non-attention weights to attention weights. This results in improved performance while reducing the parameter count.

Finally, to verify our intuition that the LoRA matrix $\boldsymbol{A}$ should be initialized with the projection onto the components that explain the most variance, we compare its performance to initializing EVA with the components that explain the *least* amount of variance. We call this method EVA-minor and present results for it in Table 12. To implement EVA-minor, we sample 20 minibatches of data and perform truncated SVD on those and select the resulting minor components. This incurs substantial additional cost, as we must compute all components, whereas for EVA we only approximate the components that explain the most variance. Hence, incremental SVD is not beneficial in this case anymore and it is also not practical as obtaining the initialization takes hours instead of seconds for EVA. Moreover, our data-driven heuristic for adaptive rank allocation is not applicable to this case anymore, therefore we consider uniform ranks. Finally, we find that EVA consistently improves over EVA-minor, highlighting the importance of initializing EVA with the major components, i.e. the ones the explain the most variance.

In addition we also fine-tune Llama-2-7B on the Code-Feedback dataset Zheng et al. (2024) consisting of multi-turn conversations between user and AI Assistant. Due to limited computational resources and the long sequence lengths of the examples in this dataset we do not fine-tune Llama-3.1-8B and Gemma-2-9B or any DoRA variants. We evaluate the fine-tuned checkpoints on four coding benchmarks: MBPP Austin et al. (2021), HumanEval Chen et al. (2021b), MBPP+ and HumanEval+ Liu et al. (2023). The results are presented in Table 13. EVA shows the best performance on MBPP and MBPP+ while also exhibiting good performance on HumanEval and HumanEval+. On the latter two datasets, PiSSA is the best performing method. For finetuning we use a maximum sequence length of 2028 with right-side truncation. For decoding we set the temperature to 0.2 and top_p to 0.7

Table 9: Comparison of EVA to LoRA using recently proposed advancements, such as rank stabilized scaling (Kalajdzievski, 2023) or different learning rates for $B$ and $A$ (Hayou et al., 2024), as well as the originally proposed scaling from Hu et al. (2022).

| Adaptation | Method | BoolQ | PIQA | SIQA | HellaSwag | Winogrande | ARC-e | ARC-c | OBQA | Avg. |
|---|---|---|---|---|---|---|---|---|---|---|
| LoRA+ | LoRA | 64.5 | 84.7 | 81.6 | 94.4 | 83.8 | 87.3 | 73.9 | 85.5 | 82.0 |
| | EVA | 68.6 | 85.0 | 81.2 | 94.2 | 84.7 | 87.4 | 73.5 | 84.1 | 82.3 |
| rsLoRA | LoRA | 71.5 | 85.3 | 82.5 | 95.2 | 84.5 | 89.0 | 75.8 | 86.8 | 83.8 |
| | EVA | 75.5 | 86.1 | 82.7 | 95.4 | 86.1 | 89.3 | 76.3 | 86.3 | 84.7 |
| $\alpha = 32$ | LoRA | 77.9 | 82.1 | 80.1 | 93.2 | 79.8 | 86.3 | 71.5 | 79.3 | 81.3 |
| | EVA | 68.6 | 84.9 | 82.2 | 94.6 | 84.1 | 87.8 | 74.7 | 84.4 | 82.7 |

Table 10: Comparison of EVA with rank redistribution ($\rho = 2$) and without rank redistribution ($\rho = 1$) for Llama-2-7B, Llama-3.1-8B, and Gemma-2-9B on common sense reasoning and math fine-tuning. Rank re-distribution works well for Gemma-2-9B and for Llama-2-7B and Llama-3.1-8B on the common sense reasoning tasks.

| Model | $\rho$ | Common sense | GSM8K | MATH |
|---|---|---|---|---|
| Llama-2-7B | 1 | 83.4 | 61.9 | 13.1 |
| | 2 | 83.4 | 61.0 | 12.5 |
| Llama-3.1-8B | 1 | 89.4 | 78.8 | 31.2 |
| | 2 | 89.5 | 78.3 | 30.8 |
| Gemma-2-9B | 1 | 92.4 | 83.6 | 41.3 |
| | 2 | 92.5 | 83.6 | 41.5 |

In Table 14 we report the standard deviation across three seeds from the results in Table 7. For Llama-3.1-8B and Gemma-2-9B EVA has the smallest average standard deviation across tasks. For Llama-2-7B the standard the variance of EVA is only slightly above average in comparison to other methods, mainly due to the high standard deviation on the BoolQ dataset.

## C   NATURAL LANGUAGE UNDERSTANDING

### C.1   DATASET STATISTICS

The dataset statistics for each task in the GLUE benchmark (Wang et al., 2019) are shown in Table 15. Generally, GLUE contains four low-resource datasets (RTE, MRPC, STS-B, and CoLA) and four high resource datasets (SST-2, QNLI, QQP, MNLI). While CoLA and SST-2 rely on single sentence classification, STS-B evaluates for similarity and the remaining tasks are based on pairwise text classification.

### C.2   IMPLEMENTATION DETAILS

We base our implementation on the codebase of LoRA[1]. For these experiments, we initially pre-compute our initialization prior to the fine-tuning stage and store it as a checkpoint. However, we also provide the possibility to directly compute the initialization during the fine-tuning stage, as done for our experiments on VTAB-1k and Meta-World. By default, we always offload the computation of the initial checkpoint to CPU to save VRAM. We ran all our experiments on nodes with four A100 GPUs and used PyTorch's data-distributed parallel functionality (Paszke et al., 2019). Runtimes ranges from as little as 10 minutes per run for smaller datasets (RTE, STS-B) to around 15 hours for the largest datasets (QQP, MNLI).

---

[1] https://github.com/microsoft/LoRA

Table 11: Comparison of number of trainable parameters between LoRA-based methods and EVA on the math and common sense reasoning tasks. Common sense reasoning is an average over eight tasks. #Trainable represents the number of trainable parameters. EVA consistently improves performance while decreasing the number of trainable parameters.

| Model | Method | #Trainable | Common sense | GSM8K | MATH |
|-------|--------|-----------|--------------|-------|------|
| Llama-2-7B | LoRA | 18.3M | 82.2 | 59.7 | 10.9 |
| | EVA | 17.3M | 83.4 | 61.9 | 13.1 |
| | EVA−− | | | | |
| Llama-3.1-8B | LoRA | 20M | 89.2 | 78.3 | 30.1 |
| | EVA | 18.9M | 89.5 | 78.8 | 31.2 |
| | EVA−− | | | | |
| Gemma-2-9B | LoRA | 24.5M | 92.2 | 83.4 | 40.7 |
| | EVA | 23.1M | 92.5 | 83.6 | 41.5 |
| | EVA−− | | | | |

Table 12: Comparison of EVA to EVA-minor, which leverages components that explain the *least* amount of variance for initialization of $A$, on the common sense reasoning tasks.

| Method | BoolQ | PIQA | SIQA | HellaSwag | Winogrande | ARC-e | ARC-c | OBQA | Avg. |
|--------|-------|------|------|-----------|------------|-------|-------|------|------|
| EVA | 68.6 | 85.0 | 81.2 | 94.2 | 84.7 | 87.4 | 73.5 | 84.1 | 82.3 |
| EVA-minor | 64.0 | 83.4 | 81.5 | 94.3 | 82.0 | 87.3 | 73.0 | 81.6 | 80.9 |

### C.3 HYPERPARAMETER SEARCH

For LoRA and EVA, we search over the number of ranks $r \in \{2, 4, 6, 8\}$ and different learning rates $\eta \in \{1e-3, 4e-4, 1e-4\}$ for RoBERTa$_{Large}$ and $\eta \in \{4e-3, 1e-3, 4e-4\}$ for DeBERTav3$_{Base}$. We report the best hyperparameter settings for both, RoBERTa$_{Large}$ and DeBERTav3$_{Base}$ for LoRA and EVA in Table 16. For AdaLoRA, we search over the same ranks and always start initial ranks with $r+4$ that are then redistributed during training. For BOFT we sweep over different combinations of block sizes $b \in \{2, 4, 8, 16\}$ which determine the number of multiplicative matrices. Additionally, for both, AdaLoRA and BOFT, we search over the same learning rates as for the other LoRA variants. Further, we introduce hyperparameters that result in additional speed-up of our initialization, namely a threshold $\tau$ that considers components as converged, and a threshold $\delta$ that stops computation of the initialization when a certain percentage of components have converged. By default, we set $\tau = 0.99$ and $\delta = 1$, i.e. we only stop when all components are converged, and they are almost exactly the same. These parameters provide additional leeway to speed up the initialization stage of EVA.

We have explored the sensitivity of LoRA to different initialization schemes and found that, similar to other prominent initialization schemes (He et al., 2015; Glorot & Bengio, 2010), scale plays an important role along with directions. Originally, (Hu et al., 2022) propose to set $\alpha = 2r$, however, we found that this parameter is quite sensitive as also shown in (Kalajdzievski, 2023). Similarly, different ranks lead to very different results on different downstream tasks. Therefore, we suggest to always search over more ranks and choose the best performing one if the required compute budget is available. We also experimented with different learning rates for the $A$ and $B$ matrices as proposed in (Hayou et al., 2024), however, this did not result in consistent improvements. Instead, we found that learning rates for LoRA-style training can be surprisingly high ($4e-3$ for DeBERTav3$_{Base}$), while for larger models the learning rate needs to be approximately a magnitude smaller. A simple recipe that worked consistently well, was setting $\alpha = 1$, which results in a similar scaling factor as in Kalajdzievski (2023), and searching over a set of small learning rates for larger models and higher learning rates for smaller ones. For EVA, the only tunable hyperparameter is the rank budget, which we recommend to tune along with the fine-tuning learning rate.

Table 13: Comparison of EVA to other initialization and rank re-distribution schemes on code fine-tuning datasets. We report mean and standard deviation across three random seeds.

| Method | MBPP | HumanEval | MBPP+ | HumanEval+ |
|--------|------|-----------|-------|------------|
| LoRA | $22.2_{\pm 1.1}$ | $\underline{18.9}_{\pm 0.6}$ | $30.7_{\pm 1.1}$ | $\underline{18.9}_{\pm 0.6}$ |
| AdaLoRA | $21.5_{\pm 0.2}$ | $17.1_{\pm 0.0}$ | $29.4_{\pm 0.7}$ | $17.1_{\pm 0.0}$ |
| PiSSA | $\underline{22.8}_{\pm 1.2}$ | $\mathbf{19.9}_{\pm 0.9}$ | $30.8_{\pm 0.7}$ | $\mathbf{19.9}_{\pm 0.9}$ |
| OLoRA | $22.3_{\pm 0.6}$ | $\underline{18.9}_{\pm 0.0}$ | $\underline{32.4}_{\pm 0.4}$ | $\underline{18.9}_{\pm 0.0}$ |
| EVA | $\mathbf{22.9}_{\pm 0.7}$ | $\underline{18.9}_{\pm 1.2}$ | $\mathbf{32.6}_{\pm 0.6}$ | $\underline{18.9}_{\pm 1.2}$ |

Table 14: Standard deviation across three seeds on common sense reasoning tasks.

| Model | Method | BoolQ | PIQA | SIQA | HellaSwag | Winogrande | ARC-e | ARC-c | OBQA |
|-------|--------|-------|------|------|-----------|------------|-------|-------|------|
| Llama-2-7B | LoRA | 1.498 | 0.252 | 0.233 | 0.102 | 0.658 | 0.072 | 0.489 | 0.822 |
| | AdaLoRA | 1.315 | 0.251 | 0.182 | 0.098 | 0.392 | 0.362 | 0.106 | 0.899 |
| | PiSSA | 0.358 | 0.294 | 0.138 | 0.096 | 0.298 | 0.386 | 0.494 | 1.117 |
| | OLoRA | 4.938 | 0.190 | 0.524 | 0.062 | 0.652 | 0.339 | 0.672 | 0.660 |
| | LoRA-GA | 10.573 | 0.416 | 1.049 | 0.115 | 0.344 | 0.170 | 0.560 | 0.721 |
| | EVA | 7.974 | 0.137 | 1.054 | 0.101 | 0.810 | 0.526 | 0.421 | 0.577 |
| | DoRA | 2.599 | 0.290 | 0.483 | 0.113 | 0.244 | 0.215 | 0.489 | 0.525 |
| | EVA+DoRA | 5.281 | 0.273 | 0.293 | 0.034 | 0.853 | 0.110 | 0.494 | 0.249 |
| Llama-3.1-8B | LoRA | 0.472 | 0.194 | 0.419 | 0.070 | 0.197 | 0.052 | 0.563 | 0.189 |
| | AdaLoRA | 0.510 | 0.044 | 0.261 | 0.040 | 0.392 | 0.201 | 0.804 | 0.748 |
| | PiSSA | 6.516 | 0.373 | 0.603 | 0.195 | 0.707 | 0.325 | 0.245 | 0.589 |
| | OLoRA | 0.298 | 0.245 | 0.397 | 0.057 | 0.451 | 0.173 | 0.329 | 0.189 |
| | LoRA-GA | 0.539 | 0.237 | 0.695 | 0.115 | 0.592 | 0.135 | 0.729 | 0.800 |
| | EVA | 0.353 | 0.031 | 0.194 | 0.046 | 0.209 | 0.292 | 0.178 | 0.808 |
| | DoRA | 0.225 | 0.112 | 0.315 | 0.014 | 0.260 | 0.119 | 0.698 | 0.000 |
| | EVA+DoRA | 0.225 | 0.168 | 0.121 | 0.117 | 0.392 | 0.105 | 0.175 | 0.249 |
| Gemma-2-9B | LoRA | 0.095 | 0.277 | 0.386 | 0.062 | 0.324 | 0.072 | 0.070 | 0.589 |
| | AdaLoRA | 0.088 | 0.353 | 0.217 | 0.033 | 0.098 | 0.209 | 0.106 | 0.432 |
| | PiSSA | 2.761 | 0.286 | 0.214 | 0.109 | 0.621 | 0.447 | 0.121 | 0.163 |
| | OLoRA | 0.066 | 0.451 | 0.501 | 0.099 | 0.501 | 0.267 | 0.448 | 0.573 |
| | LoRA-GA | 0.662 | 0.463 | 0.252 | 0.072 | 0.526 | 0.129 | 0.617 | 1.026 |
| | EVA | 0.275 | 0.136 | 0.111 | 0.094 | 0.260 | 0.119 | 0.040 | 0.249 |
| | DoRA | 0.189 | 0.420 | 0.301 | 0.074 | 0.419 | 0.091 | 0.000 | 0.499 |
| | EVA+DoRA | 0.132 | 0.296 | 0.490 | 0.070 | 0.037 | 0.150 | 0.715 | 0.340 |

## C.4 ADDITIONAL RESULTS

We report additional results for EVA compared to LoRA for different rank budgets in Table 17. We find that EVA consistently outperforms LoRA for different rank budgets. This demonstrates the effectiveness of EVA among different compute budgets. Further, we show additional rank redistributions for the CoLA, MRPC, RTE, and STSB tasks for different for $r = 2$ (Figure 7), $r = 4$ (Figure 8), $r = 8$ (Figure 9), and $r = 16$ (Figure 10) for both, RoBERTa$_{\text{Large}}$ and DeBERTav3$_{\text{Base}}$. The distributions for the different models show different patterns. For DeBERTav3$_{\text{Base}}$ the higher attention layers usually receive more ranks than lower ones. For CoLA, there is also a high number of ranks in the very first layer. For RoBERTa$_{\text{Large}}$ it seems to be the opposite, as the very first layers consistently receive more ranks compared to later layers. There is also a notable difference across tasks for both models, which demonstrates the flexibility of EVA to allocate ranks dependent on the downstream task. Interestingly, for a higher initial rank ($r = 16$), the redistribution for DeBERTav3$_{\text{Base}}$ puts more emphasis on fine-tuning the self-attention specific weight matrices. This is not true for RoBERTa$_{\text{Large}}$, as $W_{f1}$ also receives plenty of ranks across all tasks. Overall, the rank redistribution incurs different fine-tuning paradigms depending on the task and the initial rank.

Table 15: GLUE benchmark suite statistics and evaluation metric for each corpus sorted by the number of examples in the training set.

| Corpus | #Train | #Dev | #Test | Metric |
|---|---|---|---|---|
| RTE | 2.5 k | 276 | 3 k | Accuracy |
| MRPC | 3.7 k | 408 | 1.7 k | Accuracy |
| STS-B | 7 k | 1.5 k | 1.4 k | Pearson correlation |
| CoLA | 8.5 k | 1 k | 1 k | Matthew's correlation |
| SST-2 | 67 k | 872 | 1.8 k | Accuracy |
| QNLI | 108 k | 5.7 k | 5.7 k | Accuracy |
| QQP | 364 k | 40 k | 391 k | Accuracy |
| MNLI | 393 k | 20 k | 20 k | Accuracy |

Table 16: The best hyperparameters RoBERTa$_{Large}$ and DeBERTav3$_{Base}$ that were found via gridsearch for each task of the GLUE benchmark.

| Method | Dataset | MNLI | SST-2 | MRPC | CoLA | QNLI | QQP | RTE | STS-B |
|---|---|---|---|---|---|---|---|---|---|
| | Optimizer | | | | AdamW | | | | |
| | Warmup Ratio | | | | 0.06 | | | | |
| | LR Schedule | | | | Linear | | | | |
| RoBERTa$_{Large}$ LoRA | Batch Size | 8 | 16 | 8 | 8 | 8 | 8 | 16 | 8 |
| | # Epochs | 10 | 10 | 20 | 20 | 10 | 20 | 20 | 10 |
| | LoRA rank | 2 | 8 | 8 | 4 | 8 | 4 | 2 | 2 |
| | Learning rate | 4e-4 | 1e-3 | 4e-4 | 1e-3 | 1e-3 | 1e-3 | 1e-3 | 4e-4 |
| | LoRA $\alpha$ | | | | 1 | | | | |
| | Max Seq. Len. | | | | 512 | | | | |
| | DDP GPUs | | | | 4 | | | | |
| RoBERTa$_{Large}$ EVA | Batch Size | 8 | 16 | 8 | 8 | 8 | 8 | 16 | 8 |
| | # Epochs | 10 | 10 | 20 | 20 | 10 | 20 | 20 | 10 |
| | LoRA rank | 2 | 2 | 4 | 2 | 16 | 8 | 4 | 4 |
| | Learning rate | 4e-4 | 1e-3 | 4e-4 | 1e-3 | 4e-4 | 1e-3 | 1e-3 | 1e-3 |
| | LoRA $\alpha$ | | | | 1 | | | | |
| | Max Seq. Len. | | | | 512 | | | | |
| | DDP GPUs | | | | 4 | | | | |
| DeBERTav3$_{Base}$ LoRA | Batch Size | 32 | 32 | 16 | 32 | 64 | 32 | 32 | 16 |
| | # Epochs | 30 | 60 | 30 | 80 | 25 | 25 | 80 | 40 |
| | LoRA rank | 8 | 4 | 4 | 8 | 16 | 4 | 4 | 8 |
| | Learning rate | 4e-4 | 1e-3 | 4e-3 | 4e-3 | 4e-3 | 4e-3 | 4e-3 | 4e-3 |
| | LoRA $\alpha$ | | | | 1 | | | | |
| | Max Seq. Len. | | | | 512 | | | | |
| | DDP GPUs | | | | 4 | | | | |
| DeBERTav3$_{Base}$ EVA | Batch Size | 32 | 32 | 16 | 32 | 64 | 32 | 32 | 16 |
| | # Epochs | 30 | 60 | 30 | 80 | 25 | 25 | 80 | 40 |
| | LoRA rank | 8 | 2 | 4 | 8 | 16 | 4 | 2 | 2 |
| | Learning rate | 4e-4 | 4e-4 | 4e-3 | 4e-3 | 4e-3 | 4e-3 | 4e-3 | 4e-3 |
| | LoRA $\alpha$ | | | | 1 | | | | |
| | Max Seq. Len. | | | | 512 | | | | |
| | DDP GPUs | | | | 4 | | | | |

Additionally, we show results for different rank redistributions that we obtain by using alternative measures for explained variance. Specifically, we compare EVA to using, (i), the raw eigenvalues (EVA-Raw), and (ii), normalizing by the maximum eigenvalue (EVA-Max). We report results for RoBERTa$_{Large}$ on four of the GLUE tasks, namely CoLA, RTE, MRPC, and STS-B in Table 18. Our

Table 17: Comparison of LoRA to EVA using RoBERTa$_{\text{Large}}$ on all tasks from GLUE for equal rank budgets. Mean and standard deviation of Matthew's correlation for CoLA, pearson correlation for STS-B, and accuracy for remaining datasets on the development set across 5 seeds are shown.

| Method | CoLA | MRPC | RTE | STS-B | MNLI | QNLI | QQP | SST-2 | Avg |
|---|---|---|---|---|---|---|---|---|---|
| LoRA$_{r=2}$ | $68.0_{\pm 1.4}$ | $90.9_{\pm .8}$ | $88.1_{\pm 1.1}$ | $92.3_{\pm .1}$ | $91.9_{\pm .1}$ | $94.8_{\pm .3}$ | $90.6_{\pm .1}$ | $96.1_{\pm .1}$ | 89.09 |
| EVA$_{r=2}$ | $69.1_{\pm 1.4}$ | $90.8_{\pm .5}$ | $88.2_{\pm .7}$ | $92.5_{\pm .1}$ | $90.8_{\pm .1}$ | $94.9_{\pm .1}$ | $91.9_{\pm .1}$ | $96.2_{\pm .1}$ | 89.30 |
| LoRA$_{r=4}$ | $69.1_{\pm .5}$ | $90.7_{\pm .7}$ | $86.9_{\pm .2}$ | $92.3_{\pm .1}$ | $90.6_{\pm .1}$ | $94.7_{\pm .2}$ | $92.0_{\pm .0}$ | $96.0_{\pm .1}$ | 89.04 |
| EVA$_{r=4}$ | $69.5_{\pm 1.4}$ | $91.4_{\pm .8}$ | $88.8_{\pm 1.3}$ | $92.6_{\pm .1}$ | $90.7_{\pm .0}$ | $94.9_{\pm .1}$ | $91.8_{\pm .0}$ | $96.1_{\pm .1}$ | 89.48 |
| LoRA$_{r=8}$ | $68.8_{\pm 1.0}$ | $91.1_{\pm .6}$ | $87.1_{0.7}$ | $92.2_{\pm .2}$ | $90.6_{\pm .2}$ | $94.8_{\pm .1}$ | $91.8_{\pm .0}$ | $96.2_{\pm .3}$ | 89.08 |
| EVA$_{r=8}$ | $69.0_{\pm 1.4}$ | $91.1_{\pm .4}$ | $88.4_{\pm .6}$ | $92.6_{\pm .3}$ | $90.6_{\pm .1}$ | $94.9_{\pm .1}$ | $92.1_{\pm .1}$ | $96.1_{\pm .2}$ | 89.35 |
| LoRA$_{r=16}$ | $68.4_{\pm 1.0}$ | $90.5_{\pm .5}$ | $88.0_{\pm .5}$ | $92.3_{\pm .1}$ | $90.6_{\pm .1}$ | $94.8_{\pm .1}$ | $91.9_{\pm .1}$ | $96.1_{\pm .1}$ | 89.08 |
| EVA$_{r=16}$ | $69.1_{\pm .8}$ | $91.2_{\pm .8}$ | $88.0_{\pm .5}$ | $92.6_{\pm .2}$ | $90.7_{\pm .0}$ | $95.0_{\pm .2}$ | $91.8_{\pm .0}$ | $96.2_{\pm .1}$ | 89.33 |

Table 18: Comparison of LoRA to EVA, EVA-Raw, and EVA-Max for RoBERTa$_{\text{Large}}$ on the GLUE tasks CoLA, MRPC, RTE, and STS-B. We report mean and standard deviation of Matthew's correlation for CoLA, pearson correlation for STS-B, matched accuracy for MNLI, and accuracy for remaining tasks across 5 seeds.

| Method | CoLA | MRPC | RTE | STS-B | Avg |
|---|---|---|---|---|---|
| LoRA | $69.1_{\pm .5}$ | $91.1_{\pm 0.6}$ | $88.1_{\pm 1.1}$ | $92.3_{\pm 0.1}$ | 85.2 |
| EVA | $\mathbf{69.5_{\pm 1.4}}$ | $\mathbf{91.4_{\pm 0.8}}$ | $\mathbf{88.8_{\pm 1.2}}$ | $\mathbf{92.6_{\pm 0.1}}$ | $\mathbf{85.6}$ |
| EVA-Raw | $69.4_{\pm 1.1}$ | $91.0_{\pm 0.9}$ | $88.2_{\pm 0.3}$ | $92.5_{\pm 0.2}$ | 85.3 |
| EVA-Max | $69.1_{\pm 0.5}$ | $91.2_{\pm 0.5}$ | $88.4_{\pm 1.2}$ | $92.5_{\pm 0.2}$ | 85.3 |

results show that while EVA-Raw and EVA-Max slighthly improve upon LoRA, they perform worse on average than EVA.

# D  IMAGE CLASSIFICATION

## D.1  DATASET STATISTICS

The VTAB-1K benchmark consists of 19 datasets, each containing a subset of 1000 examples of their respective samples. We summarize the dataset statistics for each dataset in Table 19. While the original train sizes of the datasets vary drastically, the 1K subset provides equal datasets across tasks. The number of classes also varies from as little as two to almost 400.

## D.2  IMPLEMENTATION DETAILS

We implemented a custom pipeline to fine-tune DINOv2-L/14 on VTAB-1K that supports LoRA, DoRA and EVA. To train AdaLora, PiSSA and OLoRA, we integrate their implementation from the `peft` library (Mangrulkar et al., 2022) into our pipeline. This pipeline is designed to be highly parallelizable and to be executed on individual GPUs. A single evaluation run of a L/14 model (all 19 datasets with hyperparameter tuning and evaluation) takes roughly 160 A100 GPU-hours but can be easily parallelized. A g/14 run takes roughly 140 H100 GPU-hours. A single evaluation run consists of 1140 hyperparameter tuning runs (19 datasets * 5 learning rates * 4 ranks * 3 seeds) and 95 evaluation runs (19 datasets * 5 seeds). Details to hyperparameter tuning are described below.

We use the original DINOv2 models (Oquab et al., 2023) and train a classification head on top of the [CLS] token, where we initialize the classification head weights with a normal distribution with $\sigma = 2e-5$ and bias with zeros. We train the classification head, LoRA matrices and biases. Images are resized to $224 \times 224$ resolution with bi-cubic interpolation and normalized with the per-channel mean and variance of ImageNet. We train all models in bfloat16 precision using the AdamW optimizer with

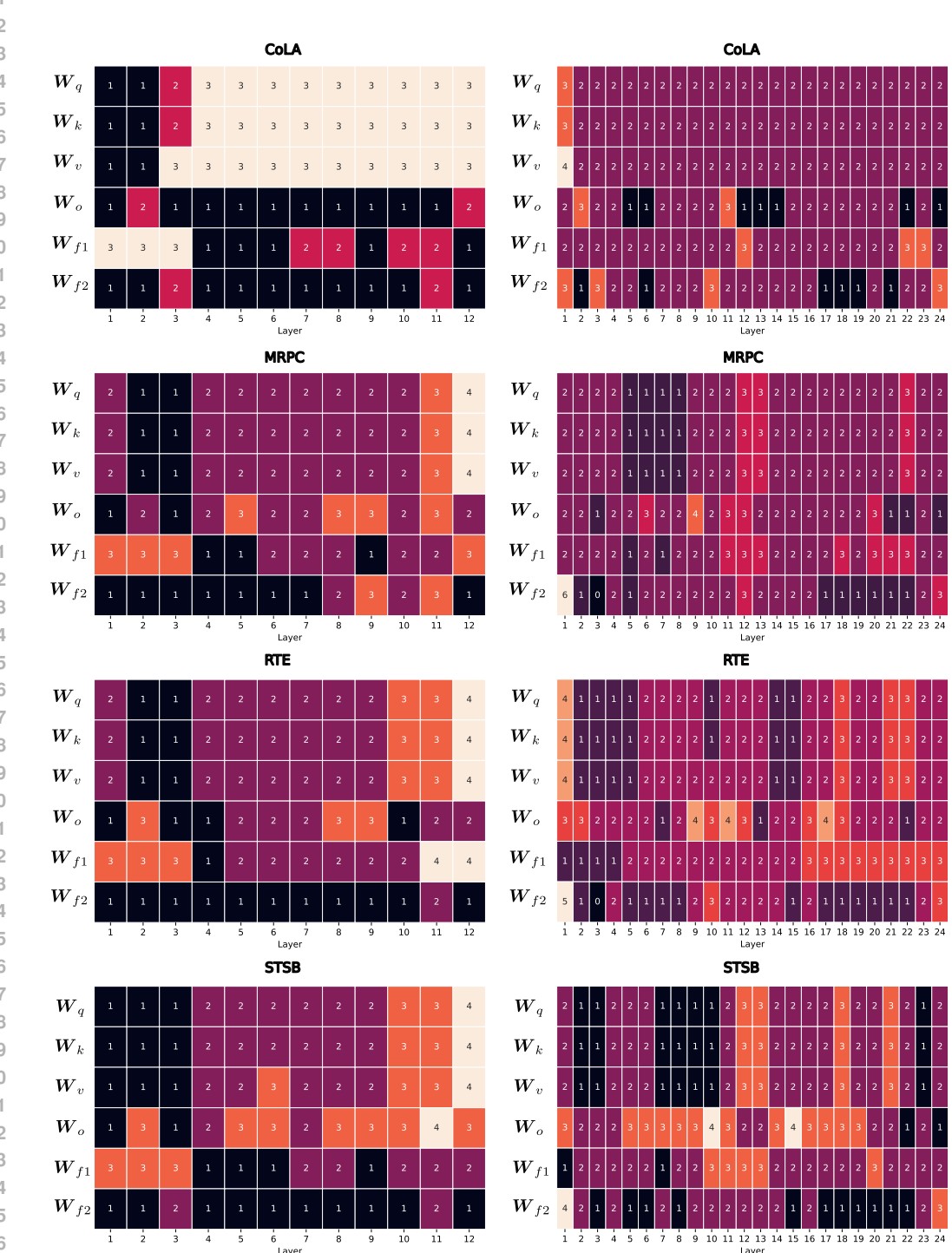

Figure 7: Rank distribution after initialization with EVA on four tasks of the GLUE benchmark (CoLA, MRPC, RTE, STSB) for DeBERTav3$_{\text{Base}}$ (left) and RoBERTa$_{\text{Large}}$ (right) with initial rank $r = 2$.

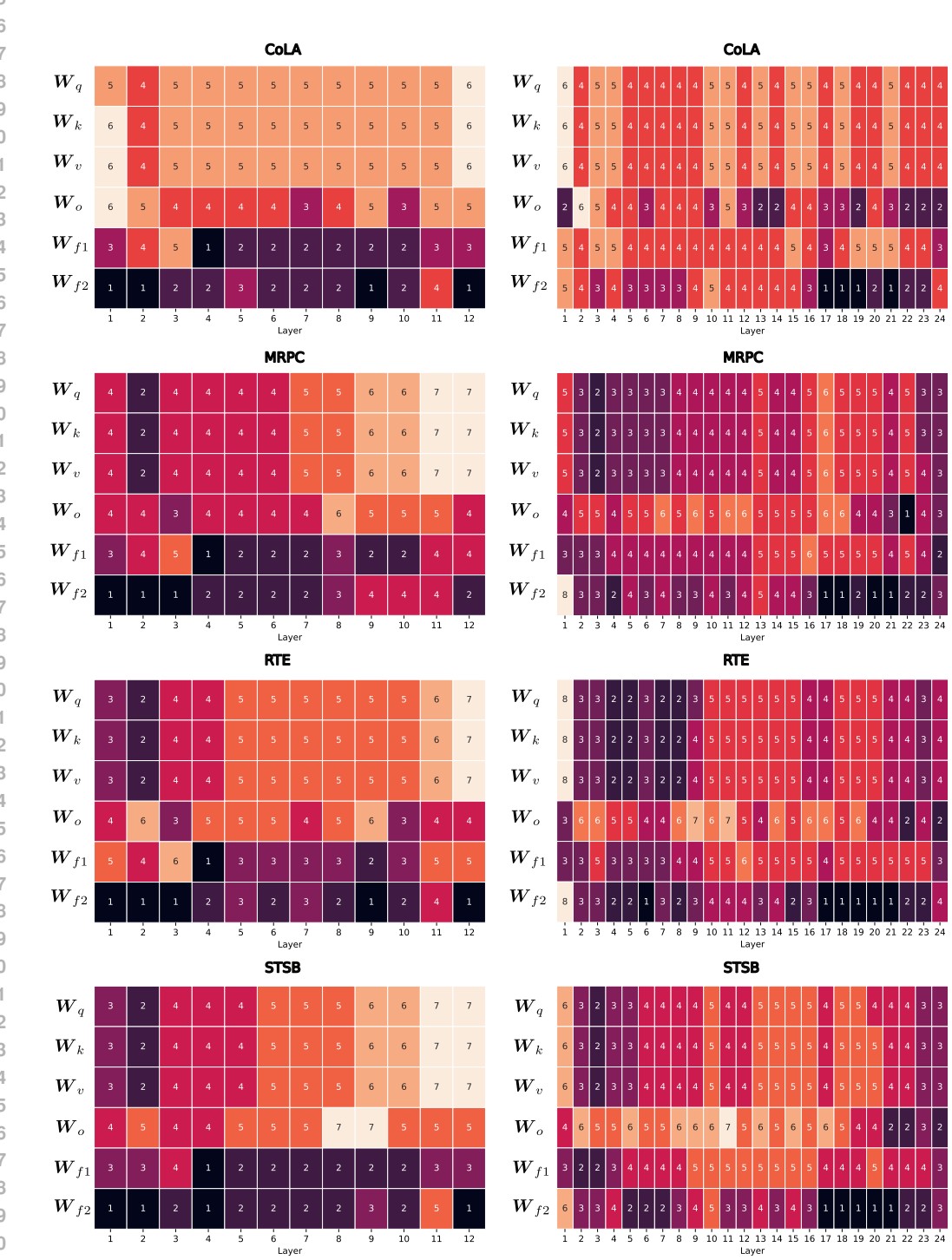

Figure 8: Rank distribution after initialization with EVA on four tasks of the GLUE benchmark (CoLA, MRPC, RTE, STSB) for DeBERTav3$_{\text{Base}}$ (left) and RoBERTa$_{\text{Large}}$ (right) with initial rank $r = 4$.

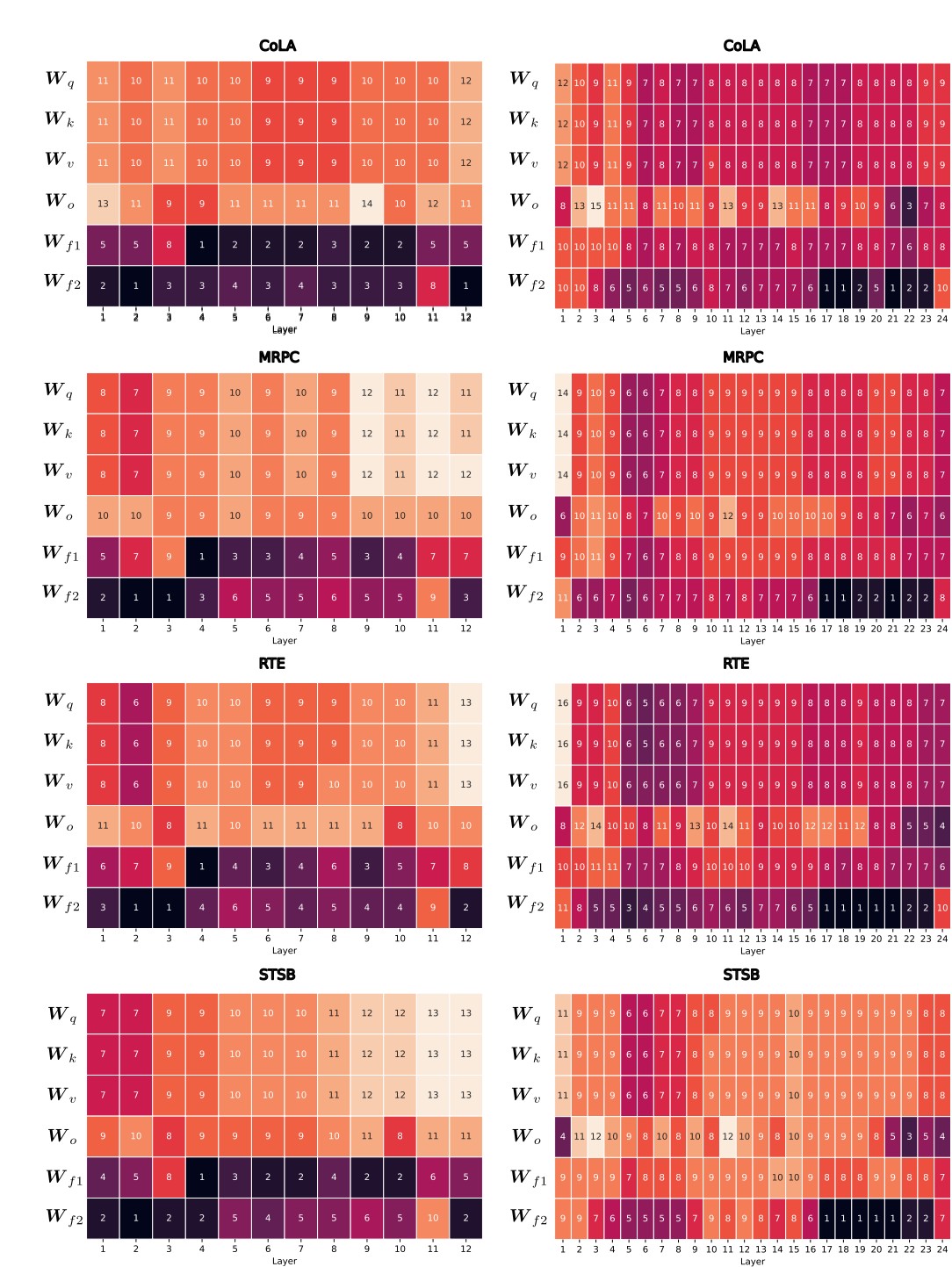

Figure 9: Rank distribution after initialization with EVA on four tasks of the GLUE benchmark (CoLA, MRPC, RTE, STSB) for DeBERTav3$_{\text{Base}}$ (left) and RoBERTa$_{\text{Large}}$ (right) with initial rank $r = 8$.

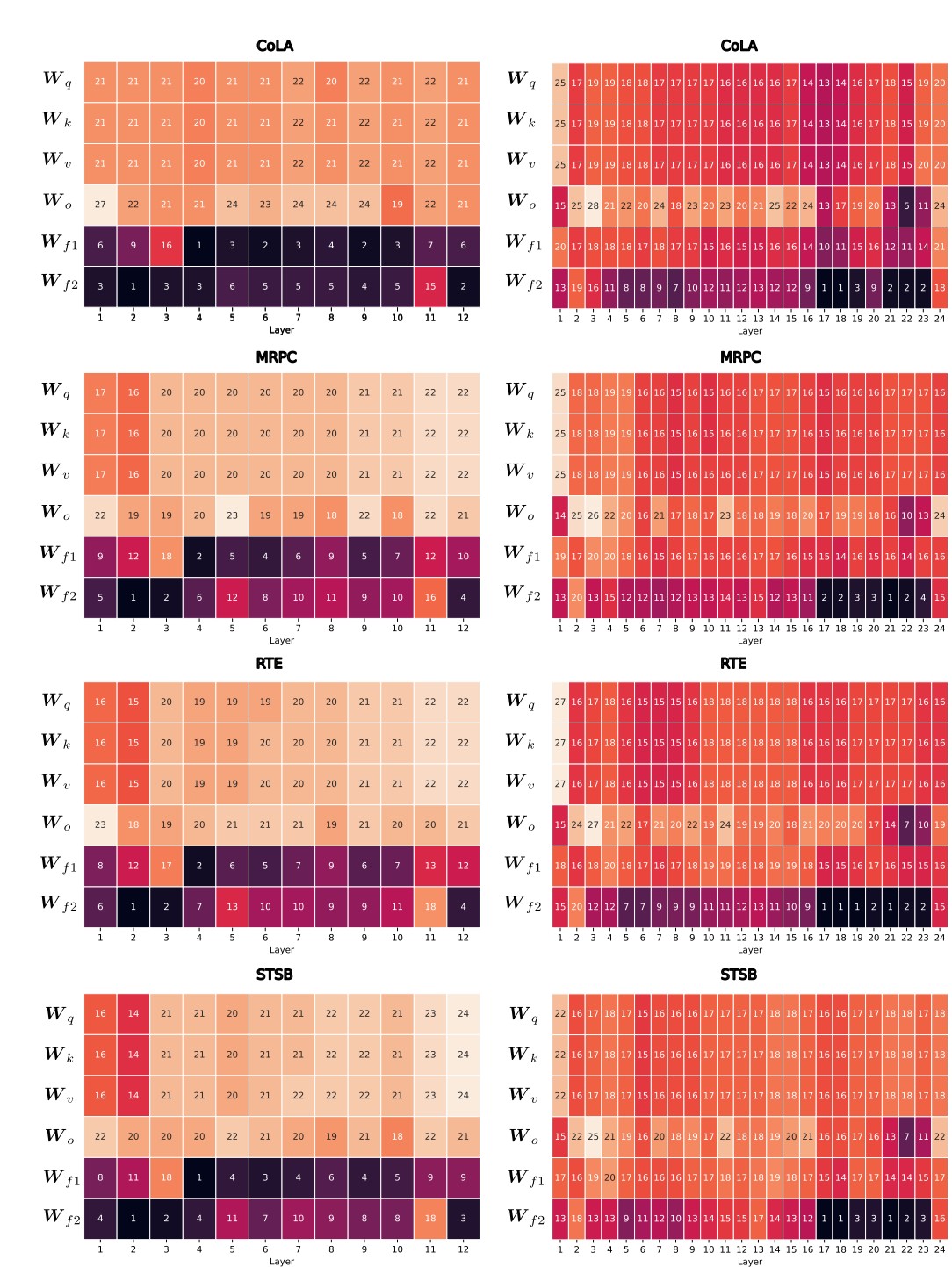

Figure 10: Rank distribution after initialization with EVA on four tasks of the GLUE benchmark (CoLA, MRPC, RTE, STSB) for DeBERTav3$_{Base}$ (left) and RoBERTa$_{Large}$ (right) with initial rank $r = 16$.

Table 19: Category, train size and classes of the VTAB-1K dataset.

| Category | Dataset | Train size | Classes |
|---|---|---:|---:|
| Natural | Caltech101 (Fei-Fei et al., 2006) | 3060 | 102 |
| Natural | CIFAR-100 (Krizhevsky, 2009) | 50000 | 100 |
| Natural | DTD (Cimpoi et al., 2014) | 3760 | 47 |
| Natural | Flowers102 (Nilsback & Zisserman, 2008) | 2040 | 102 |
| Natural | Pets (Parkhi et al., 2012) | 3680 | 37 |
| Natural | Sun397 (Xiao et al., 2010) | 87003 | 397 |
| Natural | SVHN (Netzer et al., 2011) | 73257 | 10 |
| Specialized | EuroSAT (Helber et al., 2019) | 21600 | 10 |
| Specialized | Resisc45 (Cheng et al., 2017) | 25200 | 45 |
| Specialized | Patch Camelyon (Veeling et al., 2018) | 294912 | 2 |
| Specialized | Retinopathy (Kaggle & EyePacs, 2015) | 46032 | 5 |
| Structured | Clevr/count (Johnson et al., 2017) | 70000 | 8 |
| Structured | Clevr/distance (Johnson et al., 2017) | 70000 | 6 |
| Structured | dSprites/location (Matthey et al., 2017) | 663552 | 16 |
| Structured | dSprites/orientation (Matthey et al., 2017) | 663552 | 16 |
| Structured | SmallNORB/azimuth (LeCun et al., 2004) | 36450 | 18 |
| Structured | SmallNORB/elevation (LeCun et al., 2004) | 36450 | 9 |
| Structured | DMLab (Beattie et al., 2016) | 88178 | 6 |
| Structured | KITTI/distance (Geiger et al., 2013) | 5711 | 4 |

a weight decay of 0.05 for 30 epochs. We use a cosine learning rate schedule with a linear warm-up for the first 3 epochs. Batch size is set to 64 where we use gradient accumulation if the batchsize does not fit into GPU memory. Full fine-tuning uses a layer-wise lr decay (Clark et al., 2020) of 0.75.

### D.3 Hyperparameter search

We first fine-tune on the 800 train samples of VTAB-1K datasets to find the best learning rate for the task. We sweep over learning_rate $\in \{2.5\text{e-}3, 1\text{e-}3, 7.5\text{e-}4, 5\text{e-}4, 2.5\text{e-}4\}$ and rank $\in \{2, 4, 8, 16\}$ and average the accuracy on the 200 validation samples over 3 different seeds to choose the best learning rate and rank for each dataset. For evaluation, we train on the union of train and validation set using 5 different seeds and report the average accuracy on the test set.

### D.4 Additional results

To complement our main results in Table 3, we report the respective standard deviations in Table 20.

## E Decision Making

### E.1 Dataset statistics

Meta-World (Yu et al., 2020) is an established benchmark in RL for multi-task continuous control. The benchmark consists of 50 challenging robotics tasks simulated using a Sawyer robotic arm in the MuJoCo physics engine (Todorov et al., 2012). All 50 tasks in Meta-World share the same underlying robotic arm. Therefore, all tasks share a common state (39-dimensional continuous vector) and action-space (6-dimensional). The reward functions in Meta-World are dense and based on the distance of the robotic arm to the goal location or objects. All episodes last for 200 environment interactions.

For our experiments on Meta-World, we leverage the datasets released by Schmied et al. (2024). We follow Wołczyk et al. (2021) and Schmied et al. (2024), and split the 50 tasks into 40 pre-training tasks (MT40) and 10 fine-tuning tasks (CW10). The CW10 tasks are:

Table 20: Standard deviations for the VTAB-1K results (Table 3) over 5 seeds.

| | Natural | | | | | | | Specialized | | | | Structured | | | | | | | | Average |
|---|---|---|---|---|---|---|---|---|---|---|---|---|---|---|---|---|---|---|---|---|
| | Cifar100 | Caltech101 | DTD | Flower102 | Pets | SVHN | Sun397 | Camelyon | EuroSAT | Resisc45 | Retinopathy | Clevr-Count | Clevr-Dist | DMLab | KITTI-Dist | dSpr-Loc | dSpr-Ori | sNORB-Azim | sNORB-Ele | |
| FFT | 1.5 | 1.1 | 1.6 | 0.0 | 0.4 | 1.2 | 0.9 | 14.9 | 0.4 | 0.6 | 2.7 | 1.7 | 0.9 | 1.2 | 23.6 | 0.5 | 0.4 | 1.6 | 1.9 | 3.0 |
| LoRA | 0.2 | 0.4 | 0.2 | 0.0 | 0.3 | 36.4 | **0.1** | 0.5 | 0.3 | 0.1 | 0.4 | **0.2** | 0.3 | 0.5 | 1.2 | 0.4 | 0.4 | 0.7 | 0.4 | 2.3 |
| AdaLoRA | **0.0** | **0.2** | 0.4 | 0.0 | 0.1 | 0.4 | **0.1** | 0.3 | 0.3 | 0.2 | 0.3 | 0.3 | 0.2 | 0.3 | 0.8 | 0.8 | 0.3 | 0.3 | 0.4 | **0.3** |
| PiSSA | 0.2 | 0.4 | 0.3 | 0.0 | 0.2 | 0.5 | 0.2 | 0.7 | 0.2 | **0.1** | 0.4 | 0.3 | 0.4 | **0.2** | 0.7 | **0.3** | 0.5 | 0.4 | 0.5 | 0.3 |
| OLoRA | 0.3 | 0.3 | 0.4 | 0.0 | 0.3 | 29.4 | 0.1 | 0.3 | **0.1** | 0.2 | **0.2** | 0.5 | **0.1** | 0.3 | 24.6 | 0.3 | 0.4 | 0.3 | 0.8 | 3.1 |
| EVA | 0.2 | 0.5 | **0.2** | 0.0 | **0.1** | **0.3** | **0.1** | **0.3** | 0.2 | 0.3 | 0.4 | 0.5 | 0.3 | 0.6 | 0.6 | 0.5 | 0.5 | **0.2** | 0.5 | 0.3 |
| DoRA | 0.1 | 0.2 | 0.5 | 0.0 | 0.2 | 29.7 | 0.4 | 0.7 | 0.1 | 0.2 | 0.4 | 0.4 | 0.3 | 0.3 | **0.6** | 36.2 | 0.5 | 0.3 | **0.3** | 3.8 |
| EVA+DoRA | 0.2 | 1.3 | 0.6 | 0.0 | 0.3 | 0.5 | 0.3 | 0.4 | 0.2 | 0.3 | 0.3 | 0.4 | 0.4 | 12.8 | 1.3 | 2.5 | **0.3** | 0.6 | 0.6 | 1.2 |

`hammer-v2`, `push-wall-v2`, `faucet-close-v2`, `push-back-v2`, `stick-pull-v2`, `stick-pull-v2`, `handle-press-side-v2`, `push-v2`, `shelf-place-v2`, `window-close-v2`, and `peg-unplug-side-v2`.

The datasets contain 2M transitions for every of the 50 tasks, amounting to 80M transitions (320M tokens) across all training tasks. The average success rate and rewards across all MT40 tasks are 84% and 1414.62, respectively. We list the statistics per task in Table 21.

### E.2 IMPLEMENTATION DETAILS

We implemented our pipeline that supports training for Meta-World on top of the code-base provided by Schmied et al. (2024). Our custom implementation supports training LoRA, DoRA and EVA. Furthermore, we leverage the `peft` library (Mangrulkar et al., 2022) to train the remaining methods.

For our experiments on Meta-World, we use a GPT2-like network architecture (Radford et al., 2019) with 4 Transformer layers, 8 heads, and hidden dimension of 512 resulting in 16M parameters. We use a context of 50 time steps, which amounts to a sequence length of 200, as each timestep contains states, actions, rewards and RTGs. We embed states, actions, rewards and return-to-gos (RTGs) using separate linear embedding layers per modality, as proposed by Chen et al. (2021a). We train with a batch size of 128 using a constant learning rate of $1e^{-4}$, 4000 linear warm-up steps followed by a cosine decay to $1e^{-6}$, using the AdamW optimizer (Loshchilov & Hutter, 2017). We employ gradient clipping of 0.25, weight decay of 0.01, and a dropout rate of 0.2. Our DT implementation employs global position embedding. For every task, we set the target return to the maximum return achieved in the respective training datasets, as proposed by (Schmied et al., 2024). Furthermore, we employ mixed-precision (Micikevicius et al., 2017) and flash-attention (Dao, 2023) to speed-up training.

We first **pre-train** a DT on all MT40 tasks (80M transitions) for 1M updates via next-action prediction by minimizing the mean-squared error. The resulting pre-trained model attains an average success rate of 80% across all MT40 tasks. Then we **fine-tune** the DT on each of the CW10 down-stream tasks for 100K updates with the same set of hyperparameters as used for pre-training. We run all our experiments on a public research cluster with 4xA100-40GB GPU nodes. A single fine-tuning run with EVA for one task takes roughly 1 hour on one A100.

### E.3 HYPERPARAMETER SEARCH

In line with previous experiments, we tune the rank for LoRA, DoRA, AdaLora and EVA, rank $\in$ $\{2, 4, 8, 16\}$. Further, we sweep over the same learning rates as for the GLUE tasks.

Table 21: Dataset statistics for all MT40 tasks from Schmied et al. (2024).

| Task | $|\mathcal{S}|$ | $|\mathcal{A}|$ | Success Rate | Reward |
|---|---|---|---|---|
| assembly-v2 | 39 | 4 | 0.0 | 1206.9 |
| basketball-v2 | 39 | 4 | 0.9 | 1375.95 |
| bin-picking-v2 | 39 | 4 | 0.0 | 474.81 |
| box-close-v2 | 39 | 4 | 0.0 | 759.15 |
| button-press-topdown-v2 | 39 | 4 | 1.0 | 1299.24 |
| button-press-topdown-wall-v2 | 39 | 4 | 1.0 | 1296.16 |
| button-press-v2 | 39 | 4 | 1.0 | 1430.44 |
| button-press-wall-v2 | 39 | 4 | 1.0 | 1508.16 |
| coffee-button-v2 | 39 | 4 | 1.0 | 1499.17 |
| coffee-pull-v2 | 39 | 4 | 1.0 | 1313.88 |
| coffee-push-v2 | 39 | 4 | 0.6 | 508.14 |
| dial-turn-v2 | 39 | 4 | 0.8 | 1674.29 |
| disassemble-v2 | 39 | 4 | 1.0 | 1396.55 |
| door-close-v2 | 39 | 4 | 1.0 | 1535.4 |
| door-lock-v2 | 39 | 4 | 1.0 | 1712.65 |
| door-open-v2 | 39 | 4 | 1.0 | 1544.32 |
| door-unlock-v2 | 39 | 4 | 1.0 | 1733.64 |
| drawer-close-v2 | 39 | 4 | 1.0 | 1845.92 |
| drawer-open-v2 | 39 | 4 | 1.0 | 1710.65 |
| faucet-open-v2 | 39 | 4 | 0.9 | 1727.98 |
| hand-insert-v2 | 39 | 4 | 1.0 | 1607.17 |
| handle-press-v2 | 39 | 4 | 1.0 | 1854.79 |
| handle-pull-side-v2 | 39 | 4 | 1.0 | 1613.72 |
| handle-pull-v2 | 39 | 4 | 1.0 | 1581.75 |
| lever-pull-v2 | 39 | 4 | 1.0 | 1449.05 |
| peg-insert-side-v2 | 39 | 4 | 1.0 | 1545.19 |
| pick-out-of-hole-v2 | 39 | 4 | 1.0 | 1435.64 |
| pick-place-v2 | 39 | 4 | 0.0 | 6.59 |
| pick-place-wall-v2 | 39 | 4 | 0.1 | 702.59 |
| plate-slide-back-side-v2 | 39 | 4 | 1.0 | 1766.24 |
| plate-slide-back-v2 | 39 | 4 | 1.0 | 1773.56 |
| plate-slide-side-v2 | 39 | 4 | 1.0 | 1663.35 |
| plate-slide-v2 | 39 | 4 | 1.0 | 1667.35 |
| reach-v2 | 39 | 4 | 1.0 | 1858.99 |
| reach-wall-v2 | 39 | 4 | 1.0 | 1831.14 |
| soccer-v2 | 39 | 4 | 0.4 | 445.84 |
| stick-push-v2 | 39 | 4 | 1.0 | 1470.71 |
| sweep-into-v2 | 39 | 4 | 1.0 | 1761.69 |
| sweep-v2 | 39 | 4 | 1.0 | 1458.35 |
| window-open-v2 | 39 | 4 | 1.0 | 1537.59 |
| Average | - | - | $0.84 \pm 0.34$ | $1414.62 \pm 439.39$ |

### E.4 ADDITIONAL RESULTS

In Table 22, we show the full comparison for all methods on CW10. EVA+DoRA consistently outperforms all competitors for the different rank budgets.

Table 22: Rank-wise comparison for all methods on CW10. We fine-tune a 12M DT on 10 tasks individually and report the mean success rates/rewards ($\pm$ standard error) for every task.

| Method | Rank | faucet-close | hammer | handle-press-side | peg-unplug-side | push-back | push | push-wall | shelf-place | stick-pull | window-close | Average |
|---|---|---|---|---|---|---|---|---|---|---|---|---|
| FFT | - | $0.97_{\pm0.03}$ | $0.93_{\pm0.03}$ | $1.0_{\pm0.0}$ | $0.6_{\pm0.05}$ | $0.7_{\pm0.12}$ | $1.0_{\pm0.0}$ | $0.93_{\pm0.03}$ | $1.0_{\pm0.0}$ | $0.57_{\pm0.07}$ | $1.0_{\pm0.0}$ | $0.87_{\pm0.03}$ |
| LoRA | 2 | $1.0_{\pm0.0}$ | $1.0_{\pm0.0}$ | $1.0_{\pm0.0}$ | $0.6_{\pm0.05}$ | $0.57_{\pm0.07}$ | $0.97_{\pm0.03}$ | $0.93_{\pm0.03}$ | $1.0_{\pm0.0}$ | $0.37_{\pm0.1}$ | $1._{\pm0.0}$ | $0.84_{\pm0.04}$ |
| | 4 | $1.0_{\pm0.0}$ | $0.97_{\pm0.03}$ | $1.0_{\pm0.0}$ | $0.47_{\pm0.12}$ | $0.63_{\pm0.1}$ | $0.97_{\pm0.03}$ | $1.0_{\pm0.0}$ | $1.0_{\pm0.0}$ | $0.23_{\pm0.12}$ | $1.0_{\pm0.0}$ | $0.83_{\pm0.05}$ |
| | 8 | $1.0_{\pm0.0}$ | $0.97_{\pm0.03}$ | $1.0_{\pm0.0}$ | $0.43_{\pm0.05}$ | $0.4_{\pm0.09}$ | $0.97_{\pm0.03}$ | $0.93_{\pm0.03}$ | $1.0_{\pm0.0}$ | $0.23_{\pm0.12}$ | $1.0_{\pm0.0}$ | $0.79_{\pm0.06}$ |
| | 16 | $1.0_{\pm0.0}$ | $0.97_{\pm0.03}$ | $1.0_{\pm0.0}$ | $0.43_{\pm0.03}$ | $0.47_{\pm0.03}$ | $1.0_{\pm0.0}$ | $0.97_{\pm0.03}$ | $1.0_{\pm0.0}$ | $0.4_{\pm0.09}$ | $1.0_{\pm0.0}$ | $0.82_{\pm0.05}$ |
| DoRA | 2 | $1.0_{\pm0.0}$ | $1.0_{\pm0.0}$ | $1.0_{\pm0.0}$ | $0.57_{\pm0.05}$ | $1.0_{\pm0.0}$ | $1.0_{\pm0.0}$ | $1.0_{\pm0.0}$ | $1.0_{\pm0.0}$ | $0.33_{\pm0.11}$ | $1.0_{\pm0.0}$ | $0.89_{\pm0.04}$ |
| | 4 | $1.0_{\pm0.0}$ | $1.0_{\pm0.0}$ | $1.0_{\pm0.0}$ | $0.6_{\pm0.12}$ | $1.0_{\pm0.0}$ | $1.0_{\pm0.0}$ | $1.0_{\pm0.0}$ | $1.0_{\pm0.0}$ | $0.43_{\pm0.12}$ | $1.0_{\pm0.0}$ | $0.9_{\pm0.04}$ |
| | 8 | $1.0_{\pm0.0}$ | $1.0_{\pm0.0}$ | $1.0_{\pm0.0}$ | $0.47_{\pm0.12}$ | $0.93_{\pm0.05}$ | $1.0_{\pm0.0}$ | $1.0_{\pm0.0}$ | $1.0_{\pm0.0}$ | $0.57_{\pm0.15}$ | $1.0_{\pm0.0}$ | $0.9_{\pm0.04}$ |
| | 16 | $1.0_{\pm0.0}$ | $1.0_{\pm0.0}$ | $1.0_{\pm0.0}$ | $0.57_{\pm0.12}$ | $1.0_{\pm0.0}$ | $1.0_{\pm0.0}$ | $1.0_{\pm0.0}$ | $1.0_{\pm0.0}$ | $0.67_{\pm0.15}$ | $1.0_{\pm0.0}$ | $0.92_{\pm0.03}$ |
| AdaLoRA | 2 | $1.0_{\pm0.0}$ | $0.97_{\pm0.03}$ | $1.0_{\pm0.0}$ | $0.37_{\pm0.05}$ | $0.37_{\pm0.05}$ | $0.93_{\pm0.05}$ | $0.97_{\pm0.03}$ | $1.0_{\pm0.0}$ | $0.13_{\pm0.07}$ | $1.0_{\pm0.0}$ | $0.77_{\pm0.06}$ |
| | 4 | $1.0_{\pm0.0}$ | $0.97_{\pm0.03}$ | $1.0_{\pm0.0}$ | $0.37_{\pm0.07}$ | $0.57_{\pm0.1}$ | $0.97_{\pm0.03}$ | $0.9_{\pm0.08}$ | $1.0_{\pm0.0}$ | $0.13_{\pm0.07}$ | $1.0_{\pm0.0}$ | $0.79_{\pm0.06}$ |
| | 8 | $1.0_{\pm0.0}$ | $0.97_{\pm0.03}$ | $1.0_{\pm0.0}$ | $0.3_{\pm0.05}$ | $0.57_{\pm0.14}$ | $0.93_{\pm0.03}$ | $0.87_{\pm0.07}$ | $1.0_{\pm0.0}$ | $0.0_{\pm0.0}$ | $1.0_{\pm0.0}$ | $0.76_{\pm0.06}$ |
| | 16 | $1.0_{\pm0.0}$ | $0.97_{\pm0.03}$ | $1.0_{\pm0.0}$ | $0.4_{\pm0.09}$ | $0.57_{\pm0.12}$ | $0.97_{\pm0.03}$ | $0.93_{\pm0.05}$ | $1.0_{\pm0.0}$ | $0.0_{\pm0.0}$ | $1.0_{\pm0.0}$ | $0.78_{\pm0.06}$ |
| OLoRA | 2 | $1.0_{\pm0.0}$ | $0.9_{\pm0.05}$ | $1.0_{\pm0.0}$ | $0.47_{\pm0.03}$ | $0.33_{\pm0.03}$ | $0.97_{\pm0.03}$ | $0.97_{0.03}$ | $1.0_{\pm0.0}$ | $0.27_{\pm0.11}$ | $1.0_{\pm0.0}$ | $0.79_{\pm0.05}$ |
| | 4 | $1.0_{\pm0.0}$ | $0.9_{\pm0.05}$ | $1.0_{\pm0.0}$ | $0.43_{\pm0.03}$ | $0.63_{\pm0.12}$ | $1.0_{\pm0.0}$ | $1.0_{0.0}$ | $1.0_{\pm0.0}$ | $0.6_{\pm0.12}$ | $1.0_{\pm0.0}$ | $0.86_{\pm0.04}$ |
| | 8 | $1.0_{\pm0.0}$ | $0.97_{\pm0.03}$ | $1.0_{\pm0.0}$ | $0.57_{\pm0.1}$ | $0.5_{\pm0.08}$ | $1.0_{\pm0.0}$ | $1.0_{0.0}$ | $1.0_{\pm0.0}$ | $0.53_{\pm0.14}$ | $1.0_{\pm0.0}$ | $0.86_{\pm0.04}$ |
| | 16 | $1.0_{\pm0.0}$ | $0.97_{\pm0.03}$ | $1.0_{\pm0.0}$ | $0.4_{\pm0.05}$ | $0.63_{\pm0.03}$ | $1.0_{\pm0.0}$ | $1.0_{0.0}$ | $1.0_{\pm0.0}$ | $0.43_{\pm0.05}$ | $1.0_{\pm0.0}$ | $0.84_{\pm0.04}$ |
| PiSSA | 2 | $1.0_{\pm0.0}$ | $0.97_{\pm0.03}$ | $1.0_{\pm0.0}$ | $0.43_{\pm0.11}$ | $0.53_{\pm0.07}$ | $0.97_{\pm0.03}$ | $0.9_{0.08}$ | $1.0_{\pm0.0}$ | $0.33_{\pm0.17}$ | $1.0_{\pm0.0}$ | $0.81_{\pm0.05}$ |
| | 4 | $1.0_{\pm0.0}$ | $1.0_{\pm0.0}$ | $1.0_{\pm0.0}$ | $0.37_{\pm0.07}$ | $0.7_{\pm0.05}$ | $0.97_{\pm0.03}$ | $1.0_{0.0}$ | $1.0_{\pm0.0}$ | $0.07_{\pm0.05}$ | $1.0_{\pm0.0}$ | $0.81_{\pm0.06}$ |
| | 8 | $1.0_{\pm0.0}$ | $0.97_{\pm0.03}$ | $1.0_{\pm0.0}$ | $0.3_{\pm0.0}$ | $0.57_{\pm0.03}$ | $0.97_{\pm0.03}$ | $1.0_{0.0}$ | $1.0_{\pm0.0}$ | $0.53_{\pm0.1}$ | $1.0_{\pm0.0}$ | $0.83_{\pm0.05}$ |
| | 16 | $1.0_{\pm0.0}$ | $0.93_{\pm0.03}$ | $1.0_{\pm0.0}$ | $0.33_{\pm0.12}$ | $0.47_{\pm0.03}$ | $1.0_{\pm0.0}$ | $0.97_{0.03}$ | $1.0_{\pm0.0}$ | $0.47_{\pm0.11}$ | $1.0_{\pm0.0}$ | $0.82_{\pm0.05}$ |
| EVA | 2 | $1.0_{\pm0.0}$ | $0.97_{\pm0.03}$ | $1.0_{\pm0.0}$ | $0.43_{\pm0.07}$ | $0.77_{\pm0.05}$ | $0.97_{\pm0.03}$ | $1.0_{0.0}$ | $1.0_{\pm0.0}$ | $0.63_{\pm0.07}$ | $1.0_{\pm0.0}$ | $0.88_{\pm0.04}$ |
| | 4 | $1.0_{\pm0.0}$ | $0.97_{\pm0.03}$ | $1.0_{\pm0.0}$ | $0.43_{\pm0.05}$ | $0.47_{\pm0.12}$ | $1.0_{\pm0.0}$ | $0.97_{\pm0.03}$ | $1.0_{\pm0.0}$ | $0.23_{\pm0.05}$ | $1.0_{\pm0.0}$ | $0.81_{\pm0.05}$ |
| | 8 | $1.0_{\pm0.0}$ | $0.97_{\pm0.03}$ | $1.0_{\pm0.0}$ | $0.63_{\pm0.03}$ | $0.7_{\pm0.08}$ | $1.0_{\pm0.0}$ | $1.0_{\pm0.0}$ | $1.0_{\pm0.0}$ | $0.23_{\pm0.03}$ | $1.0_{\pm0.0}$ | $0.85_{\pm0.05}$ |
| | 16 | $1.0_{\pm0.0}$ | $0.97_{\pm0.03}$ | $1.0_{\pm0.0}$ | $0.53_{\pm0.03}$ | $0.77_{\pm0.07}$ | $1.0_{\pm0.0}$ | $1.0_{\pm0.0}$ | $1.0_{\pm0.0}$ | $0.0_{\pm0.0}$ | $1.0_{\pm0.0}$ | $0.83_{\pm0.06}$ |
| EVA + DoRA | 2 | $1.0_{\pm0.0}$ | $1.0_{\pm0.0}$ | $1.0_{\pm0.0}$ | $0.8_{\pm0.08}$ | $0.97_{\pm0.03}$ | $1.0_{\pm0.0}$ | $1.0_{\pm0.0}$ | $1.0_{\pm0.0}$ | $0.43_{\pm0.12}$ | $1.0_{\pm0.0}$ | $0.92_{\pm0.03}$ |
| | 4 | $1.0_{\pm0.0}$ | $1.0_{\pm0.0}$ | $1.0_{\pm0.0}$ | $0.8_{\pm0.05}$ | $0.93_{\pm0.03}$ | $1.0_{\pm0.0}$ | $1.0_{\pm0.0}$ | $1.0_{\pm0.0}$ | $0.63_{\pm0.03}$ | $1.0_{\pm0.0}$ | $0.94_{\pm0.02}$ |
| | 8 | $1.0_{\pm0.0}$ | $1.0_{\pm0.0}$ | $1.0_{\pm0.0}$ | $0.63_{\pm0.19}$ | $0.87_{\pm0.07}$ | $1.0_{\pm0.0}$ | $1.0_{\pm0.0}$ | $1.0_{\pm0.0}$ | $0.57_{\pm0.03}$ | $1.0_{\pm0.0}$ | $0.91_{\pm0.04}$ |
| | 16 | $1.0_{\pm0.0}$ | $1.0_{\pm0.0}$ | $1.0_{\pm0.0}$ | $0.67_{\pm0.2}$ | $1.0_{\pm0.0}$ | $1.0_{\pm0.0}$ | $1.0_{\pm0.0}$ | $1.0_{\pm0.0}$ | $0.5_{\pm0.16}$ | $1.0_{\pm0.0}$ | $0.92_{\pm0.04}$ |

## F INCREMENTAL SVD CONVERGENCE ANALYSIS

For simplicity, let us assume that $A = X_0^{i\top}$ and $B = X_1^{i\top}$ are two batches of activations for weight matrix $W^i$ obtained by passing two subsequent batches of the downstream data through the model. The aim is now to compute the SVD of the concatenated activation matrix $\begin{bmatrix} A B \end{bmatrix} = U'\Sigma'V'^{\top}$ in constant memory. Further, We obtain $A = U_t \Sigma_t V_t^{\top}$ via SVD. Now let $\tilde{B}$ be the component of $B$ that is orthogonal to $U$, which can be obtained via QR-decompositon or via $\tilde{B} = \mathrm{orth}(B - UU^{\top}B)$, where $\mathrm{orth}(\cdot)$ performs orthogonalization. Then the SVD of the concatenated activation matrix can be expressed in partitioned form as

$$\begin{bmatrix} A B \end{bmatrix} = \begin{bmatrix} U \tilde{B} \end{bmatrix} \begin{bmatrix} \Sigma & U^{\top}B \\ 0 & \tilde{B}^{\top}B \end{bmatrix} \begin{bmatrix} V^{\top} & 0 \\ 0 & I \end{bmatrix}. \tag{4}$$

By setting $R = \begin{bmatrix} \Sigma & U^{\top}B \\ 0 & \tilde{B}B \end{bmatrix}$, we can obtain SVD of the concatenated activation matrix by performing SVD on $R$, $R = \tilde{U}\tilde{\Sigma}\tilde{V}^{\top}$, which is constant in time and memory as we only need to compute $\tilde{U}'$ and $\Sigma'$, which do not scale with the number of data samples. Hence, we perform

$$[A; B] = \left( \begin{bmatrix} U; \tilde{B} \end{bmatrix} \tilde{U} \right) \tilde{\Sigma} \left( \tilde{V}^{\top} \begin{bmatrix} V^{\top} & 0 \\ 0 & I \end{bmatrix} \right), \tag{5}$$

and subsequently obtain $U' = \begin{bmatrix} U \tilde{B} \end{bmatrix} \tilde{U}$ and $\Sigma' = \tilde{\Sigma}$.

As this algorithm incrementally updates the $U$ and $\Sigma$ components, we need to keep track of changing mean and variance estimates. For the mean this is trivial, but the computation of running variances can introduce numerical instabilities. To counteract this, usually the *young and cramer update* is employed (Chan et al., 1983). The supporting proof that the covariance matrix of the original data matrix is equal to the covariance matrix of the concatenated matrix up to a constant factor is given in Ross et al. (2008). In our example, the left-singular values $U$ do not scale with the number of samples. However, in our case we have $A = X_t^i$ and $B = X_{t+1}^i$, i.e. transposed data matrices, therefore it is the right-singular values $V$ that do not depend on the number of samples and can be incrementally updated in constant time and memory. We show pseudocode for the incremental SVD algorithm in Algorithm 2.

---

**Algorithm 2** Incremental SVD algorithm from Ross et al. (2008)

---

**Input:** Sequence of data batches $\{A^0, \ldots, A^T\}$, truncated SVD $\text{SVD}(\cdot)$, orthogonalization function $\text{orth}(\cdot)$, running variance update function $\text{young\_cramer\_update}(\cdot, \cdot)$

1: $\bar{m}^0 \leftarrow \frac{1}{b} \sum_{i=0}^{b} A_{:,i}$, $\sigma^0 \leftarrow \frac{\sum_{i=0}^{b}(A_{:,i} - \bar{m}^0)^2}{b-1}$  ▷ initialize incremental mean/variance

2: $U_0 \Sigma_0 V^\top \leftarrow \text{SVD}(A^0 - \bar{a}^0)$  ▷ Perform initial SVD on $A$ to get initial components

3: **for** $i$ in $1, \ldots, T$ **do**

4:  $\quad \bar{a}^i \leftarrow \frac{1}{b} \sum_b A_{:,i}^i$, $\bar{m}^i \leftarrow \bar{m}^i + \frac{a^i - \bar{m}^{i-1}}{b(i+1)}$  ▷ compute mean vectors

5:  $\quad \sigma^i \leftarrow \text{young\_cramer\_update}(\sigma^{i-1}, A^i)$  ▷ Update running variance

6:  $\quad \hat{A}^i \leftarrow \left[ A^i - \bar{a}^i; \sqrt{\frac{b(i+1)}{2b}} \left( \bar{m}^i - \bar{a}^i \right) \right]$  ▷ concatenate mean correction factor

7:  $\quad \tilde{A}^i \leftarrow \text{orth}(\hat{A}^i - U_{i-1} U_{i-1}^\top \hat{A}^i)$  ▷ Obtain orthogonal component to $U$

8:  $\quad R = \begin{bmatrix} \Sigma_{i-1} & U_{i-1}^\top \hat{A}^i \\ 0 & \tilde{A}^i \hat{A}^i \end{bmatrix}$  ▷ Define matrix $R$

9:  $\quad \tilde{U} \tilde{\Sigma} \tilde{V}^\top \leftarrow \text{SVD}(R)$  ▷ Perform SVD on $R$

10:  $\quad U_i \leftarrow \left[ U_{i-1}; \tilde{A}^i \right] \tilde{U}$, $\Sigma_i \leftarrow \tilde{\Sigma}$  ▷ Update SVD components

11: **end for**

---

In the following sections we analyze the behavior of this algorithm under different conditions, i.e. different batch sizes, etc.

## F.1  COMPLEXITY

The computation of SVD introduces computational overhead in the initial training stage. Since we do not require gradient computation or storing of optimizer states, there is no overhead in terms of memory. SVD has a time complexity of $\mathcal{O}(\min(b^2 d, bd^2))$ which can be reduced to $\mathcal{O}(k^2 b)$ for $k << d$ by randomly choosing $k$ columns from $X$ as introduced in Halko et al. (2011). Let $T$ be the number of minibatches until all components are converged for $N$ weight matrices, then the time complexity is $\mathcal{O}(NTk^2 b)$. In other words, the complexity scales linearly with the number of weight matrices and the number of minibatches. To speed up the computation of SVD, we provide an implementation that runs entirely on GPU.

## F.2  BATCH SIZE INVARIANCE

We conduct an analysis on the convergence of the components obtained via SVD. Specifically, we investigate the difference in components according to cosine similarity across different batch sizes. Previously we have seen that the components obtained across different batch orderings are heavily correlated. In Figure 11 we visualize the cosine similarities between the SVD components for different batch sizes, namely 4, 8, 16, and 32 for Llama-2-7B on the MetaMathQA dataset. We observe that the components correlate strongly and remain mostly invariant to the batch size. This indicates that smaller batch sizes may be used for obtaining the initialization which results in less computational overhead. In the case of Llama-2-7B on MetaMathQA, this means that we can use a batch size of 4 since it induces a computational overhead of around 100 seconds. Afterwards we can continue the fine-tuning process with a larger batch size.

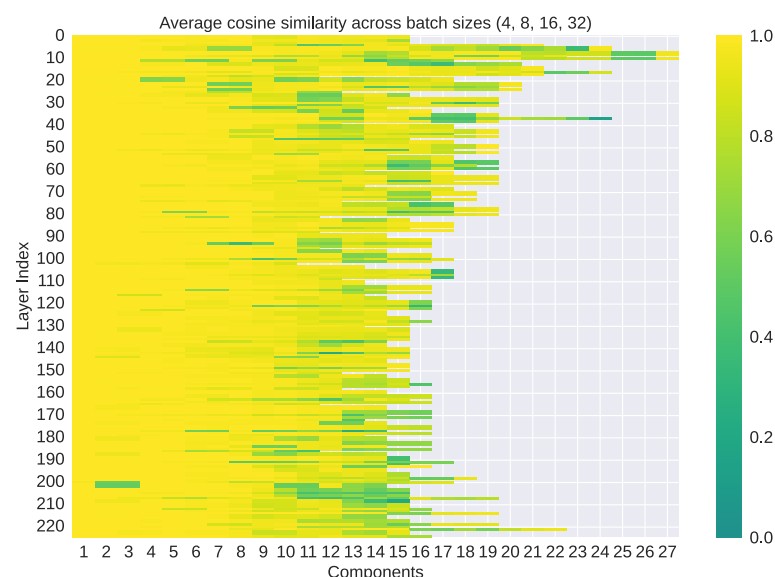

Figure 11: Average cosine similarity between components obtained via SVD on minibatches of activation vectors across different batch sizes. The components strongly correlate indicating that the SVD computation is mostly invariant to the batch size and returns mostly the same components.

### F.3 EXCLUDING IGNORED TOKENS FOR SVD

For some datasets we notice that masking out tokens for the SVD computation which are ignored for the loss calculation during finetuning can be advantageous. This can however result in a significant reduction of the effective batch size for SVD if the number of completion tokens is small. An example where this is the case in our experiments are the common sense reasoning tasks which have long prompts but completion tokens are only one word per sample. This setting can lead to cases were SVD does not converge for lower batch sizes. We therefore do not mask out the prompt tokens in our experiments. Another setting where masking ignored tokens can be advantageous are multi-turn conversation where the model is only trained on the assistant tokens. To achieve the results in Table 13 we mask out user tokens together with the prompt for the SVD computation.

### F.4 EFFICIENCY OF EVA INITIALIZATION

We investigate the efficacy of the incremental SVD for obtaining a data-driven initialization to LoRA-GA (Wang et al., 2024), another concurrent work on data-driven initialization. LoRA-GA performs SVD on the full gradient matrix to obtain a lower dimensional subspace approximation and initializes $A$ and $B$ accordingly. In Table 23 we show the wall clock time required for LoRA-GA and EVA as a fraction of the total training time. We observe that EVA takes up only 0.7% of the training time for initialization, while LoRA-GA takes approximately 4.8%. This demonstrates the EVA is approximately seven times faster than LoRA-GA while achieving better performance. Furthermore, EVA is even faster than PiSSA even though PiSSA is weight-driven. Finally, even though EVA is slightly slower than OLoRA, it attains a better performance vs complexity trade-off as it outperforms OLoRA on average on all our experiments.

## G RANK RE-DISTRIBUTION ANALYSIS

To illuminate the rank re-distribution process, we visualize the resulting ranks for each weight matrix after SVD for Llama-2-7B on the MetaMathQA dataset for different values of $\rho$. Setting $\rho = 1$ results in a uniform rank distribution as in standard LoRA. However, setting $\rho > 1$ alters the number of ranks per weight matrix. In Figure 12 we visualize the number of ranks assigned to each weight matrix for different values of $\rho > 1$ and in Figure 13 we visualize the corresponding deltas. Both

Table 23: Time in minutes required for computing initialization of LoRA-GA, PiSSA and EVA as % of total training time for Llama-2-7B on a single A100 GPU fine-tuned on the common sense reasoning tasks presented in Table 7. Training time is averaged across two runs for one epoch. For LoRA-GA we use the default number of steps (64). For EVA we report efficiency across different batch sizes.

| Initialization | Method | Initialization | Training | % of Training |
|---|---|---|---|---|
| **Weight-driven** | PiSSA | 7.43 | 482.67 | 1.5 |
| | OLoRA | 0.3 | 482.67 | 0.1 |
| **Data-driven** | LoRA-GA | 11.7 | 482.67 | 2.4 |
| | EVA$_{bs=16}$ | 3.3 | 482.67 | 0.7 |
| | EVA$_{bs=8}$ | 1.38 | 482.67 | 0.3 |
| | EVA$_{bs=4}$ | 1.17 | 482.67 | 0.2 |

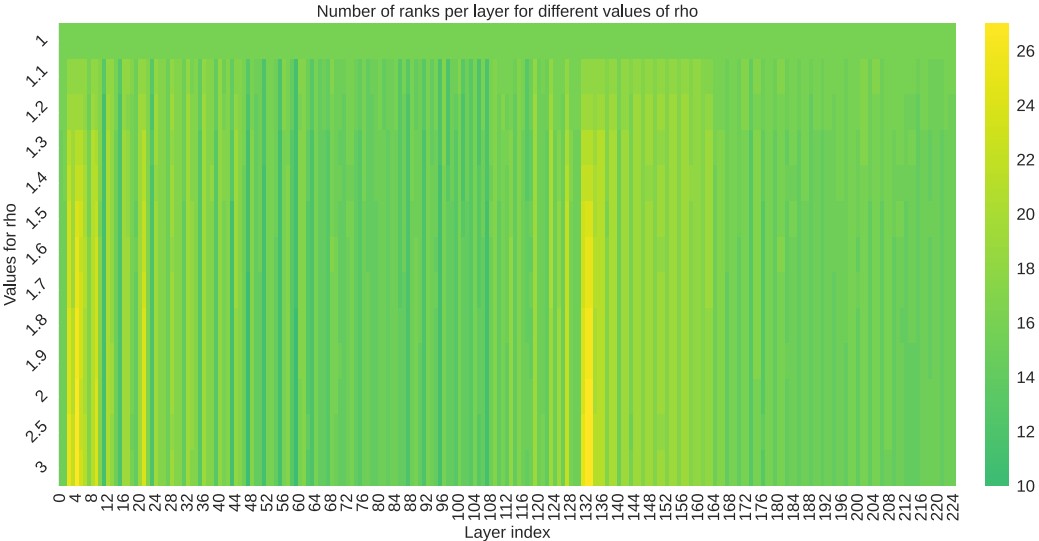

Figure 12: The resulting rank allocation per weight matrix in each layer for Llama-2-7B on the MetaMathQA dataset with different values of $\rho$. The first row represents a uniform distribution where each weight matrix receives the same rank $r = 16$. The most change occurs for $\rho < 1.5$. The re-distribution converges for larger values of $\rho$.

visualizations clearly illustrate that the most change occurs for values of $\rho < 1.5$. Setting $\rho$ to higher values results in less and less change. Interestingly, some ranks still change when going from $\rho = 2.5$ to $\rho = 3$. Finally, we conduct hyperparameter search in which we search over different values of $\rho \in \{1, 1.1, 1.2, 1.3, 1.4, 1.5, 1.6, 1.7, 1.8, 1.9, 2, 2.5, 3\}$. We report the results in Figure 14. We find that for Llama-2-7B on MetaMathQA a uniform distribution performs favorably. The second-best performance is shared by $\rho = 1.5$ and $\rho = 2$. Therefore, we always search for $\rho = 1$ and $\rho = 2$ for all our remaining experiments when we apply EVA and select the best performing one.

## H   RELATION BETWEEN SVD AND PCA

PCA (F.R.S., 1901) is a commonly used tool to decompose a matrix of datasamples $A \in \mathbb{R}^{m \times n}$ into its principal components, i.e. the directions that explain the most variance in the data. The principal components allow projection onto a lower dimensional manifold by preserving the maximal amount

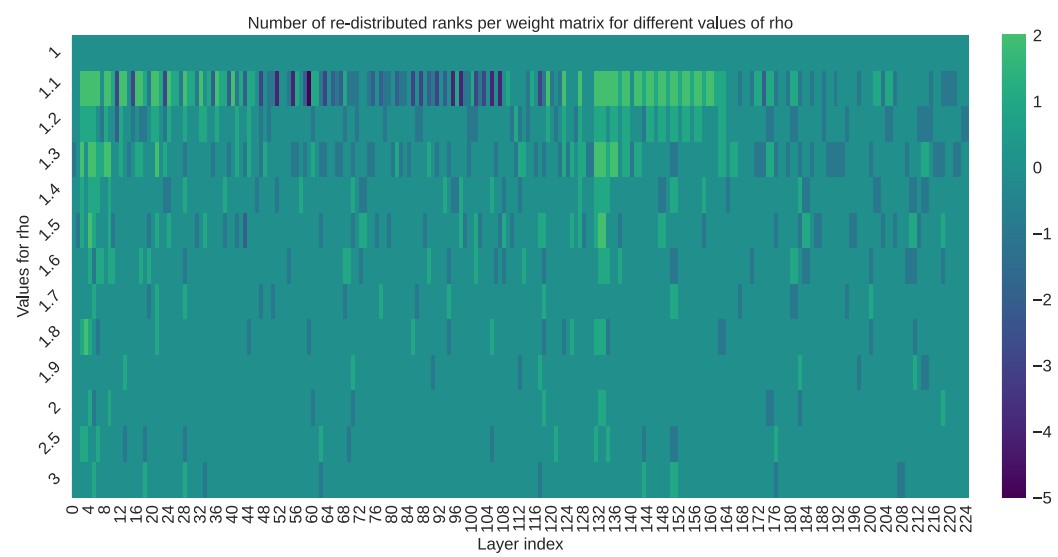

Figure 13: Deltas between rank distributions per weight matrix in each layer for Llama-2-7B on the MetaMathQA dataset with different values of $\rho$. The first row represents a uniform distribution where each weight matrix receives the same rank $r = 16$. The most change occurs in the range $\rho \in [1, 1.5]$. Larger values of $\rho$ do not induce additional significant changes to the rank distribution.

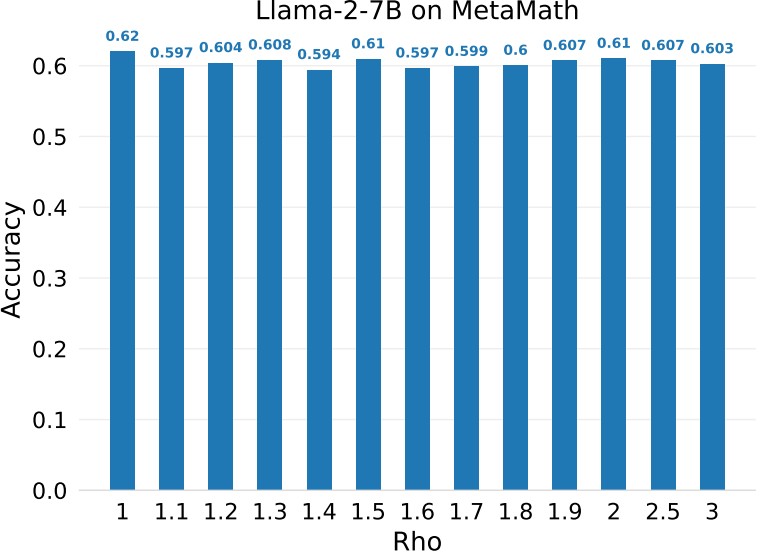

Figure 14: Accuracy for different values of $\rho$ when fine-tuning Llama-2-7B on the MetaMathQA dataset.

of variance. To this end, PCA first computes the sample covariance matrix

$$S = \frac{1}{n-1} A^\top A, \tag{6}$$

where we assume that $A$ is centered. To obtain the *principal directions* of $S$, we perform eigenvalue decomposition as

$$S = V \Lambda V^\top, \tag{7}$$

where $\Lambda = \mathrm{diag}(\lambda_1, \ldots, \lambda_n)$ and eigenvalues are sorted in descending order, i.e. $\lambda_1 \geq \lambda_2 \geq \lambda_n$. The matrix $V \in \mathbb{R}^{n \times n}$ is a matrix of eigenvectors where each column is being referred to as a *principal direction* of $S$. To project $A$ onto a lower dimensional manifold that explains the most variance we can take the top-k principal directions $V_{:,:k}$ and perform $AV$.

PCA is in practice often implemented in the form of SVD as there are efficient approximations thereof (Halko et al., 2011). As mentioned in Equation (1), SVD decomposes the matrix $A$ into

$$A = U \Sigma V^\top, \tag{8}$$

where $U \in \mathbb{R}^{m \times n}$ is a unitary matrix, $\Sigma \in \mathbb{R}^{n \times n}$ is a diagonal matrix of singular values $\Sigma = \mathrm{diag}(\sigma_1, \ldots, \sigma_n)$, and the columns of $V \in \mathbb{R}^{n \times n}$ are called the right singular vectors.

Now we can establish the equivalence between the principal directions obtained by PCA and the right-singular vectors of SVD by substituting $A$ with the right hand side of Equation (8) as

$$S = \frac{1}{n-1} A^\top A = \frac{1}{n-1} V \Sigma U^\top U \Sigma V^\top = V \hat{\Sigma} V^\top. \tag{9}$$

Here, we absorb the factor $\frac{1}{n-1}$ into $\hat{\Sigma}$. Therefore, the right-singular vectors $V$ are the principal directions and $\Sigma U^\top U \Sigma = \Sigma$ as $U^\top U = I$ because $U$ is real.

## I   ABLATION STUDIES

Finally, we conduct ablation studies on EVA to investigate important factors that contribute to its performance. Specifically, we investigate the impact of scale and directions. To this end, we use the VTAB-1K dataset because it comprises a diverse set of tasks and allows for a systematic investigation on in-domain data (natural), and out-of-distribution data (specialized and structured). We report results for our ablation studies in Table 24 and explain the different settings in the following paragraphs.

**Effect of scale.** To investigate the effect of scale on the initialization, we add a setting which uses whitening (EVA-whiten). Whitening scales the initialization by the reciprocal of their eigenvalues, which alters scale, but preserves directions. We found that whitening can significantly improve performance on structured (out-of-distribution) tasks even leading to a slightly higher average score than EVA. This indicates that scale is especially important for structured data. However, EVA-whiten experiences a slight performance drop on natural and specialized tasks.

**Effect of directions.** To address the importance of the directions of the components, we randomly permute its rows (EVA-perm). This preserves scale while corrupting directions and $\ell_2$ norm of $A$. Additionally, we add a setting where we randomly rotate $A$ (EVA-rot), which preserves $\ell_2$ norm, but alters directions. We find that altering directions leads to a performance drop on the structured tasks, while changing $\ell_2$ norm leads to a drop on the natural tasks. Both, EVA-perm and EVA-rot lead to worse average performance across all tasks compared to EVA.

Table 24: Group-wise averages for DINOv2-G/14 ablation studies on the VTAB-1K benchmark.

| Method | Nat. | Spec. | Struct. | All |
|---|---|---|---|---|
| LoRA | 83.2 | **88.8** | 69.0 | 78.4 |
| LoRA-redist | 87.3 | 88.0 | 68.2 | 79.4 |
| EVA-whiten | 87.5 | 87.5 | **69.1** | **79.8** |
| EVA-rot | **87.7** | 88.0 | 68.2 | 79.6 |
| EVA-perm | 87.4 | 87.8 | 68.3 | 79.5 |
| EVA | **87.7** | 87.9 | 68.6 | 79.7 |

**Effect of rank redistribution.** We conduct an experiment in which we randomly initialize $A$ after performing rank redistribution (LoRA-redist). This setting gives insights on the effect of the

redistribution and whether its benefits are bound to EVA. The redistribution has a positive effect on LoRA on the natural tasks, but a negative effect on both structured and specialized tasks. This illustrates that rank redistribution is most beneficial in combination with EVA's initialization of $\boldsymbol{A}$.

Generally, we can say that EVA performs particularly well on natural images and whitening can enhance its performance on out-of-distribution images. The decisive factor with respect to this improvement seems to be a controlled change in the scale of the initialization induced by the singular values. Therefore, by changing the scale in a controlled manner we can make EVA more compatible for different kinds of data. The results for EVA-perm confirm that the scale is the decisive factor for initialization.

