# OpenReview forum: "One Initialization to Rule them All: Fine-tuning via Explained Variance Adaptation"
_ICLR.cc/2025/Conference — Submitted to ICLR 2025_

### Official Review · Reviewer_EQQt · 2024-10-24

**Soundness:** 2
**Presentation:** 2
**Contribution:** 2
**Rating:** 5
**Confidence:** 4

**Summary:**

The paper proposes a novel method, Explained Variance Adaptation (EVA), to improve the fine-tuning process of foundation models. EVA builds upon LoRA (Low-Rank Adaptation) by introducing a data-driven initialization strategy that leverages the singular value decomposition (SVD) of activation vectors during training. This method aims to optimize the initialization of LoRA weights based on the explained variance of activation patterns in the downstream task data. EVA also incorporates adaptive rank allocation, allowing ranks to be distributed across layers in a way that maximizes the variance explained by the model. The paper demonstrates that EVA consistently outperforms existing methods across various tasks, including language generation, image classification, and reinforcement learning, by improving convergence speed and overall performance.

**Strengths:**

1. The proposed pipeline is easy to understand.
2. Comprehensive experiments are done, including language generation and understanding, image classification, and reinforcement learning.
3. Additional computational burden is clearly stated in the paper.

**Weaknesses:**

1. The paper's novelty is limited, as it applies SVD for LoRA initialization and rank distribution—approaches already explored in previous works. Specifically, SVD has been used for initialization in PiSSA [1] and for rank distribution in AdaLoRA [2]. The only incremental contribution appears to be applying SVD to activations, a technique previously examined in the pruning [3] and quantization [4] literature.

2. I question the motivation for initializing matrix A with right-singular vectors, despite the authors' claim that they capture the directions of variance. More analytical evidence is needed to substantiate this conjecture (i.e., why the direction of variance results in better fine-tuning performance?).

[1] Meng, Fanxu, Zhaohui Wang, and Muhan Zhang. "Pissa: Principal singular values and singular vectors adaptation of large language models." arXiv preprint arXiv:2404.02948 (2024).

[2] Zhang, Qingru, et al. "AdaLoRA: Adaptive budget allocation for parameter-efficient fine-tuning." arXiv preprint arXiv:2303.10512 (2023).

[3] Sun, Mingjie, et al. "A simple and effective pruning approach for large language models." arXiv preprint arXiv:2306.11695 (2023).

[4] Lin, Ji, et al. "AWQ: Activation-aware Weight Quantization for On-Device LLM Compression and Acceleration." Proceedings of Machine Learning and Systems 6 (2024): 87-100.

**Questions:**

Please refer to the weakness part.

---

> ### Author Response · Authors · 2024-11-16
>
> Thank you for your feedback. We address the raised weaknesses as follows.
>
> **Novelty:**
>
> We respectfully disagree with the reviewer on the novelty aspect. Incremental SVD has not been used on activations before which brings its own sets of problems. PiSSA [1] applies SVD to pre-trained weights and AdaLoRA [2] does not use SVD, but merely SVD-like re-formulation of LoRA, which is not the same. Furthermore, **neither [3] nor [4] actually apply SVD to activations**. [3] applies a dot product between activations and weights to obtain importance scores and [4] relies on scaling salient activation regions. In fact, neither of [3,4] have a single mention of SVD in their work.
>
> Applying SVD to inputs to weight matrices (we refer to them as activation vectors for simplicity) requires dealing with growing memory as the data matrix increases with each batch, therefore we employ incremental SVD to keep memory costs constant. We clarified this in our updated method section and also explicitly mention our contributions which are as follows:
> We propose a novel data-driven initialization scheme for LoRA by leveraging incremental SVD on minibatches of activation vectors.
> We propose a data-driven heuristic for adaptive rank allocation based on explained variance.
> We demonstrate EVA's effectiveness across a variety of different domains and tasks.
>
> We also added this list of contributions in our introduction in the revised version
>
> **Right-singular vectors capture directions of most variance:**
>
> We provide a proof in Appendix H which demonstrates that the right-singular values obtained via SVD are equivalent to the projection onto the principal components which is usually obtained via PCA. In fact, practical implementations of PCA (pytorch, scikit-learn, etc.) usually employ SVD under the hood to obtain the projection onto the principal components.
> The intuition for leveraging this projection as initialization for the matrix A in LoRA is that the projection onto the principal components capture the most variance, i.e. they preserve the most information of the downstream task in a linear lower dimensional subspace. By redistributing the ranks we can maximize the amount of variance that is explained throughout the model. This leads to improved performance while even reducing the amount of trainable parameters (see Table 11 in Appendix B3). For convenience we also added this table below.
>
> | Model | Method |  #Trainable | Common sense reasoning | GSM8K | MATH  |
> | --- | --- | --- | --- | --- | --- |
> | LLama-2-7B | LoRA | 18.3M | 82.2 | 59.7 | 10.9 |
> | LLama-2-7B| EVA | **17.3M** | **83.4** | **61.9** | **13.1** |
> | LLama-3.1-8B | LoRA | 20M | 89.2 | 78.3 | 30.1 |
> | LLama-3.1-8B | EVA | **18.9M** | **89.5** | **78.8** | **31.2** |
> | Gemma-2-9B | LoRA | 24.5M | 92.2 | 83.4 | 40.7 |
> | Gemma-2-9B | EVA | **23.1M** | **92.5** | **83.6** | **41.5** |
>
>
> **References:**
>
> [1] Pissa: Principal singular values and singular vectors adaptation of large language models., Meng, et al. NeurIPS 2024
>
> [2] AdaLoRA: Adaptive Budget Allocation for Parameter-Efficient Fine-Tuning, Zhang et al., ICLR 2023
>
> [3] A simple and effective pruning approach for large language models., Sun et al., arXiv preprint arXiv:2306.11695 (2023).
>
> [4] AWQ: Activation-aware Weight Quantization for On-Device LLM Compression and Acceleration., Lin et al., PMLR 2024

---

> > ### Comment · Reviewer_EQQt · 2024-11-23
> >
> > Thank you for your response.
> >
> > Firstly, I apologize for any confusion caused earlier. To clarify, I did not mean to imply that [3] or [4] implemented SVD with activations. Rather, my understanding is that this paper borrows the idea from them to **consider activations** when initializing the parameters of LoRA. While the use of SVD itself is not new (as demonstrated in [1] and [2]), the incorporation of activations in this context might be an extension. However, I believe this approach does not offer substantial novelty.
> >
> > Secondly, after reviewing Appendix H, I still have reservations about the rationale behind initializing $A$ using **directions that capture the most variance of the data**. I would like the authors to elaborate further on why focusing on **directions of maximum variance** leads to **better optimization** or **improved downstream performance**. Understanding the underlying logic here would help clarify the benefits of this initialization approach.

---

> ### Author Response · Authors · 2024-11-22
>
> Dear reviewer EQQt,
>
> We thank you again for taking the time and effort to help improve our paper and believe we have addressed all your concerns by clarifying our contributions, improving clarity and adding supplementary proofs and more mathematical formulations.
>
> We would be grateful for an opportunity to discuss wether there are any pending concerns you can point us to.
>
> Thank you, Authors

---

> ### Author Response · Authors · 2024-11-23
>
> Thank you for taking the time to clarify your concerns.
>
> **Novelty:**
>
> Firstly, we would like to emphasize that the novel idea of our work is to make initialization data-driven. This has not been considered in prior work. To achieve such an initialization, we perform incremental SVD on activations. **Using SVD as a tool on activations is just the means to achieve the data-driven initialization and does not diminish the idea of data-driven initialization.** It is not an extension to [1] as the core idea of using the downstream data is fundamentally different to using information from the pre-trained weights. We would also like to stress again that **[2] does not use SVD**. Finally, we do not just use SVD, but incremental SVD, which brings its own set of challenges, such as dealing with incremental updates and distributing them across the entire large model in an efficient manner.
> To clarify, our contributions are as follows:
> - We propose a novel data-driven initialization scheme for LoRA by leveraging incremental SVD on minibatches of activation vectors.
> - We propose a data-driven heuristic for adaptive rank allocation based on explained variance.
> - We demonstrate the effectiveness of EVA across a variety of different domains.
>
> **Rationale behind EVA:**
>
> Depending on the downstream data, each layer’s activation will exhibit different patterns as they are sensitive to the task and triggered to highlight different aspects of the pre-trained weight. Therefore, the activations capture task context. Our aim is to leverage this context for adapting the FM. In fact, we hypothesize that the more information we can leverage from this context, the better the downstream performance, i.e. we require a projection that propagates the maximum amount of information, which can be obtained by a projection onto the principal components, i.e. via SVD. We verified empirically in our experiments that EVA consistently leads to lower loss and improved performance. We are currently also running another experiment where we use the components that capture the least amount of variance and will post the performances as soon as they are finished. This experiment should verify that the directions capturing the most variance should be used for initialization.
>
>
> **References:**
>
> [1] Pissa: Principal singular values and singular vectors adaptation of large language models., Meng, et al. NeurIPS 2024
>
> [2] AdaLoRA: Adaptive Budget Allocation for Parameter-Efficient Fine-Tuning, Zhang et al., ICLR 2023

---

> > ### Author Response · Authors · 2024-11-25
> >
> > Dear reviewer EQQt,
> >
> > To verify our hypothesis that $A$ should be initialized with a projection that explains the most variance in the data, we conducted an experiment where we compare the standard EVA with EVA-minor, which leverages the minor components that explain the least variance. We report results in Table 12 in Appendix B3 in the updated manuscript as well as below for convenienve. EVA-minor performs significantly worse on 5/8 common sense reasoning tasks, while performance on the remaining three is on-par. This provides compelling evidence for initializing according to the projection onto the principal components that explain the most variance. Finally, we would like to stress that the SVD computation for EVA-minor is impractical as it takes a few hours compared to a few seconds for EVA. This is because we need to compute all components and not just the first few as we do for EVA.
> >
> > | Method | BoolQ | PIQA | SIQA | HellaSwag | Winogrande | ARC-e | ARC-c | OBQA | Avg. |
> > | --- | --- | --- | --- | --- | --- | --- | --- | --- | --- |
> > | EVA | 68.6 | 85.0 | 81.2 | 94.2 | 84.7 | 87.4 | 73.5 | 84.1 | 82.3  |
> > | EVA-minor |  64.0 | 83.4 | 81.5 |  94.3 | 82.0 | 87.3 | 73.0 | 81.6  | 80.9  |

---

> > > ### Comment · Reviewer_EQQt · 2024-11-28
> > >
> > > Thank you for your response, which partially addressed my concerns. I have raised my score to 5 as a result. However, the paper shows only marginal improvement and lack strong motivation, so I am still unable to recommend acceptance of this paper.

---

> ### Author Response · Authors · 2024-11-28
>
> Thank you for your further engagement and the increase in score.
>
> We have now further clarified the motivation in a more formal manner and related the effect of our initilization to the fine-tuning performance in the revised version. Let $X = U \Sigma V^\top$ be the SVD decomposition on activations $X$. Then initializing $A = V_{:r}$ and $B=V_{:r}^\top $ minimizes the reconstruction error $\lVert X - XV_{:r}V_{:r}^\top \rVert $. This fact is given by the Eckart-young theorem [1] as $U_{:r} \Sigma_{:r} V_{:r}^\top$ is the best reconstruction of $X$. Now by choosing $A=V_{:r}$ the downprojection $XA$ must contain the most information about $X$ according to the data processing inequality [2], as the maximum amount of information $B$ can contribute is $B=V_{:r}^\top$. Now the gradient w.r.t. $B$ is $\frac{\partial \mathcal{L}}{ \partial B} = \frac{\partial \mathcal{L}}{\partial W}A^\top$. The fine-tuning process is concerned with storing information about the data in the weights. By choosing $A = V_{:r}$ we guarantee that the maximum amount of information is available at the beginning of training, such that it only needs to be learned what information to keep, i.e. what parts of $XA$ are relevant for the downstream task.
>
> [1] The approximation of one matrix by another of lower rank. Eckart et al., Psychometrika 1936
>
> [2] An intuitive proof of the data processing inequality, Beaudry et al., Quantum Information & Computation 2012
>
> Finally, we would like to recite our presented table from our previous answer, where it is clearly visible that while improvements are marginal for some tasks, EVA also reduces the amount of trainable parameters (see Table 11 in Appendix B3). The reason for this is that more ranks are transferred from the higher dimensional feedforward weights (non-attention weights) to the attention weights. We again present this table below. It clearly shows that EVA is pareto-dominant compared to all competitors.
>
> | Model | Method |  #Trainable | Common sense reasoning | GSM8K | MATH  |
> | --- | --- | --- | --- | --- | --- |
> | LLama-2-7B | LoRA | 18.3M | 82.2 | 59.7 | 10.9 |
> | LLama-2-7B| EVA | **17.3M** | **83.4** | **61.9** | **13.1** |
> | LLama-3.1-8B | LoRA | 20M | 89.2 | 78.3 | 30.1 |
> | LLama-3.1-8B | EVA | **18.9M** | **89.5** | **78.8** | **31.2** |
> | Gemma-2-9B | LoRA | 24.5M | 92.2 | 83.4 | 40.7 |
> | Gemma-2-9B | EVA | **23.1M** | **92.5** | **83.6** | **41.5** |

---

> > ### Author Response · Authors · 2024-11-30
> >
> > Dear reviewer EQQt,
> >
> > As there were previously doubts on the effect of initialization, we now also provide empirical evidence complementary to the references we provided. Those verify that the effect of initialization is substantial. To this end, we measured the distance between learned adapters with EVA and LoRA via cosine similarity and spectral norm. The average distances (cosine/spectral) across three seeds for the different weight matrices in Llama-2 and LLama-3.1-8B are shown below. This result verifies that given a different initalization, the adapters converge to substantially different solutions as there is absolutely no overlap in cosine similarity and a large deviaton in the spectral norm.
> >
> > | Model | Query | Key | Value | Gate | Up | Down |
> > | --- | --- | --- | --- | --- | --- | --- |
> > | LLama-2 | -0.01/4.98 | 0.00/5.00 | 0.01/4.00 | 0.00/4.05 | 0.00/6.64 | -0.00/3.67 | -0.00/ 4.02|
> > | LLama-3.1-8B | -0.00/4.05 | -0.01/5.25 | -0.00/3.83 | -0.01/3.53 | -0.00/6.98 | 0.01/3.37 | -0.00/3.73 |
> >
> > Moreover, to verify that EVA provides more information at the beginning of training, we quantify how much has been learned by measuring the distance between the initial adapters and the final ones after training LoRA and EVA for Llama-2 and Llama-3.1. We again average across three seeds and show distances (cosine/spectral) below. As evident below, EVA's initialization is consistently closer to the final adapters after training than LoRA's. This demonstrates that less information is learned for EVA as it already captures the most possible amount of information in the subspace. We will add both tables in the final version in case of acceptance.
> >
> > | Method | Model | Query | Key | Value | Gate | Up | Down |
> > | --- | --- | --- | --- | --- | --- | --- | --- |
> > | LoRA | LLama-2 |0.51/3.85 | 0.48/4.08 | 0.60/3.10 | 0.59/3.09 | 0.44/5.27 | 0.62/2.83 | 0.61/ 3.13 |
> > | EVA | Llama-2 | **0.62/3.48** | **0.59/3.59** | **0.62/2.90** | **0.62/2.78** | **0.42/4.92** | **0.66/2.61** | **0.67/2.84** |
> > | LoRA | Llama-3.1-8B | 0.51/3.46 | 0.47/3.96 | 0.59/2.93 | 0.61/2.73 | 0.35/5.88 | 0.60/2.58 | 0.59/2.98 |
> > | EVA | LLama-3.1-8B | **0.64/2.93** | **0.61/3.62** | **0.63/2.46** | **0.64/2.27** | **0.41/5.12** | **0.67/2.46** | **0.67/2.71** |

---

> > > ### Author Response · Authors · 2024-12-02
> > >
> > > Dear reviewer EQQt,
> > >
> > > We thank you again for taking the time and effort to help improve our paper.
> > >
> > > Since we are at the end of the extended author-reviewer discussion period, we are reaching out to ask if our response have addressed your remaining concerns.
> > >
> > > Please let us know if you have lingering questions and whether we can provide any additional clarifications today to improve your rating of our paper.
> > >
> > > Thank you,
> > > Authors

---

### Official Review · Reviewer_iw7g · 2024-10-31

**Soundness:** 2
**Presentation:** 3
**Contribution:** 2
**Rating:** 5
**Confidence:** 4

**Summary:**

This paper focuses on the parameter-efficient fine-tuning of foundation models (FMs). Based on the commonly used low-rank adaptation (LoRA), the authors propose a new method called Explained Variance Adaptation (EVA) to fine-tuning FMs faster and better. Specifically, they emphasize the data-driven initialization and the adaptive ranks within EVA. The former is implemented via computing singular value decomposition on minibatches of activation vectors. The latter is based on explained variance. Extensive empirical results across various fine-tuning tasks have shown the effectiveness of their approaches.

**Strengths:**

1.The experiments in this paper are comprehensive, spanning multiple fields and validating the generalizability of the method sufficiently.

2.The method in this paper seems to be straightforward and easy to reproduce.

3.The data-driven manner proposed in this paper appears to be promising.

**Weaknesses:**

1.The clarity of the figures and formulas need further improvement. Figure 1 seems rough, failing to clearly illustrate the method presented in this paper. The authors can illustrate the following details in Figure 1: constructing matrix A from right singular vectors obtained through SVD decomposition; the incremental update process; and the procedure of adaptive rank allocation. Additionally, the mathematical formulation of "incrementally update" in sec 3.2 is necessary.

2.The "faster convergence" claimed by the authors in the Abstract and Figure 2 seems not sufficiently significant. Adding experiments with other models and tasks could enhance the persuasiveness, not only Llama-3.1-8B on the MetaMathQA dataset. Similar to validating the method's effectiveness on language generation, language understanding, image classification, and decision making, the author can evaluate the "faster convergence" by selecting one dataset from these tasks.
Additionally, regarding the relationship between gradient norm and convergence speed, more references should be included.

3.This paper provides extensive experiments but lacks sufficient theoretical analysis or intuitive explanations. What makes this work is more interesting to the community. A thorough analytical discussion and interpretation of the initialization technique presented in Section 3.2 is needed, along with relevant supporting references when appropriate. This will be further discussed in “Questions”.

**Questions:**

1.The proposed method is named “Explained Variance Adaptation” that also be included in the title. However, what is the relationship between "Data-driven Initialization" in sec 3.2 and “Explained Variance”? The key point of the proposed method seems to be about mining and utilizing downstream data information. “Explained Variance” appears to be just one part of it.
Furthermore, the authors should provide more detailed explanations for the operations in sec 3.2 - How does using right singular vectors for initialization affect fine-tuning? Why does it lead to better performance?

2.The author points out the performance of different variants of EVA on various types of data (natural, specialized, and structured) in Table 7. However, there is a lack of in-depth analysis. For example, what makes the differences in EVA's performance across natural, specialized, and structured data? Could an appropriate explanation for this result be provided based on the analysis about sec 3.2?

---

> ### Author Response · Authors · 2024-11-16
>
> We would like to thank the reviewer for their constructive feedback and address the raised weaknesses and questions as follows.
>
> **Presentation:**
>
> We agree that Figure 1 did not accurately reflect the procedure conducted by EVA and would like to thank the reviewer for bringing this to our attention. We updated Figure 1 in the revised manuscript. We also added the mathematical formulation of the incremental update in Section 3.2 along with pseudocode in Algorithm 2 and references to proofs plus a supporting proof in Appendix H that the components obtained via incremental SVD are projections onto the principal components that explain the most amount of variance in the linear subspace spanned by EVA. The changes are highlighted in red color in the revised version. If the reviewer has any further suggestions to enhance clarity, we will gladly update our manuscript.
>
> **Training Convergence:**
>
> Thank you for the suggestion to add more training curves to our manuscript. We added additional loss curves for Gemma-2, Llama-3.1, and Llama-2 in Figure 4 in the Appendix. They verify the results shown in Figure 2. We will add further loss curves for the different domains as well, but need to re-train some models for that. In Figure 2 on the right, we show the gradient norm at the very first logging step, which demonstrates that EVA receives the strongest gradient signal at the beginning of training.
>
> **Naming of the method:**
>
> The reason why we name our method “Explained Variance Adaptation” is because the projection onto the linear subspace which we use to initialize the matrix A explains the most variance of the input to each weight matrix. This is supported by the proof in Appendix H that we added which shows that the right singular values after SVD are a projection onto the principal components, i.e. they explain the most variance. Further, by adaptively allocating ranks we can maximize the explained variance throughout the model. The reason why this results in better performance is that EVA projects the input to a weight matrix into a lower dimensional subspace while preserving the most amount of information possible. Therefore, EVA propagates more information of the downstream data through the model compared to random or a weight-driven initialization.
>
> **Ablation Studies:**
>
> This is indeed an interesting question and the motivation to perform these experiments on the VTAB-1K dataset was to gain a better intuition on what factors are crucial for performance on in vs out-of-distribution data. We found that by varying the scale and the directions of our initialization there is variation in the downstream behavior. For example, using whitening (changing the scale) leads to improvements on structured images. Therefore, this setting may be especially useful when dealing with out-of-distribution data. Further, EVA is particularly effective on natural images. Therefore, by changing the scale in a controlled manner we can make EVA more suitable for out of distribution data. Furthermore, varying the directions of the projection matrices deteriorates performance on out-of-distribution data. Therefore, we can conclude that the combination of both scale and direction of the projection matrix are crucial for good performance of EVA. We moved this section to appendix to accommodate a more in-depth description of EVA in Section 3, however still added another paragraph to make this more explicit.

---

> > ### Comment · Reviewer_iw7g · 2024-11-23
> >
> > Thank you for your response and the revisions to the paper.
> >
> > Regarding my Question 1, this has raised consistent concerns among all four reviewers. The authors claim that has been addressed in Appendix H. However, I agree with reviewer EQQt that the proof fails to provide the underlying rationale for how the proposed initialization method leads to better downstream performance.
> > Moreover, I remain unconvinced by the evidence presented for “fast convergence”. After reading reviewer Qji4's comments, I believe this may be related to Qji4's Question 2, which suggests that different initialization methods have relatively minor impacts.
> >
> > With these considerations, I decide to keep my original rating.

---

> ### Author Response · Authors · 2024-11-22
>
> Dear reviewer iw7g,
>
> We thank you again for taking the time and effort to help improve our paper and believe we have addressed all your concerns by properly motivating EVA, updating Figure 1 and extending the methodology section, supplementary proofs and additional pseudocode, adding additional loss curves, and clearing up any confusion about the ablation studies.
>
> We would be grateful for an opportunity to discuss wether there are any pending concerns you can point us to.
>
> Thank you, Authors

---

> ### Author Response · Authors · 2024-11-23
>
> Thank you for taking the time to respond to our rebuttal!
>
> To clarify, we did not claim that Appendix H addresses any rationale about EVA, but it addresses concerns raised by reviewer EQQt that the claim about explained variance is unsupported - it is not, it is proven in Appendix H. We explicitly give a more intuitive rationale on EVA below.
>
> **Rationale behind EVA:**
>
> Depending on the downstream data, each layer’s activation will exhibit different patterns as they are sensitive to the task and triggered to highlight different aspects of the pre-trained weight. Therefore, the activations capture task context. Our aim is to leverage this context for adapting the FM. In fact, we hypothesize that the more information we can leverage from this context, the better the downstream performance, i.e. we require a projection that propagates the maximum amount of information, which can be obtained by a projection onto the principal components via SVD. We verified empirically in our experiments that EVA consistently leads to lower loss and improved performance.
>
> **Faster convergence:**
>
> We provided evidence for faster convergence for three different models on three different downstream tasks.
> Can the reviewer elaborate on how many more loss curves it takes to be convincing?
>
> **Effect on initialization:**
>
> There is plenty of evidence in the literature that initialization of LoRA significantly impacts downstream performance, may it be of empirical [1,2,3] or analytical nature [4]. Even the original work on LoRA [5] conducted pre-training on different tasks as initialization for certain other downstream tasks as it improved performance. Can the reviewer provide any references that provide evidence that initialization methods have "minor impact"?
>
> **References:**
>
> [1] Pissa: Principal singular values and singular vectors adaptation of large language models., Meng, et al. NeurIPS 2024
>
> [2] LoRA-GA: Low-Rank Adaptation with Gradient Approximation, Wang et al., NeurIPS 2024
>
> [3] OLoRA: Orthonormal Low-Rank Adaptation of Large Language Models, Büyükakyüz et al., arXiv:2406.01775
>
> [4] The Impact of Initialization on LoRA Finetuning Dynamics, Hayou et al., NeurIPS 2024
>
> [5] LoRA: Low-Rank Adaptation of Large Language Models, Hu et al., ICLR 2022

---

> > ### Comment · Reviewer_iw7g · 2024-11-26
> >
> > **Faster convergence:**
> >
> > Thank you for the additional loss curves (Appendix Fig. 4) regarding "faster convergence" in the new version. I've noted these while writing my previous response. I would like to clarify that my unconvinced stance does not stem from any concern about the quantity of results. On the contrary, multiple current results suggest that the random perturbations in loss appear to be larger than the "faster convergence" of "EVA" compared to "Random" that the authors aim to demonstrate (For a good example of "faster convergence", please refer to the results in Fig 1 of LoRA-GA [1]). In my assessment, the subtle "fast convergence" shown in the curves is potentially consistent with the modest performance gains mentioned in reviewer Qji4's Question 2, both observations suggesting that the proposed initialization method fails to demonstrate substantial impact.
> >
> > Furthermore, the authors state in Figure 2 that "EVA exhibits significantly larger gradient norm leading to faster convergence". It would be helpful if the authors could provide some references to support this causal relationship. I notice a contradicting example within Figure 2 itself, where "PiSSA" displays a larger gradient norm (as shown in Figure 2 Right) yet fails to demonstrate faster convergence compared to "Random", as evidenced by the loss curve.
> >
> > **Rationale behind EVA:**
> >
> > I agree that "data-driven" initialization potentially offers better adaptability to different downstream tasks than random initialization. The proposed EVA represents one implementation for this idea. However, current rationale behind EVA remains insufficient. I share the same concerns as reviewer EQQt regarding "why focusing on directions of maximum variance leads to better optimization or improved downstream performance". Could the authors further validate their viewpoint through mathematical formulations or analytical experiments? Additionally, Based on the current explanation of EVA, how to explain the poor performance of EVA on Spec. and Struct. (previous Table 7 in the main paper) ? The authors are encouraged to offer deeper insights about these results instead of simply describing the observed phenomena.
> >
> >
> >
> > [1] LoRA-GA: Low-Rank Adaptation with Gradient Approximation, Wang et al., NeurIPS 2024

---

> > > ### Author Response · Authors · 2024-11-26
> > >
> > > Thank you for your further engagement in this discussion, we greatly appreciate it!
> > >
> > > **Faster convergence:**
> > >
> > > Thank you for pointing out the inconsistency around the relationship between gradient norm and loss. Indeed, we admit that this was an overclaim on our side and we now removed every claim that it is the gradient norm that results in faster convergence. Figure 1 in [1] is a single loss curve for a single seed which does not properly account for random effects. In contrast, our loss curves are averaged across multiple seeds. Averaging across more than one seed is a very important in our opinion to determine whether the faster convergence holds and is not due to some random factor. This is indeed the case for EVA as we demonstrated in multiple loss curves.
> > >
> > > **Performance of EVA:**
> > >
> > > Thank you for raising this point, we added an additional paragraph in the discussion section that addresses this point. We would like to clarify a few things in this regard.
> > >
> > > > Based on the current explanation of EVA, how to explain the poor performance of EVA on Spec. and Struct. (previous Table 7 in the main paper) ?
> > >
> > > First of all, we never claimed that EVA is better on each and every task, our claims are always with respect to average performance. Neither our motivation, nor our claims suggests that EVA outperforms every other method on every task.
> > >
> > > > how to explain the poor performance of EVA on Spec. and Struct. ?
> > >
> > > The claim that results are poor is very subjective and the higher average accuracy of EVA compared to all other methods highlights that other methods perform worse on different tasks, leading to a better average score of EVA. We believe the fact that we observed these results is simply because there is no free lunch, i.e. there is no one algorithm that performs the best across all tasks and we believe that the community should be made aware of this. Our empirical evidence verifies this observation. Finally, EVA is pareto-dominant compared to all competitors as it decreases the number of trainable parameters via rank redistribution, while attaining better average scores (evidenced in Table 11 in Appendix B3).
> > >
> > > > why focusing on directions of maximum variance leads to better optimization or improved downstream performance?
> > >
> > > Let $X = U \Sigma V^\top$ be the SVD decomposition on activations $X$. Then initializing $A = V_{:r}$ and $B=V_{:r}^\top $ minimizes the reconstruction error $\lVert X - XV_{:r}V_{:r}^\top \rVert $. This fact is given by the Eckart-young theorem [2] as $U_{:r} \Sigma_{:r} V_{:r}^\top$ is the best reconstruction of $X$. Now by choosing $A=V_{:r}$ the downprojection $XA$ must contain the most information about $X$ according to the data processing inequality [3], as the maximum amount of information $B$ can contribute is $B=V_{:r}^\top$. Now the gradient w.r.t. $B$ is $\frac{\partial \mathcal{L}}{ \partial B} = \frac{\partial \mathcal{L}}{\partial W}A^\top$. The fine-tuning process is concerned with storing information about the data in the weights. By choosing $A = V_{:r}$ we guarantee that the maximum amount of information is available at the beginning of training, such that it only needs to be learned what information to keep, i.e. what parts of $XA$ are relevant for the downstream task.
> > >
> > > To empirically verify that it should be the components that propoagate the most information, we conduct an experiment, where we fine-tune Llama-2 on the common sense reasoning tasks and use the components that propoagate the least amount of information, i.e. $A = V_{-r:}$. We call this method EVA-minor and report results below and in Table 12 in Appendix B3. EVA-minor performs significantly worse on 5/8 common sense reasoning tasks, while performance on the remaining three is on-par. This result further verifies our intuition on the benefits of initialization according to EVA.
> > >
> > > | Method | BoolQ | PIQA | SIQA | HellaSwag | Winogrande | ARC-e | ARC-c | OBQA | Avg. |
> > > | --- | --- | --- | --- | --- | --- | --- | --- | --- | --- |
> > > | EVA | 68.6 | 85.0 | 81.2 | 94.2 | 84.7 | 87.4 | 73.5 | 84.1 | 82.3  |
> > > | EVA-minor |  64.0 | 83.4 | 81.5 |  94.3 | 82.0 | 87.3 | 73.0 | 81.6  | 80.9 |
> > >
> > > **References:**
> > >
> > > [1] LoRA-GA: Low-Rank Adaptation with Gradient Approximation, Wang et al., NeurIPS 2024
> > >
> > > [2] The approximation of one matrix by another of lower rank. Eckart et al., Psychometrika 1936
> > >
> > > [3] An intuitive proof of the data processing inequality, Beaudry et al., Quantum Information & Computation 2012

---

> > > > ### Author Response · Authors · 2024-11-30
> > > >
> > > > Dear reviewer iw7g,
> > > >
> > > > As there were previously doubts on the effect of initialization, we now also provide empirical evidence complementary to the references we provided. Those verify that the effect of initialization is substantial. To this end, we measured the distance between learned adapters with EVA and LoRA via cosine similarity and spectral norm. The average distances (cosine/spectral) across three seeds for the different weight matrices in Llama-2 and LLama-3.1-8B are shown below. This result verifies that given a different initalization, the adapters converge to substantially different solutions as there is absolutely no overlap in cosine similarity and a large deviaton in the spectral norm.
> > > >
> > > > | Model | Query | Key | Value | Gate | Up | Down |
> > > > | --- | --- | --- | --- | --- | --- | --- |
> > > > | LLama-2 | -0.01/4.98 | 0.00/5.00 | 0.01/4.00 | 0.00/4.05 | 0.00/6.64 | -0.00/3.67 | -0.00/ 4.02|
> > > > | LLama-3.1-8B | -0.00/4.05 | -0.01/5.25 | -0.00/3.83 | -0.01/3.53 | -0.00/6.98 | 0.01/3.37 | -0.00/3.73 |
> > > >
> > > > Moreover, to verify that EVA provides more information at the beginning of training, we quantify how much has been learned by measuring the distance between the initial adapters and the final ones after training LoRA and EVA for Llama-2 and Llama-3.1. We again average across three seeds and show distances (cosine/spectral) below. As evident below, EVA's initialization is consistently closer to the final adapters after training than LoRA's. This demonstrates that less information is learned for EVA as it already captures the most possible amount of information in the subspace. We will add both tables in the final version in case of acceptance.
> > > >
> > > > | Method | Model | Query | Key | Value | Gate | Up | Down |
> > > > | --- | --- | --- | --- | --- | --- | --- | --- |
> > > > | LoRA | LLama-2 |0.51/3.85 | 0.48/4.08 | 0.60/3.10 | 0.59/3.09 | 0.44/5.27 | 0.62/2.83 | 0.61/ 3.13 |
> > > > | EVA | Llama-2 | **0.62/3.48** | **0.59/3.59** | **0.62/2.90** | **0.62/2.78** | **0.42/4.92** | **0.66/2.61** | **0.67/2.84** |
> > > > | LoRA | Llama-3.1-8B | 0.51/3.46 | 0.47/3.96 | 0.59/2.93 | 0.61/2.73 | 0.35/5.88 | 0.60/2.58 | 0.59/2.98 |
> > > > | EVA | LLama-3.1-8B | **0.64/2.93** | **0.61/3.62** | **0.63/2.46** | **0.64/2.27** | **0.41/5.12** | **0.67/2.46** | **0.67/2.71** |

---

> > > > > ### Author Response · Authors · 2024-12-02
> > > > >
> > > > > Dear reviewer iw7g,
> > > > >
> > > > > We thank you again for taking the time and effort to help improve our paper.
> > > > >
> > > > > Since we are at the end of the extended author-reviewer discussion period, we are reaching out to ask if our response have addressed your remaining concerns.
> > > > >
> > > > > Please let us know if you have lingering questions and whether we can provide any additional clarifications today to improve your rating of our paper.
> > > > >
> > > > > Thank you,
> > > > > Authors

---

### Official Review · Reviewer_nwRB · 2024-11-02

**Soundness:** 2
**Presentation:** 2
**Contribution:** 2
**Rating:** 6
**Confidence:** 3

**Summary:**

This paper presents a novel initialization method called EVA for Low-Rank Adaptation (LoRA), which combines a weight-driven SVD with rank redistribution. The authors initialize matrix A by applying SVD to the input data X and then utilize a pruning method to implement rank redistribution. They further demonstrate the effectiveness of the EVA method through experiments on a variety of downstream tasks.

**Strengths:**

1.	This paper is well-written.
2.	The authors evaluate the method from multiple aspects, including language understanding, language generation, and image classification.

**Weaknesses:**

1.	Although the authors validate the effectiveness of EVA in various aspects, they do not provide insights into the reasoning behind their method. Specifically, they explain the process of initializing via SVD decomposition of activation values but fail to elaborate on the underlying principles. I would like the authors to clarify the rationale behind the EVA method.
2.	Using rank redistribution within the same budget may lead to unfair comparisons; the effects of increasing the rank in small matrices versus large matrices differ significantly. For instance, adjusting (16, 512) to (17, 512) versus (16, 4096) to (17, 4096) increases the rank by one, but the latter clearly introduces more parameters. Additionally, the authors perform hyperparameter searches for $\rho$ at {1, 2}, and results with $\rho=1$ (uniform rank) sometimes yield better outcomes, suggesting that rank redistribution may not be as effective.
3.	There is a lack of comparisons with related works, such as rsLoRA[1], LoRA+[2], and LoRA-GA[3] (also a data-driven initialization).
4. The experimental setting of $\alpha=1$ seems unusual; for a rank of 16, a more common setting would be $\alpha=32$. A specific comparison of their results would be beneficial.

[1] Kalajdzievski, Damjan. "A rank stabilization scaling factor for fine-tuning with lora.", arXiv preprint, 2023.

[2] Hayou, Soufiane, et al. "Lora+: Efficient low rank adaptation of large models." arXiv preprint, 2024.

[3] Wang, Shaowen, et al. "LoRA-GA: Low-Rank Adaptation with Gradient Approximation." NeurIPS, 2024.

**Questions:**

See Weaknesses.

---

> ### Author Response · Authors · 2024-11-16
>
> We would like to thank the reviewer for the time spent to provide constructive feedback that helps us significantly improve our manuscript. We adress all raised weaknesses as follows.
>
> **Rationale behind EVA:**
>
> Thank you for pointing out the lack of clarity! The rationale behind EVA is that we would like to initialize the matrix A in a way such that the most information of the downstream task is being propagated through the model. The reasoning behind this is that random initialization introduces wasteful update steps especially in the beginning of training [1]. We can remediate this by initializing A with a projection onto the principal components, because those explain the most amount of variance in the data. In our case the data is the input to each weight matrix, which we refer to as activation vectors for simplicity. An efficient way of obtaining this projection is via SVD (we added a proof in Appendix H for equivalence between right-singular vectors and projection onto principal components). A naive way  would be to collect activations for the entire dataset and perform SVD on the stacked matrix. However since the dataset can be arbitrarily large and we want to avoid any overhead in memory we leverage incremental SVD [2]. We added the mathematical formulation of the incremental procedure and references to proofs in Section 3.2 along with pseudocode for the incremental update in Algorithm 2 in the Appendix. Now, to maximize the amount of variance that is explained throughout the model , we perform rank redistribution by sorting and assigning ranks according to their explained variance. To enhance clarity, we also updated Figure 1 such that it comprises all parts of EVA.
>
> **Parameter budget and adaptive ranks:**
>
> Thank you for raising this issue! For $\rho=1$ the parameter count is exactly the same as for LoRA, however this changes for $\rho=2$. We added a table containing the number of trainable parameters and performance for the larger models (Table 11 in Appendix B.3) which demonstrates that **EVA with rank redistribution actually reduces the number of trainable parameters** while improving performance. We added a reduced version of this table below for convenience.
>
> | Model | Method |  #Trainable | Common sense reasoning | GSM8K | MATH  |
> | --- | --- | --- | --- | --- | --- |
> | LLama-2-7B | LoRA | 18.3M | 82.2 | 59.7 | 10.9 |
> | LLama-2-7B| EVA | **17.3M** | **83.4** | **61.9** | **13.1** |
> | LLama-3.1-8B | LoRA | 20M | 89.2 | 78.3 | 30.1 |
> | LLama-3.1-8B | EVA | **18.9M** | **89.5** | **78.8** | **31.2** |
> | Gemma-2-9B | LoRA | 24.5M | 92.2 | 83.4 | 40.7 |
> | Gemma-2-9B | EVA | **23.1M** | **92.5** | **83.6** | **41.5** |
>
> The reason for this is that ranks are often transferred from the higher dimensional feed-forward weights to lower dimensional attention weights. Furthermore, the reviewer is correct in highlighting that adaptive rank allocation is not always the best performing method and this heavily depends on the downstream task, i.e. on common sense reasoning $\rho=2$ performs better and on math tasks $\rho=1$ performs slightly better. Further, for the image classification experiments and the language understanding experiments we used adaptive ranks, while for the RL tasks uniform ranks performed better. We have made this more explicit in our revised version.  We also added an additional paragraph in the discussion section where we elaborate on the effect of rank redistribution in more detail. We would like to stress that there is no one method that performs best across all tasks, i.e. no free lunch, but rank redistribution can improve EVA while simultaneously reducing the parameter count. Therefore we believe it is a valuable contribution.
>
> **References:**
>
> [1] Pissa: Principal singular values and singular vectors adaptation of large language models., Meng, et al. NeurIPS 2024
>
> [2] Incremental learning for robust visual tracking., Ross et al., NeurIPS 2004
>
> See next response for remaining points.

---

> ### Author Response · Authors · 2024-11-16
>
> **Lack of baselines:**
>
> Thank you for making us aware of the concurrent work on LoRA-GA, we incorporated it into our experiments and added results for it in Table 2 and 3. We find that EVA mostly outperforms LoRA-GA, see reduced table below.
>
> | Model | Method |  #Trainable | Common sense reasoning | GSM8K | MATH  |
> | --- | --- | --- | --- | --- | --- |
> | LLama-2-7B | LoRA-GA | 18.3M | **83.4** | 60.2 | 11.7 |
> | LLama-2-7B| EVA | **17.3M** | **83.4** | **61.9** | **13.1** |
> | LLama-3.1-8B | LoRA-GA | 20M | 89.0 | **78.8** | 30.0 |
> | LLama-3.1-8B | EVA | **18.9M** | **89.5** | **78.8** | **31.2** |
> | Gemma-2-9B | LoRA-GA | 24.5M | 91.8 | 82.8 | 40.4 |
> | Gemma-2-9B | EVA | **23.1M** | **92.5** | **83.6** | **41.5** |
>
>
> Furthermore, we analyzed the efficiency of LoRA-GA in Table 22 in Appendix F4 and show that initialization with EVA is approximately 12 times faster for Llama-2-7B on MetaMath. We again show a reduced version of this table below.
>
> | Method    | Initialization Time | Training Time | % of Training |
> | -------- | ------- |----| --- |
> | PiSSA  | 7.43    | 482.67 | 1.5 |
> | OLoRA | 0.3      | 482.67 | 0.1 |
> | LoRA-GA | 11.7 | 482.67 |   2.4 |
> | EVA | 1.17 | 482.67 | 0.2 |
>
> rsLoRA and LoRA+ are orthogonal approaches as they focus on rank-stabilized scaling and learning rate optimization. Suggested modifications by either one of them can be applied to **ALL methods** including EVA. To investigate their effect, we added an experiment where we compare rsLoRA and LoRA+ with EVA under the same conditions to fine-tuning LLama-2-7B on the common sense reasoning tasks. We find that (1) alpha=1 performs better in our experiment setup than alpha=32 (**83.4 vs 82.7 avg perfomance**, respectively), (2) rsLoRA improves LoRA as well as EVA, (3) LoRA+ does not lead to any gains in our setup, (4) EVA consistently outperforms LoRA across all these settings. We provide the table below.
>
> | Adaptation | Method | BoolQ | PIQA | SIQA| HellaSwag | Winogrande | ARC-e | ARC-c | OBQA | Avg |
> | --- | --- | --- | --- | --- | --- | --- | --- | --- | --- | --- |
> | LoRA+ | LoRA | 64.5 | 84.7 | **81.6** | **94.4** | 83.8 | 87.3 | **73.9** | **85.5** | 82.0 |
> | LoRA+ | EVA | **68.6** | **85.0** | 81.2 | 94.2 | **84.7** | **87.4** | 73.5 | 84.1 | **82.3** |
> | rsLoRA | LoRA | 71.5 | 85.3 | 82.5 | 95.2 | 84.5 | 89.0 | 75.8 | **86.8** | 83.8 |
> | rsEVA | EVA | **75.5** | **86.1** | **82.7** | **95.4** | **86.1** | **89.3** | **76.3** | 86.3 | **84.7** |
> | $\alpha=32$ | LoRA | **77.9** | 82.1 | 80.1 | 93.2 | 79.8 | 86.3 | 71.5 | 79.3 | 81.3 |
> | $\alpha=32$ | EVA | 68.6 | **84.9** | **82.2** | **94.6** | **84.1** | **87.8** | **74.7** | **84.4** | **82.7**|
> | $\alpha=1$ | LoRA | 67.2 | 83.9 | 82.0 | 94.7 | 84.0 | 87.8 | 74.1 | 84.0 | 82.2 |
> | $\alpha=1$ | EVA | **68.3** | **85.3** | **82.9** | **95.2** | **85.2** | **88.6** | **75.8** | **86.3** | **83.4** |

---

> ### Author Response · Authors · 2024-11-22
>
> Dear reviewer nwRB,
>
> We thank you again for taking the time and effort to help improve our paper and believe we have addressed all your concerns by properly motivating EVA, demonstrating that EVA actually reduces the number of trainable parameters compared to LoRA,
>  and by providing comparisons to works such as rsLoRA, LoRA+, LoRA-GA and different values for $\alpha$.
>
> We would be grateful for an opportunity to discuss wether there are any pending concerns you can point us to.
>
> Thank you,
> Authors

---

> > ### Author Response · Authors · 2024-12-02
> >
> > Dear reviewer nwRB,
> >
> > We thank you again for taking the time and effort to help improve our paper.
> >
> > Since we are at the end of the extended author-reviewer discussion period, we are reaching out to ask if our response have addressed your remaining concerns.
> >
> > Please let us know if you have lingering questions and whether we can provide any additional clarifications today to improve your rating of our paper.
> >
> > Thank you,
> > Authors

---

> > > ### Comment · Reviewer_nwRB · 2024-12-03
> > >
> > > Thank you for your response and the additional experiments. Like the other reviewers, I still find the rationale behind using SVD of activation for weight initialization unclear. However, I have decided to raise my score to encourage the author to explore this issue further in future work.

---

> > > > ### Author Response · Authors · 2024-12-03
> > > >
> > > > Thank you for engaging in the discussion and for raising your score!
> > > >
> > > > We have now further clarified the motivation in a more formal manner and related the effect of our initilization to the fine-tuning performance in the revised version. Let $X = U \Sigma V^\top$ be the SVD decomposition on activations $X$. Then initializing $A = V_{:r}$ and $B=V_{:r}^\top $ minimizes the reconstruction error $\lVert X - XV_{:r}V_{:r}^\top \rVert $. This fact is given by the Eckart-young theorem [1] as $U_{:r} \Sigma_{:r} V_{:r}^\top$ is the best reconstruction of $X$. Now by choosing $A=V_{:r}$ the downprojection $XA$ must contain the most information about $X$ according to the data processing inequality [2], as the maximum amount of information $B$ can contribute is $B=V_{:r}^\top$. Now the gradient w.r.t. $B$ is $\frac{\partial \mathcal{L}}{ \partial B} = \frac{\partial \mathcal{L}}{\partial W}A^\top$. The fine-tuning process is concerned with storing information about the data in the weights. By choosing $A = V_{:r}$ we guarantee that the maximum amount of information is available at the beginning of training, such that it only needs to be learned what information to keep, i.e. what parts of $XA$ are relevant for the downstream task.
> > > >
> > > > Or put differently, during the fine-tuning process information about the downstream data is stored in the newly introduced LoRA adapters. Our motivation is to initialize these adapters such that they already provably contain the most information of the downstream data. This way, it only needs to be learned what information to maintain or discard. We can obtain such an initialization by applying SVD to the inputs of the weight matrices, as it provides us with the projection onto the principal components, i.e. we perform PCA on activations to obtain our initialization. This initialization provably stores the most information of the downstream data in the adapter weights. We updated our motivation in the uploaded version.
> > > >
> > > > [1] The approximation of one matrix by another of lower rank. Eckart et al., Psychometrika 1936
> > > >
> > > > [2] An intuitive proof of the data processing inequality, Beaudry et al., Quantum Information & Computation 2012

---

### Official Review · Reviewer_Qji4 · 2024-11-02

**Soundness:** 2
**Presentation:** 2
**Contribution:** 2
**Rating:** 3
**Confidence:** 4

**Summary:**

This paper introduces an enhancement to LoRA by initializing new weights through a data-driven approach. It computes singular value decomposition on mini-batches of activation vectors, using the resulting right-singular vectors to initialize the LoRA matrices. Additionally, ranks are re-distributed among weight matrices to maximize explained variance across layers, followed by standard LoRA fine-tuning. This new method, called Explained Variance Adaptation (EVA), is tested across various fine-tuning tasks, including language generation, understanding, image classification, and reinforcement learning. EVA demonstrates faster convergence and achieves the highest average performance across a range of tasks in each domain.

**Strengths:**

1.	The proposed idea is quite straightforward, which I consider a strength rather than a limitation.

**Weaknesses:**

1.	It's unclear why performing SVD on activation vectors for the initial mini-batches is beneficial for initializing the LoRA matrix. This could be explained further.
2.	The visualizations in the paper could be improved for professionalism, for instance, the text in Figure 1 is too large, while in the left part of Figure 2, it is too small.
3.	The improvement achieved by the proposed method is minimal in most of the tables, which raises doubts about its true effectiveness.

**Questions:**

1. Why is performing SVD on activation vectors for the initial mini-batches beneficial for initializing the LoRA matrix? This could be explained further, either from a theoretical or experimental perspective.

2. In most experiments, the improvement achieved by the proposed method is minimal. How can it be demonstrated that the proposed initialization is indeed effective? From my perspective, given the limited parameters in the LoRA adapter, the influence of different initializations on the final performance should be relatively small.

3. Computing the SVD introduces additional computational overhead, which diminishes the efficiency advantage of vanilla LoRA.

---

> ### Author Response · Authors · 2024-11-16
>
> **Motivation behind EVA:**
>
> LoRA initializes the matrix B with zeros and matrix A at random. This has beneficial properties (such as performing well under high learning rates [1]), but leads to a weak and noisy gradient signal at the initial training stage [2]. To obtain a better training signal we initialize the matrix A with the projection onto the principal components, i.e. the directions that explain the most variance. This has the effect that the most information possible is propagated through the linear subspace. The projection can be obtained by performing SVD on inputs to weight matrices (we refer to them as activation vectors for simplicity). Since we need to handle arbitrarily large datasets it is not a good solution to collect and store activations and perform SVD on the full datamatrix. Therefore, we perform SVD incrementally and stop after we do not observe any change in the projection onto the principal components anymore. We clarified this point in the revised version and added the relation to PCA (proof in Appendix H) along with  formulation of the incremental SVD (Section 3.2) accompanied by pseudocode (Algorithm 2) and references to supplementary proofs.
>
> **Presentation:**
>
> Thank you for pointing out the inconsistency in the figures. We updated the fontsize of the left part of Figure 2. Furthermore, we updated Figure 1 to clarify the initialization procedure performed by EVA. It should be a lot more clear now. If the reviewer has any further suggestions for enhancing clarity we will gladly include them into our manuscript.
>
> **Significance of results:**
>
> There is compelling evidence in the literature that proper initialization of the LoRA subspace results in significant improvements [2,3,4]. While the differences may appear small at first glance, it is important to keep in mind that a small difference in average performance requires improvement in several single tasks. As EVA consistently improves upon its competitors on average across all domains, our reported results suggest statistical significance. Furthermore, we added a table that shows that after rank redistribution EVA actually **reduces** the number of trainable parameters while simultaneously improving performance. In the table below we illustrate that EVA improves upon LoRA for the same rank budget while reducing the parameter count.
>
> | Model | Method |  #Trainable | Common sense reasoning | GSM8K | MATH  |
> | --- | --- | --- | --- | --- | --- |
> | LLama-2-7B | LoRA | 18.3M | 82.2 | 59.7 | 10.9 |
> | LLama-2-7B| EVA | **17.3M** | **83.4** | **61.9** | **13.1** |
> | LLama-3.1-8B | LoRA | 20M | 89.2 | 78.3 | 30.1 |
> | LLama-3.1-8B | EVA | **18.9M** | **89.5** | **78.8** | **31.2** |
> | Gemma-2-9B | LoRA | 24.5M | 92.2 | 83.4 | 40.7 |
> | Gemma-2-9B | EVA | **23.1M** | **92.5** | **83.6** | **41.5** |
>
> **Computational Overhead of EVA:**
>
> We disagree that the additional computational overhead diminishes the effect of LoRA fine-tuning. In fact, we added Table 22 in the appendix that demonstrates that initialization with EVA merely constitutes 0.2% of the total training time (in minutes), i.e. initialization takes approx. 70 seconds compared to 8 hours of training for Llama-2-7B on the common sense reasoning tasks. We also show a reduced form of the table below for convenience.
>
> | Method    | Initialization Time | Training Time | % of Training |
> | -------- | ------- |----| --- |
> | PiSSA  | 7.43    | 482.67 | 1.5 |
> | OLoRA | 0.3      | 482.67 | 0.1 |
> | LoRA-GA | 11.7 | 482.67 |   2.4 |
> | EVA | 1.17 | 482.67 | 0.2 |
>
> In this table LoRA-GA is concurrent work on data-driven initialization which we added to our experiments in Table 2 and 3 in the revised version of our manuscript. The only other initialization method that is faster than EVA is OLoRA which is weight-driven and performs worse in our experiments. Therefore, EVA attains a better performance vs complexity trade-off. Finally, as demonstrated above, EVA additionally reduces the parameter count via rank redistribution which further reduces computational complexity. Considering these facts, we believe EVA provides significant contributions to the community.
>
> **References:**
>
> [1] The Impact of Initialization on LoRA Finetuning Dynamics, Hayou et al., NeurIPS 2024
>
> [2] Pissa: Principal singular values and singular vectors adaptation of large language models., Meng, et al. NeurIPS 2024
>
> [3] LoRA-GA: Low-Rank Adaptation with Gradient Approximation, Wang et al., NeurIPS 2024
>
> [4]  OLoRA: Orthonormal Low-Rank Adaptation of Large Language Models, Büyükakyüz et al., arXiv:2406.01775

---

> ### Author Response · Authors · 2024-11-22
>
> Dear reviewer Qji4,
>
> We thank you again for taking the time and effort to help improve our paper and believe we have addressed all your concerns by properly motivating EVA, updating Figure 1, and clearing any doubts of significance of results, since EVA reduces the number of trainable parameters, and providing evidence that overhead induces from EVA is negligible.
>
> We would be grateful for an opportunity to discuss wether there are any pending concerns you can point us to.
>
> Thank you,
> Authors

---

> > ### Author Response · Authors · 2024-11-30
> >
> > Dear reviewer Qji4,
> >
> > In addition to the provided references on initialization of LoRA adapters, we empirically verified that the effect of initialization is substantial. To this end, we measured the distance between learned adapters with EVA and LoRA via cosine similarity and spectral norm. The average distances (cosine/spectral) across three seeds for the different weight matrices in Llama-2 and LLama-3.1-8B are shown below. This result verifies that given a different initalization, the adapters converge to substantially different solutions as there is absolutely no overlap in cosine similarity and a large deviaton in the spectral norm.
> >
> > | Model | Query | Key | Value | Gate | Up | Down |
> > | --- | --- | --- | --- | --- | --- | --- |
> > | LLama-2 | -0.01/4.98 | 0.00/5.00 | 0.01/4.00 | 0.00/4.05 | 0.00/6.64 | -0.00/3.67 | -0.00/ 4.02|
> > | LLama-3.1-8B | -0.00/4.05 | -0.01/5.25 | -0.00/3.83 | -0.01/3.53 | -0.00/6.98 | 0.01/3.37 | -0.00/3.73 |
> >
> > Moreover, to verify that EVA provides more information at the beginning of training, we quantify how much has been learned by measuring the distance between the initial adapters and the final ones after training LoRA and EVA for Llama-2 and Llama-3.1. We again average across three seeds and show distances (cosine/spectral) below. As evident below, EVA's initialization is consistently closer to the final adapters after training than LoRA's. This demonstrates that less information is learned for EVA as it already captures the most possible amount of information in the subspace. We will add both tables in the final version in case of acceptance.
> >
> > | Method | Model | Query | Key | Value | Gate | Up | Down |
> > | --- | --- | --- | --- | --- | --- | --- | --- |
> > | LoRA | LLama-2 |0.51/3.85 | 0.48/4.08 | 0.60/3.10 | 0.59/3.09 | 0.44/5.27 | 0.62/2.83 | 0.61/ 3.13 |
> > | EVA | Llama-2 | **0.62/3.48** | **0.59/3.59** | **0.62/2.90** | **0.62/2.78** | **0.42/4.92** | **0.66/2.61** | **0.67/2.84** |
> > | LoRA | Llama-3.1-8B | 0.51/3.46 | 0.47/3.96 | 0.59/2.93 | 0.61/2.73 | 0.35/5.88 | 0.60/2.58 | 0.59/2.98 |
> > | EVA | LLama-3.1-8B | **0.64/2.93** | **0.61/3.62** | **0.63/2.46** | **0.64/2.27** | **0.41/5.12** | **0.67/2.46** | **0.67/2.71** |

---

> > > ### Author Response · Authors · 2024-12-02
> > >
> > > Dear reviewer Qji4,
> > >
> > > We thank you again for taking the time and effort to help improve our paper.
> > >
> > > Since we are at the end of the extended author-reviewer discussion period, we are reaching out to ask if our response have addressed your remaining concerns.
> > >
> > > Please let us know if you have lingering questions and whether we can provide any additional clarifications today to improve your rating of our paper.
> > >
> > > Thank you,
> > > Authors

---

> > > > ### Comment · Reviewer_Qji4 · 2024-12-02
> > > >
> > > > I have read the authors' responses and the comments from other reviewers. I appreciate the authors' efforts to improve the quality of the paper, however, I still have concerns about the motivation of the proposed method, paper writing and especially the experiments. As such, I tend to maintain my original rating and cannot recommend accepting this paper.

---

> > > > > ### Author Response · Authors · 2024-12-02
> > > > >
> > > > > Der reviewer Qji4,
> > > > >
> > > > > Thank you for your repsponse and investing time in reading our response.
> > > > >
> > > > > To the best of our knowledge we have addressed all the concerns of the initial review.
> > > > > In particular, we have addressed the motivation on EVA, showed that EVA improves performance while reducing the number of trainable parameters, demonstrated the effect of initialzation and also verified that EVA contains a larger constituent of informaion compared to the final solution, making it a better initialzation for LoRA.
> > > > > Finally, we showed that initalization with EVA merely amounts to 0.2% of actual training time, diminishing any doubts about efficiency.
> > > > >
> > > > > We kindly ask the reviewer be more specific about the remaining concerns, as this helps us to improve our manuscript.

---

> > > > > > ### Comment · Reviewer_Qji4 · 2024-12-02
> > > > > >
> > > > > > Hi, I personally feel that the method lacks significance, justification and necessity. This concern seems to be echoed by other reviewers who have also raised similar points in the post-rebuttal stage.

---

### Author Response · Authors · 2024-11-16

We would like to thank all reviewers for their constructive feedback that helps us to significantly improve our manuscript.
We addressed all points raised by the reviewers and added the following changes in the revised manuscript:

- We updated Figure 1 and extended the Introduction to give a better motivation for EVA.
- We extended Section 3 with mathematical formulations of the incremental SVD update plus pseudocode in Algorithm 2 in Appendix F.  We furthermore add an additional proof in Appendix H that the right-singular values are the projection onto the principal components that explain the most variance of the data.
- We added additional results for LoRA-GA (Table 2 and 3), which is concurrent work on data-driven initialization, for common sense reasoning and math tasks. EVA mostly outperforms LoRA-GA on all three model classes..
- We added a comparison of EVA to LoRA in the rsLorA and LoRA+ settings, where EVA consistently outperforms LoRA (Table 9 in Appendix B3).
- We added a more elaborate discussion on performance for $\rho=1$ vs $\rho=2$ (Section 5), along with Table 10 in Appendix B3 which compares performance of both setups on common sense reasoning and math tasks. Further, we added Table 11 in Appendix B3 on trainable parameters showing that **rank-redistribution consistently reduces the number of trainable parameters while improving performance compared to LoRA.**
- We added Table 22 in Appendix F4 on efficiency of obtaining the data-driven initialization. EVA is significantly faster than LoRA-GA and only slightly slower than OLoRA (which is not a data-driven initialization) while exhibiting better performance and reducing the number of trainable parameters.
- We added additional training loss curves (Figure 4) showing the same trend as Figure 2, that EVA leads to faster convergence.

All changes are highlighted in red in the updated manuscript. We are looking forward to an engaging discussion and further feedback from the reviewers to improve our manuscript.

---

### Comment · Area_Chair_TLAf · 2024-11-22
**Interactive Discussions**

Dear Reviewers,

Thank you for your efforts in reviewing this paper. We highly encourage you to participate in interactive discussions with the authors before November 26, fostering a more dynamic exchange of ideas rather than a one-sided rebuttal.

Please feel free to share your thoughts and engage with the authors at your earliest convenience.

Thank you for your collaboration.

Best regards,
ICLR 2025 Area Chair

---

### Meta-Review · Area_Chair_TLAf · 2024-12-19

**Metareview:**

This submission proposes an enhancement to LoRA by initializing new weights through a data-driven approach. Specifically, it computes singular value decomposition (SVD) on mini-batches of activation vectors and uses the resulting right-singular vectors for initialization. Additionally, ranks are re-distributed among weight matrices to maximize explained variance across layers. Various benchmarks are used in the experiments to evaluate the effectiveness of the proposed initialization method.

In the rebuttal, a common concern raised is the motivation behind the proposed method and its marginal performance compared to existing approaches. After reviewing the discussion, the Area Chair aligns with most reviewers and leans towards rejection. However, the authors are encouraged to address these concerns and improve the submission for a future resubmission.

**Additional Comments On Reviewer Discussion:**

As mentioned above, the primary concern raised by the reviewers is the motivation and rationale behind the proposed method, which the rebuttal fails to adequately address. Additionally, several reviewers found the experimental results to be marginal, and this issue remains unresolved in the rebuttal.

---

### Decision · Program_Chairs · 2025-01-22

Reject